# A new process-based and scale-aware desert dust emission scheme for global climate models – Part II: Evaluation in the Community Earth System Model (CESM2)

Danny M. Leung[1], Jasper F. Kok[1], Longlei Li[2], Natalie M. Mahowald[2], David M. Lawrence[3], Simone Tilmes[4], Erik Kluzek[3], Martina Klose[5], and Carlos Pérez García-Pando[6,7]

[1]Department of Atmospheric and Oceanic Sciences, University of California – Los Angeles, Los Angeles, California, USA
[2]Department of Earth and Atmospheric Sciences, Cornell University, Ithaca, New York, USA
[3]Climate and Global Dynamics Laboratory, National Center for Atmospheric Research, Boulder, Colorado, USA
[4]Atmospheric Chemistry Observations and Modeling Laboratory, National Center for Atmospheric Research, Boulder, Colorado, USA
[5]Department Troposphere Research, Institute of Meteorology and Climate Research (IMK-TRO), Karlsruhe Institute of Technology (KIT), Karlsruhe, Germany
[6]Barcelona Supercomputing Center (BSC), Barcelona, Spain
[7]Catalan Institution for Research and Advanced Studies (ICREA), Barcelona, Spain

*Correspondence to*: Danny M. Leung (dannymleung@ucla.edu)

**Abstract**

Desert dust is an important atmospheric aerosol that affects the Earth's climate, biogeochemistry, and air quality. However, current Earth system models (ESMs) struggle to accurately capture the impact of dust on the Earth's climate and ecosystems, in part because these models lack several essential aeolian processes that couple dust with climate and land surface processes. In this study, we address this issue by implementing several new parameterizations of aeolian processes detailed in our companion paper into the Community Earth System Model version 2 (CESM2). These processes include (1) incorporating a simplified soil particle size representation to calculate the dust emission threshold friction velocity, (2) accounting for the drag partition effect of rocks and vegetation in reducing wind stress on erodible soils, (3) accounting for the intermittency of dust emissions due to unresolved turbulent wind fluctuations, and (4) correcting the spatial variability of simulated dust emissions from native to higher spatial resolutions on spatiotemporal dust variability. Our results show that the modified dust emission scheme significantly reduces the model bias against observations compared to the default scheme and improves the correlation against observations of multiple key dust variables such as dust aerosol optical depth (DAOD), surface particulate matter (PM) concentration, and deposition flux. Our scheme's dust also correlates strongly with various meteorological and land surface variables, implying higher sensitivity of dust to future climate change than other schemes' dust. These findings highlight the importance of including additional aeolian processes for improving the performance of ESM aerosol simulations and potentially enhancing model assessments of how dust impacts climate and ecosystem changes.

## 1 Introduction

Desert dust is responsible for over half of the atmospheric mass loading of particulate matter (PM) (Kinne et al., 2006; Kok et al., 2017) and produces multiple impacts on the Earth system. Dust contributes to the aerosol radiative effect and forcings directly by absorbing and scattering solar and terrestrial radiation (Di Biagio et al., 2020; Ke et al., 2022; Adebiyi et al., 2023; Kok et al., 2023), and indirectly by regulating liquid and ice cloud formation (e.g., McGraw et al., 2020; Froyd et al., 2022). Furthermore, dust provides essential nutrients such as iron and phosphorus to terrestrial and ocean ecosystems, thereby promoting biogeochemical activities and enhancing ecosystem carbon uptake (e.g., Mahowald et al., 2010; Hamilton et al., 2020). However, dust also causes pulmonary and cardiovascular diseases, posing a threat to human health (e.g., Esmaeil et al., 2014; Goudie, 2014; Achakulwisut et al., 2019). Despite its significance, global climate models (GCMs) and Earth system models (ESMs) still face challenges in accurately simulating the spatiotemporal distribution of dust aerosols (Zhao et al., 2022), leading to significant uncertainties in the assessments of their climatic impacts (Klose et al., 2021; Li et al., 2021). Current models also struggle to simulate the impacts of how past and future climate and land use changes impact dust emissions (Kok et al., 2023). Therefore, it is critical to improve dust simulations to better predict future dust changes and better simulate dust impacts on climate and climate change.

The difficulty that GCMs and ESMs face in capturing the spatiotemporal variability of atmospheric dust can be attributed to two main factors. First, current dust emission parameterizations in ESMs are likely conceptually incomplete. There is still a limited understanding of dust emission mechanics, and several aeolian processes are not yet included in model parameterizations. For instance, the wind drag partition effect due to the presence of nonerodible roughness elements, interparticle forces involved in soil crusts (Rodriguez-Caballero et al., 2018), and human impacts such as agriculture (e.g., Kandakji et al., 2020; Xia et al., 2022) on dust emission are not accounted for in many existing ESM dust parameterizations. As a result, many ESMs use preferential source masks (e.g., Ginoux et al., 2001; Zender et al., 2003a) to eliminate dust from marginal regions. Second, the existing dust emission parameterizations are not well constrained due to inadequate information and constraints on relevant parameters. For example, past studies show that the dust emission threshold wind speed should be modeled using a median soil particle diameter $D_p$ (Martin and Kok, 2019). Leung et al. (2023) used soil data previous studies to show that the median $D_p$ is about ~130 $\mu$m, in contrast to the existing parameter range of 75–500 $\mu$m (e.g., Zender et al., 2003a; Laurent et al., 2008). Furthermore, many meteorological and land surface variables such as wind speed and soil moisture, which dust emissions are heavily dependent on (Zender et al., 2003a), contain biases and are challenging to model well in ESMs. There is a need to improve dust emission modeling in ESMs by incorporating more physical aeolian processes and setting more accurate parameter constraints.

Additionally, dust modeling in GCMs and ESMs suffers from a grid resolution-dependence problem, especially since dust emissions depend nonlinearly on meteorological and land surface fields (Feng et al., 2022). Coarse GCMs with horizontal resolutions of 100 km cannot capture local-scale (~1km scale) wind maxima, as well as other small-scale meteorological processes such as mesoscale convective systems (MCS) and low-level jets, leading to an underestimation of emissions over specific regions (Ridley et al., 2013; Gliß et al., 2021; Meng et al., 2021). This scale dependence problem is exacerbated by dust emission being a threshold process that scales with friction velocity $u_*$ to the power of ~2–4, resulting in dust emission schemes being further sensitive to inaccuracies in wind speed and other input data (such as soil moisture and vegetation cover). Although some ESMs employ a Weibull distribution to address the subgrid spatial variability of winds (e.g., Menut, 2018), it is challenging to represent the shape parameter $k$ of the Weibull distribution because of a lack of fine-resolution global wind datasets to calibrate a global distribution of $k$ (Tai et al., 2021). Moreover, many GCMs simply do not employ any subgrid wind distribution to address the scale-dependence problem. In addition, other meteorological and

land surface variables such as soil moisture and vegetation also contribute to the scale-dependence problem of dust emissions. To improve the accuracy of simulations and to make ESM dust emission simulations self-consistent across different horizontal grid resolutions, it is crucial to address and mitigate this scale-dependence problem.

In our companion paper (Leung et al., 2023), we presented four improvements to enhance the physical realism of dust emission parameterizations in ESMs. These include: 1) a revised soil median diameter for better estimating the dust emission threshold, 2) a drag partition scheme considering the impacts of both rocks and vegetation in reducing soil erosion by winds, 3) a dust emission intermittency parameterization accounting for boundary-layer turbulent wind fluctuations that initiate and cease dust emissions, and 4) an upscaling approach to correct the spatial variability of dust emissions from native-resolution ESMs to match that of high-resolution dust emission simulations, with the collective aim of these improvements to better capture the subgrid spatial variations of dust emissions. Our implemented scheme contains updated and more comprehensive dust emission processes. We will examine in this study how including more aeolian processes will benefit dust modeling performance.

In this study, we integrate the improved dust emission scheme from Leung et al. (2023) into a premier ESM, the Community Earth System Model version 2 (CESM2). We describe the default and the updated dust emission modules in Sects. 2 and 3, respectively. In Sect. 4, we provide an overview of the observational and reanalysis datasets used to assess the effectiveness of our new scheme, including datasets of dust aerosol optical depth (DAOD), dust PM concentration, and dust deposition flux. In Sect. 5, we then evaluate the new dust emission scheme by comparing the simulations against observations. We summarize our study in Sect. 6.

## 2 Description of CESM2 and its default dust scheme

In this section, we summarize the default schemes and settings in CESM2. Section 2.1 describes the default dust emission scheme in the Community Land Model version 5 (CLM5), the land component of CESM2, including the dust emission threshold scheme (Sect. 2.1.1) and the emission flux parameterization (Sect. 2.1.2). Section 2.2 summarizes the atmospheric dust simulation in the Community Atmosphere Model version 6 (CAM6), the atmospheric component of CESM2, including transport, size distribution, and deposition. Section 2.3 describes the CESM2 configuration in this study.

## 2.1 Default CESM2 dust emission scheme

### 2.1.1 Dust emission threshold scheme

Recent findings indicated that the dust emission process is a double-threshold mechanics problem (Kok et al., 2012; Comola et al., 2019). The fluid threshold, or static threshold $u_{*ft}$ is the threshold friction velocity above which winds initiate emissions, whereas the impact threshold, or dynamic threshold $u_{*it}$ is the threshold friction velocity below which winds are too weak to sustain emissions (Kok et al., 2012). Without considering the soil moisture effect $f_m$ on enhancing the fluid threshold (Eq. 1), $u_{*it}$ is $\sim 80\%$ of the "dry" fluid threshold $u_{*ft0}$ (Sect. 3.4; Kok et al., 2012; Comola et al., 2019). However, if substantial soil moisture is present (e.g., over semiarid regions), the difference between $u_{*it}$ and $u_{*ft}$ could be very large (see Fig. S3a–b) since $u_{*it}$ is not a function of soil moisture (see Eq. 11). Nevertheless, most dust emission schemes in global and regional models employ $u_{*ft}$ as the single threshold for both the initiation and termination of dust emission flux in models (Menut et al., 2013; Klose et al., 2021; Tai et al., 2021; Li et al., 2022; LeGrand et al., 2023), which could be problematic (see Sect. 3.4). The current $u_{*ft}$ parameterization scheme assumes that $u_{*ft}$ is dependent on the particle size distribution (PSD) and the

amount of moisture in the soil (Iversen and White, 1982; Marticorena and Bergametti, 1995; Zender et al., 2003a). $u_{*ft}$ is modeled as follows:

$$u_{*ft} = u_{*ft0}(D_p, \rho_a)f_m(w) \tag{1}$$

where $u_{*ft0}$ is the "dry" fluid threshold friction velocity (in m s$^{-1}$) with no soil moisture on a smooth and bare surface. $u_{*ft0}$ is a function of $D_p$, which in this study will be the median diameter of a mixed soil, and $\rho_a$ the air density (kg m$^{-3}$). $f_m$ is the correction factor for the presence of gravimetric soil moisture $w$ (kg water / kg soil); $f_m \geq 1$ (mainly over semiarid regions) such that soil moisture protects soil particles

from being lifted. $u_{*ft}$ is the "wet" fluid threshold accounting for the moisture effect. We note that other factors can also affect $u_{*ft}$, such as salt concentration, electrostatics (Kok and Renno, 2009), and surface crusts (Rodriguez-Caballero et al., 2022), but most of these factors are not included in most modeling studies because they are not well understood and modeled (Shao et al., 2011; Foroutan et al., 2017).

The variables in Eq. 1 are computed as follows. First, $u_{*ft0}$ is parameterized in CLM following

the Iversen and White (1982; hereafter I&W82) scheme (Oleson et al., 2013) as a function of $D_p$ and $\rho_a$. CLM5 uses a global soil diameter of $D_p = 75\,\mu$m that corresponds to the lowest emission threshold (Zender et al., 2003), and thus the spatiotemporal variability of $u_{*ft0}$ purely follows that of $\rho_a$. Then, CLM5 calculates $f_m$, the effect of soil moisture on enhancing $u_{*ft}$ following Fécan et al. (1999). $f_m$ is a function of the difference between the gravimetric soil moisture $w$ (kg water / kg soil) and a threshold value $w_t$.

$f_m > 1$ once gravimetric moisture is bigger than $w_t$, leading to an increase in $u_{*ft}$ (see Oleson et al., 2013; also see the CLM5 technical documentation at github: https://escomp.github.io/ctsm-docs/versions/master/html/tech_note/Dust/CLM50_Tech_Note_Dust.html):

$$f_m = \sqrt{1 + 1.21[100(w - w_t)]^{0.68}} \qquad \text{for } w > w_t \tag{2a}$$
$$w_t = 0.01a(17f_{clay} + 14f_{clay}^2) = 0.01a(0.17(\%\text{clay}) + 0.0014(\%\text{clay})^2) \tag{2b}$$

where $f_{clay} \in [0,1]$ is the clay fraction, $\%\text{clay} = 100f_{clay}$ is the clay percentage, and $a$ is a tunable constant typically around 0.5–2 ($a = 1$ was adopted in Kok et al., 2014b) and was set to be $1/f_{clay}$ for tuning purposes in CLM5 (Oleson et al., 2013). The threshold moisture $w_t$ increases with $f_{clay}$, since clay efficiently adsorbs water such that more moisture is required to enhance $u_{*ft}$. Note that we express $w$ as a fraction (kg water / kg soil), while previous dust modeling studies usually expressed gravimetric soil

moisture $w'$ in % (i.e., $w' = 100w$; Fécan et al., 1999). Eq. 2 is thus identical to those in other dust modeling studies (e.g., Kok et al., 2014b; Foroutan et al., 2017). CLM5 currently uses the soil texture dataset from the Food and Agriculture Organization (FAO) for $f_{clay}$, but will likely update to more recently developed datasets (e.g., SoilGrids; Hengl et al., 2017) in the future.

**2.1.2 Dust emission flux calculation**

After obtaining $u_{*ft}$, there are multiple published dust emission equations that relate the global dust emission flux to a given $u_{*ft}$ and friction velocity $u_*$ (Gillette and Passi, 1988; Shao et al., 1996; Ginoux et al., 2001; Zender et al., 2003a; Klose et al., 2014). The default CLM5 uses the Zender et al. (2003a) scheme (hereafter Z03), also known as the DEAD scheme. Z03 is based on the Marticorena and

Bergametti (1995) scheme and the White (1979) equation for saltation, which are used by many other global models (e.g., Foroutan et al., 2017; Meng et al., 2021; Klose et al., 2021; Wu et al., 2021). The Z03 dust emission equation has a form of:

$$F_d = STC_{MB}\varphi f_{bare} \frac{\rho_a}{g} u_{*s}^3 \left(1 - \frac{u_{*t}^2}{u_{*s}^2}\right)\left(1 + \frac{u_{*t}}{u_{*s}}\right) \qquad \text{for } u_{*s} > u_{*t} \tag{3}$$

where $u_{*s}$ is the soil surface friction velocity (m s$^{-1}$; $u_{*s} = u_*$ in default CESM2; Oleson et al., 2013), $F_d$

is the dust emission flux (kg m$^2$ s$^{-1}$), $u_{*t}$ is the dust emission threshold (m s$^{-1}$; $u_{*t} = u_{*ft}$ in Z03), $T = 5 \times 10^{-4}$ is a proportionality constant in CLM5 (Oleson et al., 2013), $C_{MB} = 2.61$ is the saltation constant (Oleson et al., 2013), and $\varphi$ is the sandblasting efficiency (m$^{-1}$). $S$ is the source function used to

characterize the preferential source regions where fluvial sediment accumulates and to scale down the emission flux out of desert regions (Zender et al., 2003b). $f_{bare}$ is the bare land fractional area; CLM5 uses a simple parameterization in which $f_{bare}$ is a function of vegetation area index (VAI) defined as a sum of the leaf area index (LAI) and stem area index (SAI), so that dust emission scales down linearly with VAI and drops to zero when VAI > $VAI_{thr}$ (= 0.3; Mahowald et al., 2010; Kok et al., 2014b):

$$f_{bare} \propto (1 - f_v) \tag{4}$$

where $f_v$ = VAI/$VAI_{thr}$ is the vegetation cover fraction. Other factors also considered to decrease bareness of the land such as the grid fraction of lake, snow cover, and the soil liquid content (see Oleson et al., 2013; also see Eq. 13 in Zender et al., 2003a).

## 2.2 Atmospheric dust simulation

The land model (CLM5) simulates the dust emission as a function of soil and land properties (following Sect. 2.1), and the atmospheric model (CAM6) then takes the emission fluxes from the land model and simulates the transport, deposition, and microphysics (e.g., coagulation) of dust aerosols. The tropospheric modal aerosol model (MAM4) in CAM6 contains four aerosol modes (Liu et al., 2016): the Aitken mode (dust, sulfate, secondary organic matter, and sea salt), the accumulation mode (sulfate, secondary organic matter, primary organic matter, black carbon, sea salt, and dust), coarse mode (dust, sea salt, and sulfate), and the primary carbon mode (primary organic matter and black carbon). The size distribution of each mode is assumed to be log normal with fixed geometric standard deviations (GSDs) for each mode as 1.6 (Aitken), 1.6 (accumulation), 1.2 (coarse), and 1.6 (primary carbon). The geometric median diameters (GMDs) of the aerosol modes are then simulated accordingly. The emitted dust size distribution is derived from a parameterization based on brittle fragmentation theory (Kok, 2011) with the respective ratios of 0.1 %, 1.0 %, and 98.9 % for Aitken, accumulation, and coarse modes (Li et al., 2022). Note that the coarse mode in CAM6 includes dust up to a diameter of ~10 $\mu$m and therefore misses the super-coarse dust ranged between 10 and 50 $\mu$m, and recent studies have therefore attempted to add more modes or particle bins to CAM (e.g., Ke et al., 2022; Meng et al., 2022). CAM6 then uses a tracer advection scheme to transport dust aerosols (Neale et al., 2012). Aerosols in each mode are transported as an internal mixture of the species present, with its physical properties (e.g., optical properties and density) predicted based upon the volume fraction of each species, while aerosol species from different modes are externally mixed. CAM6 simulates the removal of aerosols via dry deposition and wet deposition. Dry deposition includes turbulent and gravitational settling, as described in Zender et al. (2003a). Wet deposition includes in-cloud and below-cloud scavenging (Neale et al., 2012) of aerosols. The below-cloud precipitation provides rain and snow scavenging as a first-order process, which is the product of aerosol mass mixing ratio, precipitation flux, and scavenging coefficient (Dana and Hales, 1976). The in-cloud scavenging calculation assumes aerosols inside the cloud water are removed by precipitation, in proportion to the fraction of cloud water converted to rain through coalescence and accretion (Neale et al., 2012). The wet deposition rate depends on various factors including the prescribed dust hygroscopicity (0.068; Scanza et al., 2015) and the scavenging coefficient (0.1; Neale et al., 2012).

## 2.3 Coupled model configuration

The above dust emission equations are embedded into the CESM2.1 (hereafter CESM2; Danabasoglu et al., 2020), a coupled ESM with multiple earth system components including atmosphere, land, ocean, sea ice, etc. We use a component set (FHIST) of CESM2 that couples the land model component (CLM5) with the atmospheric component (CAM6), while other components (ocean, sea ice, glacier/land ice, etc.) are not active. The dust emission equations are simulated in CLM5. The meteorological and land surface variables that dust emission depends on, such as $u_*$, $w$, and $\rho_a$, are simulated by CLM5 and CAM6. The vegetation phenology in this configuration is prescribed from

remote-sensing data (satellite phenology) in CLM5. We implement the new parameterizations described in Sect. 3 into CLM5 and evaluate the simulation performance with these new additional physics in Sect. 5. In the model configuration that we utilize in this study, atmospheric variables (e.g., wind and temperature) are simulated with a 30-minute timestep and are nudged every 3 hours toward the assimilated meteorology from the Modern Era Retrospective analysis for Research and Applications v2 (MERRA-2; Gelaro et al., 2017) obtained from the Global Modeling and Assimilation Office (GMAO). CAM6 has a default vertical resolution of 32 levels, and in this study, both CLM5 and CAM6 use a default horizontal resolution of 0.9°×1.25° with a default time step of 30 minutes. In this study, simulations are performed for 2003–2008 with 2003 discarded as a spinup year and 2004–2008 used for analysis purposes. We choose this time period because most of the observed and reanalysis datasets used for evaluation (described in Sect. 4) contain data over 2004–2008.

## 3 Modifications to the CESM2 dust emission scheme

In this section, we summarize the main improvements to the dust emission scheme proposed in Leung et al. (2023) and describe the new dust-related variables that these changes create.

### 3.1 A new physical dust emission equation

In this study, we first replace the Z03 dust emission equation with a more physical dust emission equation from Kok et al. (2014b; hereafter K14), which has been adopted by a number of other global and regional models (Evan et al., 2015; Ito and Kok, 2017; Mailler et al., 2017; Li et al., 2021; Tai et al., 2021), as the base dust emission scheme for additional modifications in Sect. 3.2–3.5. One key difference between K14 and Z03 is that Z03 uses a spatial source function $S$ to tune the dust emission flux to capture the magnitude of observed dust concentrations. $S$ essentially quantifies the soil erodibility, defined as the efficiency of a soil in producing dust aerosols under a given wind stress (Zender et al., 2003b). The need for this source function indicates that Z03 is unable to capture the physical processes that determine soil erodibility across the globe. The largest difference between Z03 and K14 (and our scheme in Leung et al., 2023) is that Kok et al. (2014a) argued that soil erodibility ($C_d$ in K14) can be directly related to soil aridity as characterized by the standardized fluid threshold:

$$u_{*st} = u_{*ft}\sqrt{\rho_a/\rho_{a0}} \tag{5a}$$

because more erodible soils generally tend to have lower $u_{*ft}$ and moisture values. $u_{*st}$ is a pure function of moisture $w$ since $\sqrt{\rho_a}$ cancels the $\rho_a^{-0.5}$ dependence in $u_{*ft0}$ (in I&W82 or Eq. 6 below). Note that $u_{*st}$ is only a proxy of $u_{*ft}$ and is not used as a real emission threshold (i.e., $u_{*st}$ should not be used as $u_{*t}$ in Eq. 5c). Then, the soil erodibility coefficient (or dust emission coefficient) in K14 is a pure function of $u_{*st}$:

$$C_d = C_{d0}\exp\left(-C_e\frac{u_{*st}-u_{*st0}}{u_{*st0}}\right) \tag{5b}$$

where $C_{d0} = (4.4 \pm 0.5) \times 10^{-5}$, $C_e = 2.0 \pm 0.3$, and $u_{*st0} = 0.16$ m s$^{-1}$ are constants. The soil erodibility $C_d$ increases with the dryness of the soil and is a pure function of the standardized fluid threshold $u_{*st}$ (and thus $u_{*ft}$) and the soil moisture effect $f_m$. Following Kok et al. (2014b), the dust emission flux (kg m$^{-2}$ s$^{-1}$) is:

$$F_d = \eta C_{tune}C_d f_{bare}f_{clay}'\frac{\rho_a(u_{*s}^2-u_{*t}^2)}{u_{*st}}\left(\frac{u_{*s}}{u_{*t}}\right)^\kappa \qquad \text{for } u_{*s} > u_{*t} \tag{5c}$$

Where $u_{*s}$ is the soil surface friction velocity ($= u_*$ in K14; to be detailed in Sect. 3.3), $u_{*t} = u_{*ft}$ was assumed by K14, $\kappa = C_\kappa\frac{(u_{*st}-u_{*st0})}{u_{*st0}}$ is the fragmentation exponent as a function of $u_{*st}$ quantifying the

sensitivity of $F_d$ to $u_*$, $C_\kappa = 2.7 \pm 1.0$ is a constant, $C_{tune} = 0.05$ is the proportionality constant (previously set in Kok et al., 2014b to scale their global K14 emission to the same global Z03 emission), $f_{clay}'$ is the soil clay fraction $f_{clay}$ but capped at 0.2 (i.e., $f_{clay}' \in [0, 0.2]$), and $\eta$ is the intermittency factor (= 1 in K14; to be detailed in Sect. 3.4). The biggest difference between Z03 and K14 is that the spatiotemporal variability of the K14 dust emissions is much more sensitive to the emission threshold $u_{*ft}$ and the moisture $w$ than Z03 (since, from Eq. 5b, $C_d$ increases exponentially with $u_{*st}$). K14 showed improvements compared with Z03 when evaluated against ground-based dust AOD measurements (Kok et al., 2014b; Li et al., 2022). Same as Z03, $\text{VAI}_{thr}$ was set to be 0.3 in the K14 scheme. In the Leung et al. (2023) scheme, however, we set $\text{VAI}_{thr} = 1$ mainly because observations show that dust is emitted from semiarid regions with VAI > 0.3 (e.g., Okin, 2008). Using $\text{VAI}_{thr} = 1$ will thus enable emissions from more marginal dust source regions, which reduces the spatial contrast of dust emissions between hyperarid and semiarid regions.

**3.2 A revised dust emission threshold description**

Based on the K14 scheme, the first proposed change by Leung et al. (2023) is to update a new representation of the effects of soil particle sizes to the modeling of the emission threshold. This includes simplifying the dust emission threshold parameterization and updating the soil particle diameter in the threshold scheme.

Following Leung et al. (2023), we first employ an alternative dust emission threshold scheme by Shao and Lu (2000; hereafter S&L00), which is derived from a more physical approach, is computationally much simpler than I&W82, and produces a $u_{*ft0}$ that is slightly more sensitive to $D_p$. S&L00 is given as:

$$u_{*ft0} = \sqrt{A(\rho_p g D_p + \gamma/D_p)} \, \rho_a^{-0.5} \tag{6}$$

where $A = 0.0123$ and $\gamma = 1.65 \times 10^{-4}$ kg s$^{-2}$ are empirical constants accounting for the magnitude of interparticle forces. S&L00 has a parabolic shape as a function of $D_p$, and $D_p \sim 80\ \mu m$ corresponds to the smallest $u_{*ft0}$ of around 0.2 m s$^{-1}$ (contingent upon the values of $\rho_p$, $\gamma$, and $\rho_a$). For larger sizes soil particles are heavier to lift; for smaller sizes soil particles are more strongly bound by interparticle forces. The S&L00 threshold scheme largely simplifies the I&W82 scheme by dropping the $u_{*ft}$ dependence on the particle Reynold's number $\text{Re}_p$ and avoids the need of using an iterative method to calculate $u_{*ft}$ (Oleson et al., 2013). We thus replace I&W82 with S&L00 in this study for CLM5 (following Leung et al., 2023). Then, $u_{*ft}$ is modeled following Eqs. 1–2 with the soil moisture effect.

For the soil moisture effect, instead of using $a = 1$ following K14, we assign $a = 2$ for our scheme for the soil moisture effect in this paper. We use a slightly larger $a$ than K14 and our previous study (Leung et al., 2023) mainly because CLM5 in CESM2 has higher soil moisture across most of the globe than other soil moisture data, such as MERRA-2/NOAH-MP (Gelaro et al., 2017) and CESM1/CLM4 (see a global soil moisture comparison in Fig. S1). However, our choice of $a$ in this paper is generally smaller than the choice of $a = 1/f_{clay}$ in CESM2's default Z03 scheme. Given $f_{clay}$ from the FAO database typically ranges between 0.1 and 0.4, using $a = 1/f_{clay}$ gives bigger values of $a$ ranging from 2.5 to 10, generally mitigating the soil moisture effect on dust emissions in Z03. For our experiments using the K14 and Z03 schemes (Sect. 5), we will maintain all the default parameter values in CLM5 (e.g., $a = 1$ in K14 and $a = 1/f_{clay}$ in Z03) and use $a = 2$ for our scheme in this paper.

Then, we follow Leung et al. (2023) and employ a globally constant soil median diameter $D_p$ of $127\ \mu m$ for S&L00. Default CLM5 followed Zender et al. (2003a) and used a globally constant soil particle diameter of $D_p = 75\ \mu m$ in I&W82, based on the argument that it is the optimal particle size that is the easiest to lift ($D_p = 75\ \mu m$ corresponds to the smallest $u_{*ft0}$ in I&W82; see discussions in Kok et

al., 2012). However, Martin and Kok (2019) showed that for mixed sandy soils (i.e., soils with multiple sizes of soil particles mixed together), $u_{*ft}$ should be a function of the median particle diameter of the soil PSD instead of the optimal particle size that produces the smallest $u_{*ft}$ possible; we thus assume here that $u_{*ft}$ for soils containing fine particles is also determined by the median particle diameter because emission of dust aerosols from these soils is driven by impacts of saltating sand particles (e.g., Shao et al., 1993). Leung et al. (2023) then used soil PSD observations from a suite of 14 in-situ soil studies (47 data points) to show that the median $D_p$ of the soil PSD measurements over arid regions were within a range of 40–250 $\mu$m. Regression analysis showed insignificant relationships between $D_p$ and other soil textures and properties, which indicated that the limited variability of the soil dataset did not allow us to precisely define the global $D_p$ distribution and thus its impact on the global distribution of the emission thresholds. Thus, Leung et al. (2023) simplified and approximated the global median $D_p$ by taking a mean across all the $D_p$ observations, which was 127 $\mu$m. Leung et al. (2023) also showed that the $D_p$ uncertainty range of 40–250 $\mu$m translates to a $u_{*ft0}$ range of 0.204–0.268 m s$^{-1}$ using S&L00, much smaller than the magnitude of $u_{*ft}$ which goes beyond 1 m s$^{-1}$. Thus, Leung et al. (2023) argued it was reasonable to simplify and approximate the global median $D_p$ by taking a mean across all the $D_p$ observations, which was 127 $\mu$m. In this study, we introduce the use of $D_p = 127$ $\mu$m as a global constant for the threshold schemes because it is conceptually more correct than using the optimal diameter of $D_p = 75$ $\mu$m; however, the resulting value of $u_{*ft0}$ using the S&L00 scheme is 0.215 m s$^{-1}$, which is similar to $u_{*ft0} = 0.204$ m s$^{-1}$ using $D_p = 75$ $\mu$m with the I&W82 scheme in Z03.

### 3.3 A wind drag partition scheme for reduced wind stress due to rocks and vegetation

The second modification we proposed in Leung et al. (2023) is to include the effect of wind drag partitioning due to the presence of surface obstacles or roughness elements, such as vegetation, rocks, pebbles, and gravel, which protect the soil surface from wind erosion by absorbing part of the surface wind momentum. We account for the drag partitioning in the soil surface friction velocity $u_{*s}$:

$$u_{*s} = u_* F_{eff} \tag{7}$$

where $F_{eff} \in [0,1]$ is the drag partition factor, the fraction of wind drag available for wind erosion, which is reduced by wind momentum absorption by surface obstacles (rocks and plants). In the following, we describe the Leung et al. (2023) drag partition scheme, which combines the effects of surface roughness due to rocks (Marticorena and Bergametti, 1995) and vegetation (Okin, 2008) to parameterize $F_{eff}$.

Leung et al. (2023) and previous studies (e.g., Menut et al., 2013; Klose et al., 2021) used the aeolian roughness length $z_{0a}$ to represent the roughness of rocks. $z_{0a}$ represents small-scale objects/obstacles of length scales of 1–10 m and is different from the typical aerodynamic momentum roughness length $z_0$ that represents the orography, terrain, and large-scale canopy roughness (Prigent et al., 2012; Menut et al., 2013). Leung et al. (2023) used the global aeolian $z_{0a}$ dataset from Prigent et al. (2005) (hereafter Pr05), which contains the climatological $z_{0a}$ (12 monthly values per grid) derived from the backscatter coefficient at 5.3 GHz measured by the European Remote Sensing (ERS) satellite. Because $z_{0a}$ quantifies the roughness of both rocks and vegetation, we take the minimum value out of 12 months for all grids to obtain an aeolian $z_{0a}$ map to eliminate the effect of vegetation as much as possible. Furthermore, we apply this map over regions with VAI < 1, where the backscatter signal is mainly generated by rocks with lower contribution from vegetation roughness. Then, Marticorena and Bergametti (1995; hereafter M&B95) previously derived a parameterization to quantify the drag partition effect $f_{eff,r} \in [0,1]$ of obstacles as a function of $z_{0a}$, which drops from one to zero as nonerodible roughness elements becomes more abundant over a surface (Darmenova et al., 2009):

$$f_{eff,r} = 1 - \frac{\ln\left(\frac{z_{0a}}{z_{0s}}\right)}{\ln\left[b_1\left(\frac{X}{z_{0s}}\right)^{b_2}\right]} \qquad (8)$$

where $z_{0s} = 2D_p/30$ is the smooth roughness length (Sherman, 1992; Farrell and Sherman, 2006; Pierre et al., 2014b; Klose et al., 2021), and $b_1 = 0.7$ and $b_2 = 0.8$ are empirical constants (Darmenova et al., 2009). $X$ is the distance downstream the location of an obstacle, a length parameter that roughly scales with the internal boundary layer (IBL) height $\delta$ (Marticorena and Bergametti, 1995). Previous studies used different $X$ values, from $X = 0.1$ m for small, dense blocks (0.025 m of height) in wind tunnel experiments (Marshall, 1971; Marticorena and Bergametti, 1995) to $X = 122$ m for shrubs (MacKinnon et al., 2004). $X$ thus should vary with land type and implicitly with space and time (e.g., Foroutan et al., 2017), but most dust modeling studies have thus far used a globally constant of $X$ for simplicity. Leung et al. (2023) used $X{\sim}\delta{\sim}10$ m for rocks, which is within the range of parameter choices, assuming the obstacles are a few meters apart and the IBL usually gets to a few meters high. We thus use the Pr05 global $z_{0a}$ to obtain the rock drag partitioning $f_{eff,r}$, as shown in Fig. S2a for CLM5.

For vegetation drag partitioning, Leung et al. (2023) used the Okin (2008; hereafter O08) formulation, later simplified by (Pierre et al., 2014; hereafter P14) for GCMs, for modeling vegetation drag partitioning as a single function of VAI. $f_{eff,v}$ drops with increasing VAI:

$$f_{eff,v} = \frac{K + f_0 c}{K + c} \qquad (9a)$$

$$K = 2\left(\frac{1}{f_v} - 1\right) \qquad (9b)$$

where $f_{eff,v} \in [f_0, 1]$ is the area-averaged plant drag partitioning, $K$ (dimensionless) is the normalized mean gap length between obstacles (plants), and $f_0 = 0.32$ and $c = 4.8$ are constants (Leung et al., 2023). As the land gets more densely covered by vegetation, $K \rightarrow 0$ and $f_{eff,v} \rightarrow f_0$. The normalized mean gap length between obstacles $K$ is a function of vegetation cover fraction $f_v = \text{VAI}/\text{VAI}_{\text{thr}}$ (Leung et al., 2023), which is more valid for small VAI (plants are further apart and do not overlap each other). We thus only apply this model over dust emission regions (VAI $\leq$ VAI$_{\text{thr}}$). VAI is thus the only input for Eq. 9. Using VAI (= LAI + SAI) that include both leaf and stem areas, this scheme is accounting for drag partitioning due to both green and brown vegetation. Fig. S2b shows the resulting 2004–2008 mean global $f_{eff,v}$ map in CLM5.

After obtaining both the static $f_{eff,r}$ map for rocks and the time-varying $f_{eff,v}$ map for vegetation, we combine the two drag partition sources to capture and represent the total drag partition effect for dust emission. Leung et al. (2023) obtained the fractions of a grid consisting of areas dominated by rocks and areas dominated by plants from the European Space Agency Climate Change Initiative (ESA CCI) dataset (ESA, 2017; https://www.esa-landcover-cci.org/?q=node/164, last access: 21 June 2022). The land cover product classifies the land cover of the whole globe into 37 categories (Li et al., 2018), with relevant land cover over arid regions such as shrub, herbaceous, sparse vegetation, cropland, grassland, as well as consolidated (gravels and rocks) and unconsolidated (soil) bare land. Leung et al. (2023) proposed to parameterize the total dust emission flus $F_d$ for each grid box as a function of its fractional rock area $A_r$ and fractional vegetation area $A_v$:

$$F_d = F_d(u_* F_{eff}) = A_r F_{d,r} + A_v F_{d,v} = A_r F_d(u_* f_{eff,r}) + A_v F_d(u_* f_{eff,v}) \qquad (10a)$$

where $F_{eff}$ is the hybrid drag partition factor. Leung et al. (2023) further formulated the hybrid drag partition factor $F_{eff}$ that encapsulates both rock and vegetation partition effects for these ESMs:

$$F_{eff}{}^3 = A_r f_{eff,r}{}^3 + A_v f_{eff,v}{}^3 \qquad (10b)$$

where $F_{eff}$ is simply the weighted mean of drag partition effects, and the exponent of three is the dust emission exponent ($\kappa + 2$) of ~3 over deserts. An advantage of this weighted mean approach is that it

produces a very smooth transition of the drag partition effect from a rock-dominated regime (e.g., the Sahara) to a plant-dominated regime (e.g., the Sahel), following the transition in land cover. We use Eq. 10 to obtain the global time-varying $F_{eff}$. Fig. 1a shows 2004–2008 mean of $F_{eff}$ in CLM5, with more
grassy areas resembling $f_{eff,v}$ (e.g., the Southern Hemisphere, the U.S., the Tibetan Plateau, etc.) and barer areas resembling $f_{eff,r}$ (e.g., the Dust Belt).

**3.4 A dust emission intermittency scheme**

Our third modification is to account for the effects of boundary-layer turbulent fluctuations on dust emission intermittency. Dust emission intermittency exists because saltation is driven by high-frequency turbulent surface winds (with frequencies of ~ 1 minute or less), which exhibit strong spatiotemporal fluctuations in speed and direction. Instantaneous winds can thus pass within timescales much shorter than a model time step (e.g., the CESM2 time step is about ~30 minutes with a ~100 km grid size) across both
the fluid (static) threshold $u_{*ft}$ for initiating saltation and the impact (dynamic) threshold $u_{*it}$ for ceasing saltation (Martin and Kok, 2018). Consequently, saltation can be highly intermittent (Comola et al., 2019), with pronounced variability in timescales of seconds to hours (Dupont et al., 2013). However, existing dust emission parameterizations describe saltation as uniform in time and space and driven by a constant downward momentum flux within a model time step (typically 30 minutes for CESM2). Neglecting
intermittent dust emissions in current models thus likely degrades the accuracy of dust emission simulations for arid regions during low-wind periods (when $u_{*s} < u_{*ft}$), but especially for marginal dust source regions since $u_{*ft}$ values are much greater than $u_{*it}$ in high moisture regions (using $u_{*ft}$ to model dust will strongly underestimate dust emissions).
Since ESMs cannot explicitly resolve high-frequency turbulent fluctuations, C19 employed
turbulent statistics to estimate the effect of high-frequency turbulent winds on generating dust emissions within a time step. Note that the C19 scheme focuses on incorporating the effect of turbulent wind fluctuations on the saltation-driven dust emission; it does not address the convective turbulent dust emission (CTDE) with direct aerodynamic lifting of dust particles from the land surface, as addressed by other studies (e.g., Klose et al., 2014). Here we briefly describe the C19 scheme that accounts for the
turbulence effect on intermittent dust emissions. C19 first formulates $u_{*it}$ as a linear function of $u_{*ft0}$ (Kok et al., 2012) from S&L00:
$$u_{*it} = B_{it} u_{*ft0} \tag{11a}$$
where $B_{it} = 0.82$ is assumed to be a global constant. Dust emission intermittency happens when $u_{*s}$ lies between both thresholds ($u_{*it} < u_{*s} < u_{*ft}$). If $u_{*s}$ within a model time step has a value between $u_{*it}$
and $u_{*ft}$, there will be small and fluctuating emission fluxes in reality, while LSMs using a $u_{*ft}$ scheme would predict zero emission within a model time step, thereby underestimating the emissions. Many field-based studies showed that saltation flux sustains as long as the wind speed is above the dynamic threshold, i.e., $u_{*s} > u_{*it}$ (Sørensen, 2004; Durán et al., 2011; Ho et al., 2011; Martin and Kok, 2017). Therefore, it is important for climate models to employ $u_{*it}$ instead of $u_{*ft}$ in the dust emission equation. C19 thus
updates K14 by setting all $u_{*t}$ terms in Eq. 5c as $u_{*it}$ instead of $u_{*ft}$:
$$F_d = \eta C_{tune} C_d f_{bare} f_{clay}' \frac{\rho_a (u_{*s}^2 - u_{*it}^2)}{u_{*it}} \left(\frac{u_{*s}}{u_{*it}}\right)^\kappa \qquad \text{for } u_{*s} > u_{*it} \tag{11b}$$
where $C_d = C_d(u_{*st})$, $\kappa = \kappa(u_{*st})$, and $u_{*st} = u_{*ft}\sqrt{\rho_a/\rho_{a0}}$ is the same standardized fluid threshold as in K14. Note that the denominator $u_{*st}$ in Eq. 5c is replaced with $u_{*it}$ in Eq. 11b (following Leung et al., 2023). Because $u_{*it} < u_{*ft}$, employing $u_{*it}$ in the dust emission equation in Eq. 11b allows more small
emission fluxes over the marginal source regions that are otherwise missed by employing $u_{*ft}$ as the threshold. Also, we follow Leung et al. (2023) to cap $\kappa$ at three in Eq. 5c since a large $\kappa$ (e.g., $> 10$) combined with a small $u_{*it}$ will occasionally produce unrealistically high emissions over semiarid regions

(which would not happen when using K14 with a large $u_{*ft}$ over semiarid regions). The intermittency factor $\eta \in [0,1]$ denotes the fraction of time within an ESM time step (e.g., 30 minutes for CESM2) that saltation and dust emission are active (see a complete description of $\eta$ in Leung et al., 2023):

$$\eta = \eta(u_{*s}, \sigma_{\tilde{u}_s}, u_{*it}, u_{*ft}) \tag{11c}$$

$\eta$ is formulated as a function of the time-step (30-minutes) mean $u_{*s}$, $u_{*it}$, $u_{*ft}$, as well as the time-step standard deviation $\sigma_{\tilde{u}_s}$ of instantaneous wind $\tilde{u}_s$ at the typical saltation height of $z_{sal} = 0.1$ m (Leung et al., 2023). The instantaneous fluctuation $\sigma_{\tilde{u}_s}$ is dependent on the wind shear and buoyancy of that time step as quantified using the similarity theory (Panofsky et al., 1977; Comola et al., 2019; Leung et al., 2023). $u_{*s}$ and $\sigma_{\tilde{u}_s}$ together control how frequently the instantaneous $\tilde{u}_s$ will sweep across the thresholds in a time step. The relationship between $u_{*s}$ and $\eta$ was portraited in Fig. 6a of Leung et al. (2023). Basically, $\eta$ approaches one when $u_{*s} - \sigma_{\tilde{u}_s} \gg u_{*ft}$ (continuous emission as the instantaneous wind distribution does not cross the threshold), approaches 0 when $u_{*s} + \sigma_{\tilde{u}_s} \ll u_{*it}$ (no emission), and values between zero and one when $u_{*it} < u_{*s} < u_{*ft}$ (intermittent emission as $\tilde{u}_s$ sweeps through the thresholds that initiate and terminate dust emission). Figure 1b shows the 2004–2008 averaged intermittency factor $\eta$. Figure S3 shows the 2004–2008 averaged global distribution of $u_{*it}$, $u_{*ft}$, and $u_{*ft}/u_{*it} (= f_m/B_{it})$ which shows the spatial pattern of the moisture effect $f_m$.

### 3.5 An upscaling correction map for coarse-grid simulations

The final modification in Leung et al. (2023) intends to address the long-standing issue of grid-resolution dependence of ESM-modeled dust emissions (Ridley et al., 2013; Feng et al., 2022; Meng et al., 2022). The grid-scale dependence issue exists because ESMs normally use coarse gridboxes of ~100 km to simulate dust emission, which depends on local-scale processes with typical length scales smaller than 1 km (Marsham et al., 2012; Heinold et al. 2013; Ridley et al., 2013). ESMs with horizontal grid resolutions of ~100 km likely fail to capture locally high emissions because the coarse meteorological and land surface fields used in the emission schemes are smoothed and do not accurately represent the subgrid variability of dust emissions within a 100 km grid (Feng et al., 2022). It is generally believed that the higher the horizontal resolution of an ESM, the better it simulates the local spatial variability of emissions and captures the locally high emission peaks (Ridley et al., 2013). Moreover, dust emission has nonlinear dependencies on multiple variables, especially $u_{*s}$ ($F_d \propto u_{*s}^{\kappa+2} \sim u_{*s}^3$ typically over deserts); as such, capturing the subgrid high wind peaks will result in more emissions in a high-resolution simulation since the sensitivity $\partial F_d / \partial u_*$ is much stronger toward the higher end of $u_*$. Thus, simulating dust in finer horizontal resolutions will generally result in higher global dust emission fluxes (Ridley et al., 2013). The grid-scale dependence problem here thus means that the simulated global dust emission maps are grid-resolution dependent and possess different magnitudes and spatiotemporal variability across resolutions. Linearly interpolating the input variables, such as $u_{*s}$, to calculate dust emissions would be inaccurate as it is different from an area-weighted average of high-resolution dust emissions per se (Ridley et al., 2013). There is a need to better upscale low-resolution dust emissions to match the variability of high-resolution emissions, such that dust emission simulations tend to be less resolution-dependent. In addition, upscaling the coarse-resolution dust emission simulations can have the advantage of reducing the computational expense while achieving performance similar to that of high-resolution simulations.

To mitigate the scale dependence of dust emission simulations, Leung et al. (2023) proposed to rescale the spatial variability of the modeled dust emissions in ESM native grid resolution by a map of correction factors to account for the spatial variability of higher-resolution dust emissions. We follow the approach in Leung et al. (2023) to yield a map of scaling factors $\widetilde{K}_c$ for CESM2 that corrects the spatial variability of the 0.9°×1.25° emissions $F_{d,c}$ to that of the 0.47°×0.62° emissions $F_{d,f}$. We conduct a

0.47°×0.62° simulation and a 0.9°×1.25° simulation for year 2006 to yield a fine-resolution emission map $F_{d,f}$ and a coarse-resolution emission map $F_{d,c}$. We normalize both emissions to have the same global total emission (following Leung et al., 2023) to focus on the main differences in their spatial variability instead of their magnitude differences. Then, dividing the annual $F_{d,f}$ map by the annual $F_{d,c}$ map for all grid cells results in an annual scaling map $\widetilde{K}_c$ that accounts for the changes in the spatial variability of dust

emissions between high- and low-resolution simulations due to the subgrid variability of all meteorological and land surface variables in the emission scheme:

$$\widetilde{K}_c(\text{long, lat}) = F_{d,f}(\text{long, lat})/F_{d,c}(\text{long, lat}) \tag{12}$$

where (long, lat) indicate the longitude and latitude of each grid cell. $\widetilde{K}_c$ could then be multiplied by $F_{d,c}$ in CLM5 in the native 0.9°×1.25° simulation to adjust the spatial variability of $F_{d,c}$ to $F_{d,f}$ during the

native grid simulation, such that the subsequent dust cycle simulation in CAM6 can yield improved spatial representation of dust variables such as DAOD.

Leung et al. (2023) proposed this method with the annual $\widetilde{K}_c$ scaling instead of seasonal scaling because, while dust emissions exhibit seasonality and interannual variability (e.g., see Fig. S10 in Leung et al., 2023), the mismatch between $F_{d,f}$ and $F_{d,c}$ are largely due to subgrid spatial heterogeneity such as

local topography and soil properties, which are slowly varying variables and partially shared across different model configurations. $\widetilde{K}_c$ in Fig. 1c thus captures the main characteristics of this subgrid variability, even though the ability of $\widetilde{K}_c$ to represent higher-resolution emissions could be improved even further if $\widetilde{K}_c$ was derived specifically for each season, year, or model configuration. In Sect. 5.6 of this paper, we will adjust the spatial distribution of the CESM2 dust emissions for 2004–2008 by multiplying

the 0.9°×1.25° dust emissions by the annual $\widetilde{K}_c$ map from 2006.

The resulting annual $\widetilde{K}_c$ map in Fig. 1c shows the difference in spatial variability between the high- and low-resolution emission simulations. The higher resolution run tends to produce more dust over the semiarid and marginal source regions (red color), producing > 3–5 times more emissions than in the lower resolution run. The reason is that lower-resolution runs employ coarse-resolution winds that smooth out

small-scale wind peaks, and marginal source regions have relatively high emission thresholds such that the spatially averaged wind speed could easily be lower than the emission thresholds, leading to zero emissions for the entire coarse grid. Therefore, low-resolution models will generally underestimate emissions from marginal sources and create emission biases over hyperarid and other prominent source regions. Since high-resolution simulations typically pick up more emissions from marginal sources, the

ratios over major sources (e.g., Sahara) are slightly smaller than 1 (light blue) as compensation to match the same global total emission.

Leung et al. (2023) suggested that modeled dust emissions should be multiplied by the $\widetilde{K}_c$ map to adjust the spatial variability of dust emissions and mitigate coarse model bias due to grid resolution. The degree of how much local-scale dust variability that the scaling map in Fig. 1c can capture is limited by

the spatial resolutions and accuracies of the available input datasets, since some of the input fields (e.g., MERRA-2 meteorological fields) have a native horizontal resolution of ~ 0.5° that represents the highest local-scale variability of dust emissions the $\widetilde{K}_c$ map can capture. The emission increase over marginal sources may be even larger if the scaling factors were calculated using higher resolution inputs such as 0.25°×0.25° or finer (e.g., using ERA5 meteorology).

Finally, we note that the upscaling approach is different from other process-based formulations of saltation processes in Sect. 3.1–3.4, in that Sect. 3.5 is an empirical formulation. The need to employ this scale-aware adjustment will gradually mitigate with increasing ESM horizontal grid resolution, but the importance of the process-based modifications remains regardless of grid resolutions. Since ESMs nowadays at 0.47°×0.62° typically cannot fully resolve smaller-scale meteorological features that drive

dust emission (e.g., mesoscale convections and low-level jets), the $\widetilde{K}_c$ derived from the 0.47°×0.62° $F_{d,f}$ will only remedy the scale dependence issue due to the smoothed meteorological inputs in coarser models,

but will not represent emissions induced by those finer-scale meteorological features. As ESMs resolve the small-scale meteorology better in the future, $F_{d,f}$ and $\widetilde{K}_c$ will become more capable of capturing emissions generated by the small-scale meteorology.

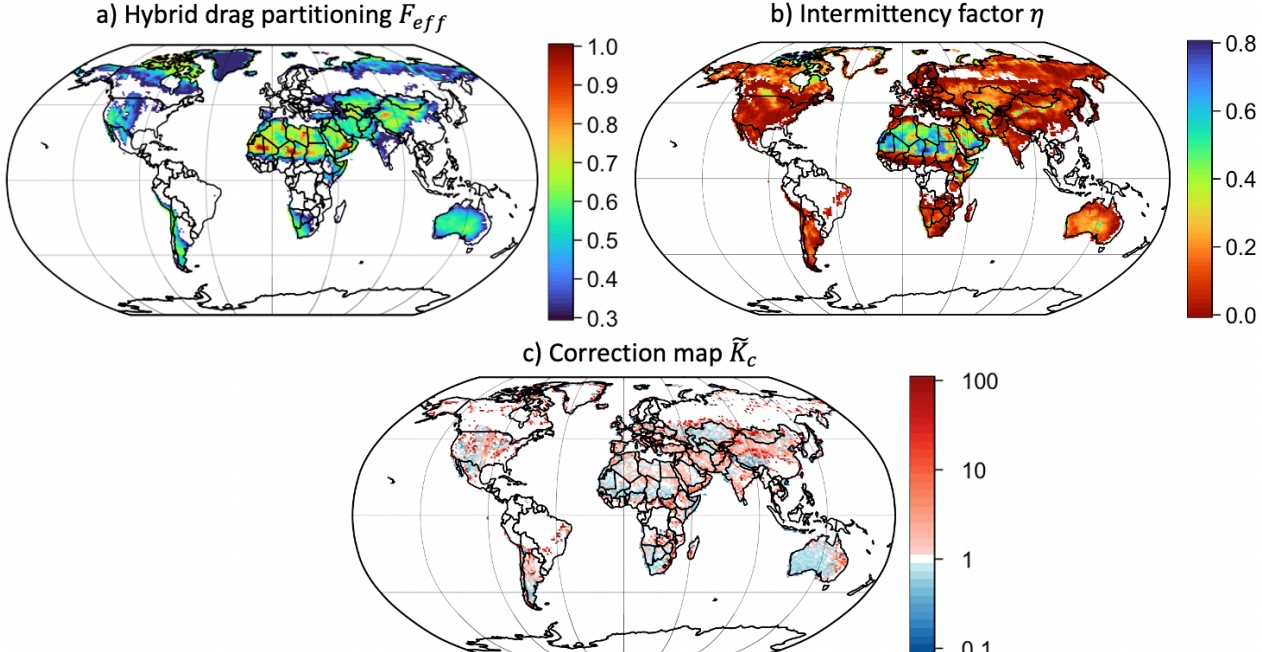

Figure 1. Implementations of proposed modifications in Leung et al. (2023) into the Community Land Model version 5 (CLM5). (a) Simulated hybrid drag partition effect $F_{eff}$ on wind friction velocity considering the land surface roughness due to rocks and green vegetation. (b) The fraction of time that dust emission is active (intermittency factor $\eta$), averaging across all time steps when the emission flux $F_d > 0$. Note that the colorbar in (b) is inverted compared to (a) to show the contrasts in $\eta$ between hyperarid and nonarid regions more clearly. (c) Correction map $\widetilde{K}_c$ for 0.9°×1.25° dust emission from the standalone dust emission model obtained in Leung et al. (2023).

## 4 observational and reanalysis datasets for evaluating the dust emission schemes

To evaluate the CESM dust cycle simulations using different dust schemes, we employ multiple observational and reanalysis datasets of atmospheric dust over various spatial scales for comparisons. This section briefly summarizes the independent datasets that we use to evaluate the CESM dust simulations.

### 4.1 Ridley et al. (2016) regional mean DAOD

We first employ a regional mean dust optical depth (DOD) dataset, constrained by Ridley et al. (2016) and compiled by Adebiyi et al. (2020) and Kok et al., (2021) (see Fig. 4 below). It is an observational–modeling constraint dataset on regional mean DAOD at 550 nm. Ridley et al. (2016) used various satellite AOD retrievals including the Multiangle Imaging Spectroradiometer (MISR) as well as the Moderate Resolution Imaging Spectroradiometer (MODIS), all bias-corrected by the more accurate ground-based AOD measurements by the Aerosol Robotic Network (AERONET). They then obtained the fraction of AOD due to dust using an ensemble of state-of-the-art global and region models, and combined the retrieved satellite AOD and the modeled dust fraction to total AOD to yield the DAOD. To reduce data uncertainties, Ridley et al. (2016) only chose 15 major dusty regions where dust contributed to a significant portion of total AOD (see Fig. S5 for the defined dusty regions) and obtained the regional mean

DAOD instead of yielding grid-by-grid DAOD. Additionally, averaging across space and time (2004–2008) enables error quantification of the regional mean DAOD. Nonetheless, Ridley's DAOD values over the Southern Hemisphere (SH) are subject to more biases than those over the Northern Hemisphere (NH), mainly because the dust fraction contributing to total AOD is much smaller over the SH. Following Kok et al., (2021), we thus instead use the regional mean DAOD values estimated by Adebiyi et al. (2020), which is based on reanalysis products, with smaller uncertainties over the three SH sources. Also, the regional mean DAOD over North America was obtained from Adebiyi et al. (2020). This dataset represents seasonal DAOD values averaged over 2004–2008; the Ridley and Adebiyi regional DAOD values are listed in Table 2 of Kok et al. (2021). We will compare this dataset to our gridded DAOD simulations, averaged across years and grids to regional mean, for evaluation purposes.

## 4.2 AERONET and AERONET–SDA AOD

Additional observational dust properties are provided by the Aerosol Robotic Network (AERONET; e.g., Holben et al., 1998; Dubovik and King, 2000; Dubovik et al., 2000). For AOD, we consider the AOD v3 Direct Sun Algorithm level 2 (prefield and postfield calibrated, cloud-cleared, and manually inspected) data. We selected 39 stations following Kok et al. (2014b) and Albani et al. (2014), based on the filtering criterion that only the dust-dominant AERONET sites are picked (see Fig. S11 in Kok et al., 2014b for all selected sites). These "dusty" stations are mostly located over the Sahara (as seen in Fig. 5). We further employ the AERONET coarse mode AOD data as retrieved by the spectral deconvolution algorithm (SDA; O'Neill et al., 2003), which was also used by other studies to represent DAOD (O'Neill et al., 2003; Capelle et al., 2018). Following Capelle et al. (2018), for some stations that do not contain level 2 data (not quality controlled and/or cloud-cleared), we use level 1.5 data for those sites instead. AERONET takes multiple measurements within an hour during the daytime, with sub-hourly data available on the AERONET website. The website also compiles daily mean AOD data for the stations. We thus take the daily mean AERONET–SDA values, which are helpful for examining the spatiotemporal variability of the model simulations. The 2004–2008 mean AERONET–SDA coarse mode AOD values are shown in the upper panels of Fig. 5 as overlaid points and more clearly in Fig. S6. The locations of the sites could be found in Table S1.

## 4.3 MIDAS DAOD

In addition to the ground-based AOD observations, we also employ a globally gridded reanalysis DAOD product provided by Gkikas et al. (2021), namely the MODIS Dust Aerosol (MIDAS) dataset. MIDAS combines quality-filtered MODIS/Aqua AOD collection 6.1 level 2 at 550 nm with DAOD-to-AOD ratios from MERRA-2 reanalysis to yield DAOD on the MODIS native grid. The resulting dataset has a fine spatial resolution of $0.1° × 0.1°$ and contains daily DAOD and AOD over 2003–2017. The uncertainties of the Aqua AOD and MERRA-2 dust fraction are incorporated into the final MIDAS DAOD uncertainty. MIDAS DAOD highly complements AERONET AOD by providing global coverage of DAOD, with gridded AOD in high agreement with AERONET AOD (Fig. 3 of Gkikas et al., 2021). Another advantage of this dataset is that Gkikas et al. (2021) analyzed both land and ocean AOD, and thus MIDAS also provides DAOD over ocean surfaces. We will use the MIDAS dataset to examine the day-to-day variability of our gridded DAOD simulations. To match the horizontal resolution of CESM2, we regridded MIDAS DAOD from $0.1° × 0.1°$ to $0.9° × 1.25°$ (see Fig. 3d).

## 4.4 In-situ PM concentration and deposition flux measurements

We also use site measurements of dust PM (e.g., Prospero and Nees, 1986; Prospero and Savoie, 1989) and dust deposition flux (e.g., Ginoux et al., 2001; Tegen et al., 2002; Lawrence and Neff, 2009; Mahowald et al., 2009; Albani et al., 2014) as climatological datasets for evaluating the spatial variability of dust PM and deposition flux simulations (see data availability section). Previous studies compiled dust PM measurements using high-volume filter collectors at the University of Miami Ocean Aerosol Network

as well as station data that were previously compiled on annual averages (Mahowald et al., 2009; Zuidema et al., 2019). The dust deposition flux climatology used here was compiled by Albani et al. (2014) and used in later studies (e.g., Li et al., 2022). Since CESM2 only simulated dust < 10 $\mu$m, Li et al. (2022) processed the data to estimate concentration and deposition only below the size cutoff using the reported parameters. The upper panels of Figs. 8–9 show the site dust $PM_{10}$ (dust particulate matter of diameter > 10 $\mu$m) concentrations ($\mu$g m$^{-3}$) and dust deposition fluxes (kg m$^{-2}$ yr$^{-1}$) as overlaid points.

## 5. Model evaluation

In this section, we evaluate the performance of the different dust emission schemes in CESM2 – Z03, K14, and our scheme – by comparing the spatial and temporal variability of the modeled dust against observations and reanalysis datasets. We first evaluate in Sect. 5.1–5.4 the use of our process-based dust emission scheme (in Sect. 3.1–3.4) without the use of the empirical upscaling method. Sect. 5.5 then briefly examines a sensitivity test of separating the effects of drag partition and intermittency on the resulting dust cycle simulations. Then, we also evaluate in Sect. 5.6 the effects of additionally using the empirical scaling map $\widetilde{K}_c$ (Sect. 3.5) to rescale our scheme's emissions on the resulting CESM2 atmospheric dust simulation, in order to clearly separate the effects of the process-based modification and the scale-aware adjustment.

We note that global dust simulations typically employ a global tuning factor that scales the global dust emission to a reasonable level that matches observations, since thus far there is no known *a priori* physical principles that govern the order of magnitude of global total dust emission in the dust emission schemes. Past studies (e.g., Klose et al., 2021; Li et al., 2022) scaled the global dust emissions to produce a global mean modeled DAOD of 0.03±0.01 (95 % confidence interval), which is a global constraint given by Ridley et al. (2016). In this section, we thus also scale our dust simulations with a global tuning factor in the CAM6 namelist variable (dust_emis_fact) like past studies (e.g., Li et al., 2022) did. Here we scaled the simulations with K14 and our new scheme such that their simulated global mean DAOD in CESM2 is 0.03. We did not need to scale the Z03 simulation since the default CESM2–Z03 dust simulation already yielded a global mean DAOD of 0.03 during the CESM2 benchmarking.

### 5.1 CESM2 dust emissions using different emission schemes

Figure 2 shows the dust emissions (for dust $PM_{10}$) that arise from Z03, K14, and the Leung et al. (2023) scheme for 2004–2008. The emission maps are normalized such that the global mean DAOD is 0.03±0.01 following ridley et al. (2016). The global sum of emission fluxes for each scheme are indicated at the bottom of the panels in Tg yr$^{-1}$. They have different magnitudes because dust emissions originated from different geographical locations can be subject to different deposition rates (e.g., tropical dust particles experience stronger wet scavenging). Note that the global total emissions in other ESMs could be larger than those from our runs if they account for dust particles > 10 $\mu$m. Even if they scale their emissions to yield global DAOD of 0.03, they will yield larger global emissions than ours mainly because coarse dust particles have smaller optical thickness than fine dust (Adebiyi et al., 2023).

The spatial variability of the emissions for Z03 (Fig. 2a) is controlled by the geomorphic source function *S* developed by Zender et al. (2003b). *S* was a continuous function when formulated by Zender, but in CESM2 the source function is truncated for all values of S smaller than 0.1 (also see Fig. 2 in Li et al., 2022), resulting in a rather spatially discrete and disjointed pattern of emissions. The Z03 scheme captures some major and marginal dust sources, such as the Bodélé Depression in Chad, El Djouf in Mali and Mauritania, the Namib Desert in Namibia, the Nubian Desert in Sudan and Egypt, the Taklamakan Desert in China, Patagonia in Argentina, the Karakum/Kyzylkum Deserts in central Asia, and the Strzelecki Desert in Australia. It does not fully capture some other major and secondary sources, such as

the Rub' al Khali Desert over Saudi Arabia and deserts in the U.S. Several other regions like the Nubian Desert in Sudan/Egypt appear as prominent sources, which is not supported by satellite retrievals (Fig. 3d).

K14 emissions (Fig. 2b) show a much more continuous spatial pattern. K14 successfully captures emissions not only over major sources such as the Sahara and the Arabian Peninsula, but also emissions
over semiarid regions and secondary sources such as the United States and central Asian deserts. Without the constraint of soil erodibility $S$ in Z03, K14 produces much higher emissions over Australia because of the low moisture effect, and over the Horn of Africa (HoA) because of its very high $u_*$ compared to other hyperarid regions (see Fig. S4) especially during boreal summertime. In the CESM2 simulation of K14, some major sources like the Taklamakan Desert have comparable or smaller emissions than some semiarid
regions such as the deserts in Australia, which could be a result of bias of input meteorological fields or not including enough aeolian physics in the K14 parameterization.

Our scheme (Fig. 2c) adds extra aeolian physics on top of K14. While using $D_p$ has little effect on the spatial variability of the dust emission thresholds and the emission fluxes, the drag partition effect $F_{eff}$ modifies $u_{*s}$ and highlights the major sources over the Bodélé Depression, El Djouf, and the Rub' al
Khali Desert. $F_{eff}$ suppresses emissions from most semiarid regions with higher surface roughness. The intermittency effect increases emissions from remote regions such as the northern U.S., northern Canada, and Siberia, and possibly overemphasizes emissions over the Tibetan Plateau.

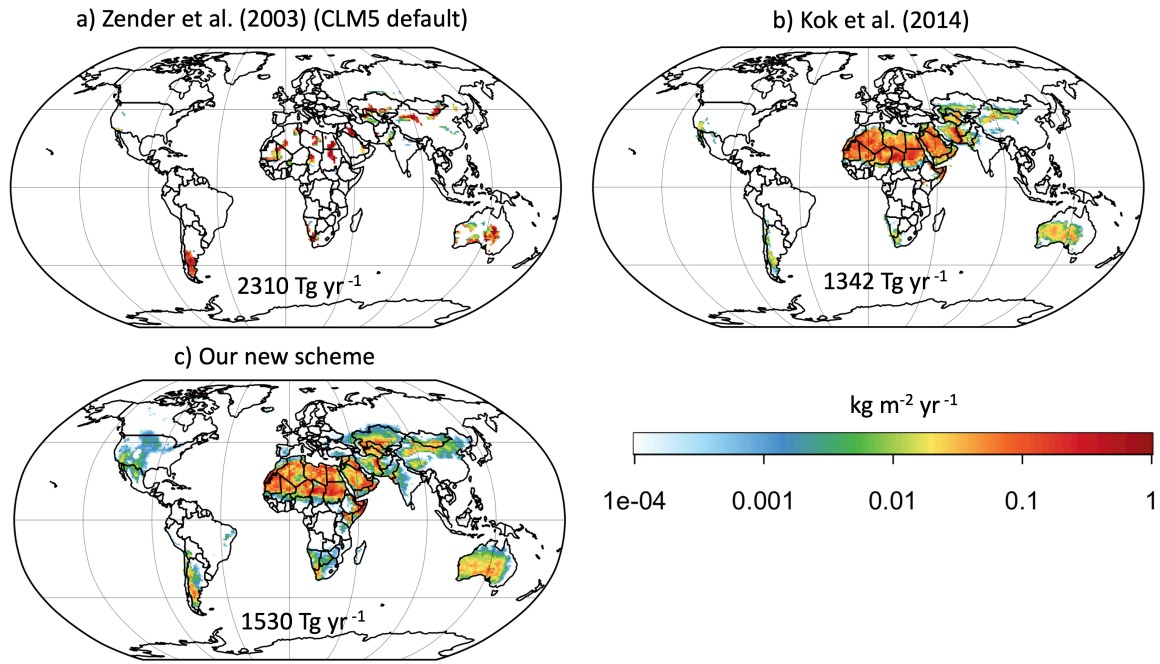

Figure 2. CESM/CLM5 dust emissions averaged across 2004–2008, for (a) Z03, (b) K14, and (c) our new scheme (Leung et al., 2023). All emissions are normalized such that the corresponding CAM6 dust aerosol optical depth (DAOD) is 0.03±0.01 (following the global DAOD constraint obtained by Ridley et al., 2016). The global total emissions (in Tg yr$^{-1}$) to yield a global mean DAOD of 0.03 are indicated in each panel.

**5.2 DAOD spatial variability**

Here we compare the spatial distributions of DAOD maps simulated by CAM6 using Z03 (Fig.
3a), K14 (Fig. 3b), and our scheme (Fig. 3c), as well as derived from MIDAS (Fig. 3d), averaged across

2004–2008. Figure 3d shows the MIDAS DAOD, with its peak over the Bodélé Depression of the Sahel of ~ 0.6. DAOD is also moderately high over El Djouf and the southwestern Sahara (~0.3–0.4). The annual MIDAS DAOD has several local peaks of > 0.3 (yellow color) over the Arabian Desert, the Thar Desert in India, and the Taklamakan Desert over northwestern China. There are some modest DAOD levels over central China, e.g., < 0.2 over the Sichuan Basin and the North China Plain (NCP), which are metropolitan regions of high anthropogenic aerosol pollution (e.g., Leung et al., 2018). This indicates that MIDAS might occasionally not be able to truncate all anthropogenic aerosol signals from the MODIS/Aqua AOD data product.

Figure 3a shows the Z03 DAOD simulated by CAM6. The spatial pattern of Z03 DAOD in CAM6 is largely shaped by the Z03 source function (soil erodibility map $S$). It has multiple high DAOD regions (> 1), including the Bodélé Depression, the Nubian Desert over Sudan, the eastern Arabian Peninsula, the Taklamakan Desert, the Strzelecki Desert in Australia, and some small peaks over southern Africa and South America. The DAOD values over these regions are all scaled up by the source function $S$ and are unreasonably high compared with the MIDAS DAOD. The source function also generates DAOD peaks that are absent in observations, e.g., the Nubian Desert in Sudan.

Figure 3b shows the K14 DAOD simulated by CAM6. Without the source function, K14 has reduced DAOD over many source regions. K14 calculates the time-varying soil erodibility $C_d$, which indicates the most erodible region to be the Bodélé Depression, El Djouf, and the southern Sahara, resulting in the high DAOD (~ 0.6–0.7) over the south of Sahara. The western Sahel has a larger area of high DAOD (~0.6) over Mali/Niger, which is different from MIDAS that indicates a higher DAOD peak over the Bodélé Depression than El Djouf. Due to the equatorial easterlies, dust advection toward the west leads to a DAOD 0.4–0.5 over a significant part of the tropical Atlantic Ocean. DAOD is also ~ 0.3–0.4 over most of the Arabian Peninsula. Over Australia, the western region becomes the most erodible region because of low simulated soil moisture, which is not in agreement with observations which indicate the Strzelecki Desert (central Australia) has the highest DAOD across Australia (annual mean ~ 0.084).

Our new scheme's DAOD (Fig. 3c) shares a similar spatial variability with K14 DAOD. The main difference between K14 and our scheme's DAOD is the relatively lower DAOD levels over the Mali/Niger region where El Djouf is located because the drag partition effect reduces emissions over most of the Mali/Niger region (Fig. 3c). Figure S7 shows the difference between our scheme's DAOD and K14 DAOD. Comparing against MIDAS DAOD, our scheme and K14 overestimate dust over Australia and the HoA, which is possibly due to the biases in the meteorological variables (e.g., $u_*$ and $w$) of CESM2. K14 and our scheme both overestimate DAOD over Sudan compared with the MIDAS DAOD because the dust emission equation is very sensitive to the low CESM soil moisture there, but our DAOD's high bias is smaller than K14 DAOD's. Both K14 and our scheme underestimate DAOD levels over the Taklamakan/Thar Deserts, which is also seen in other studies employing K14, e.g., Li et al. (2022) and Klose et al. (2021). None of the scheme captures the DAOD levels over the Thar Desert as shown by MIDAS. The overall improvements of our scheme's DAOD is that it better captures the DAOD values over El Djouf and reduces the DAOD overestimations over the Arabian Peninsula and Sudan. Our scheme also has higher DAOD levels over semiarid regions.

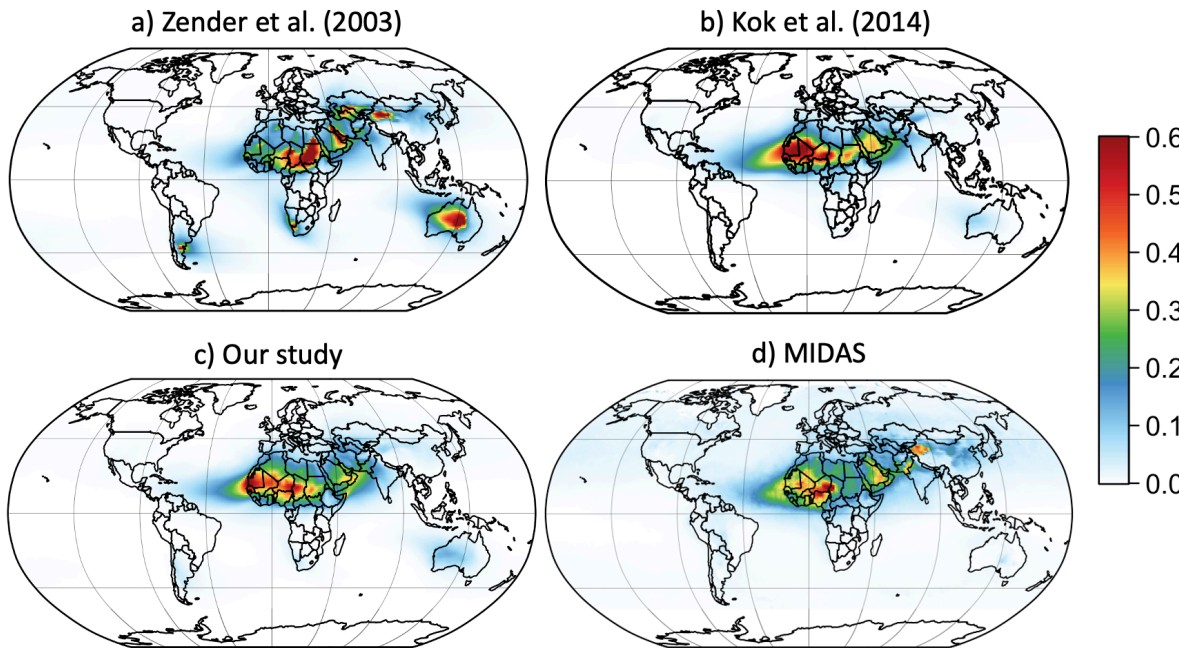

Figure 3. Global DAOD averaged across 2004–2008 from CESM2 and MIDAS. (a-c) CESM DAOD for (a) Z03, (b) K14, and (c) our new scheme (Leung et al., 2023). (d) MIDAS DAOD (Gkikas et al., 2021). All maps have a global mean of 0.03±0.01 consistent with the global DAOD constraint obtained by Ridley et al., 2016).

Next, we compare Ridley et al. (2016) regional mean DAOD with CAM6 simulated DAOD using Z03, K14, and our scheme in Fig. 4. Simulated regional DAOD are regionally averaged following the definition of 15 dusty regions (see Fig. S5) in Kok et al. (2021). The upper and lower panels show the annual and seasonal mean regional DAOD, respectively. Our scheme's DAOD (Fig. 4c) shows the highest correlations with Ridley's DAOD (annual $R^2 = 0.82$; seasonal $R^2 = 0.76$), matching the regional DAOD distribution the best, whereas Z03 DAOD (Fig. 4a) produces the lowest correlations (annual $R^2 = 0.44$; seasonal $R^2 = 0.42$) and the highest root-mean-square error (RMSE). Z03 overestimates annual DAOD over Bodélé/Sudan and Australia but underestimates DAOD over Mali/Niger and western Africa, which are primarily controlled by the strength of the source function $S$. K14 (Fig. 4b) shares a similar performance with our scheme matching against Ridley's regional DAOD values (annual $R^2 = 0.77$; seasonal $R^2 = 0.67$), but K14 overestimates the high regional DAOD values (e.g., Mali/Niger and Bodélé/Sudan). K14 also tends to overestimate wintertime and springtime dust over the tropical Atlantic and western Africa. Both K14 and our scheme underestimate DAOD levels over the Taklamakan/Gobi Deserts and the Thar Desert (Figs. 4b and c), mostly due to underestimations of dust in the springtime (MAM; green color). Finally, MIDAS DAOD (Fig. 4d) has the highest consistency with Ridley's annual and seasonal mean DAOD (annual $R^2 = 0.96$; seasonal $R^2 = 0.95$).

Our new scheme has the reduced major axis (RMA) regression slopes the closest to the 1:1 line (annual slope = 0.92, seasonal slope = 0.82), demonstrating the smallest fitting bias among the three schemes. K14 DAOD has larger regional DAOD over Mali/Niger and El Djouf (Fig. 3b) and the RMA regression slopes moderately smaller than 1 (annual slope = 0.72, seasonal slope = 0.67). Z03 in CESM2 is pre-tuned, but also overestimates dust over major source regions (Fig. 3a) and the RMA slopes also deviate from 1 (annual slope = 0.81, seasonal slope = 0.78).

All simulations, regardless of the dust emission scheme employed, show systematic underestimations for lower regional DAOD values and overestimations for higher regional DAOD values, consistent with the findings of Zhao et al. (2022). The reasons for the underestimations of lower regional DAOD values could be because the schemes (mainly Z03 and K14) are underestimating dust emissions

from marginal source regions (with lower regional DAOD values), which is partially corrected in our scheme by producing more emissions from semiarid regions. It could be further because ESMs overestimate wet depositions of dust over tropical oceans (Albani et al., 2014; van der Does et al., 2020), for possible reasons including an overestimated light rain frequency (Wang et al., 2021) and a higher hygroscopicity due to internal mixing with other aerosols (Neale et al., 2012). ESMs also overestimate dry depositions for reasons that remain unclear but could include turbulence in dusty layers and an underestimation of the extent to which particle asphericity enhances drag (Weinzierl et al., 2017; Huang et al., 2020; Meng et al., 2022; Drakaki et al., 2022). These factors all contribute to a shorter lifetime of dust, enhancing the dust concentration contrasts between sources and downwind / far-field regions.

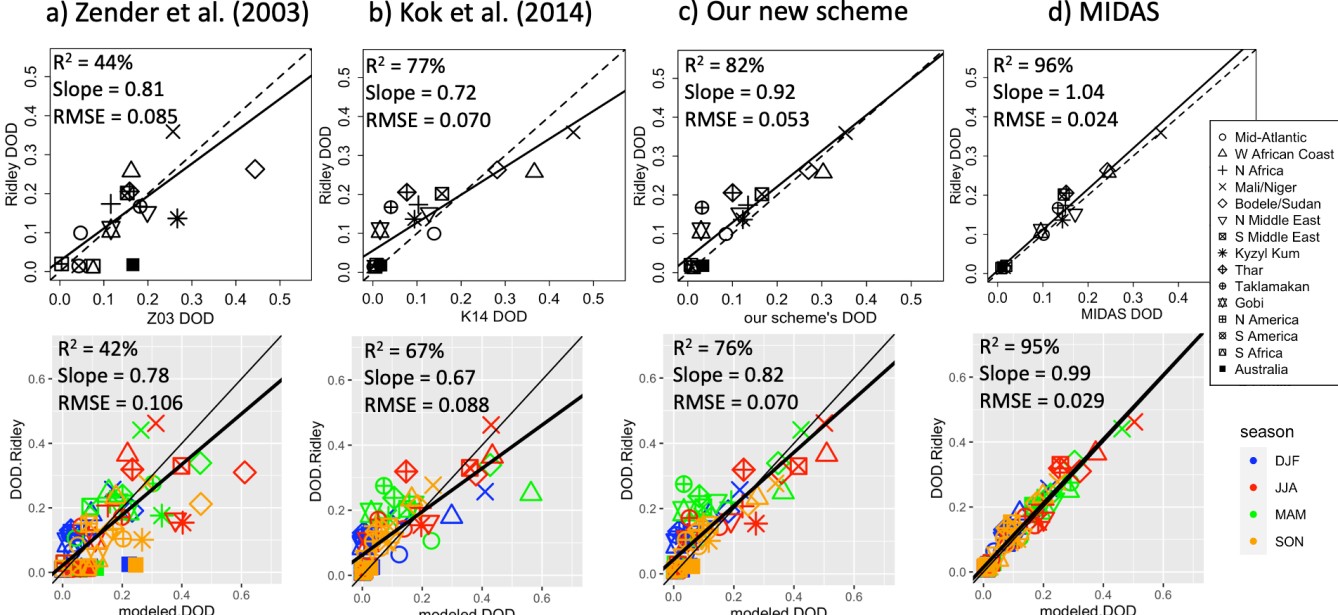

Figure 4. Ridley et al. (2016) regional mean DAOD vs. modeled CESM2 DAOD using (a) Z03, (b) K14, and (c) our scheme, as well as (d) MIDAS DAOD (Gkikas et al., 2021), for 2004–2008 over 15 dusty regions (see Fig. S5) defined following Kok et al. (2021). Top panels show annual mean DAOD scatterplots, with dashed lines as 1:1 lines and solid lines as the reduced major axis (RMA) regression lines. Bottom panels show seasonal mean DAOD scatterplots for the three schemes, with thin lines as 1:1 lines and thick lines as the RMA regression lines. Seasons are defined as: DJF (December–January–February), MAM (March–April–May), JJA (June–July–August), and SON (September–October–November).

Next, we evaluate the simulated spatial DAOD variability against coarse mode AOD observations at multiple AERONET stations. Figure 5 compares the satellite-derived MIDAS DAOD and the CESM2 simulations against the AERONET–SDA coarse mode AOD. The 39 site locations we chose (Sect. 4.2) are over arid regions such that the coarse mode aerosols are mostly dust. MIDAS (Figs. 5d and h) gives the best agreement when compared against AERONET, yielding the largest coefficient of determination ($R^2$) of 0.76 and the smallest RMSE of 0.065. RMA regression gives a slope of 1.11 (blue line), which is close to the 1:1 line (black line).

Evaluating the dust emission schemes using the AERONET AOD measurements gives the similar conclusion as using the Ridley DAOD values. The Z03 scheme (Fig. 5a and e) shows the lowest degree of agreement against AERONET with an RMSE of 0.21, more than three times the RMSE of MIDAS DAOD. Z03 substantially overestimates DAOD over Australia (Fig. 5a) because of the large source function $S$ there (AOD values are < 0.1 for the Australian AERONET sites). There are also multiple

820 underestimations of Z03 DAOD of ~0.3 over the Sahel, which can be > 0.5 for AERONET sites (Fig. 5a). Note that although Z03 has a relatively decent regional RMA regression slope in Fig. 4a, Z03 shows much stronger bias against AERONET AOD with an RMA slope of 0.66 because it strongly overestimates DAOD over hyperarid regions. Meanwhile, K14 (Figs. 5b and f) yields a much higher spatial $R^2$ of 0.70 and a much smaller RMSE of 0.080 against AERONET data. K14 has fewer DAOD underestimations
825 over the Sahara–Sahel region and reduced DAOD overestimations in the Arabian Peninsula and Australia (Fig. 5f). The RMA regression slope of 0.85 shows that K14 simulates the spatial variability of AERONET AOD relatively well compared to Z03, different from Fig. 4b which shows that K14 regionally has stronger seasonal DAOD bias than Z03. This suggests that evaluating dust schemes against regional and local station data can yield different conclusions regarding biases. Our new scheme (Figs. 5c and g)
830 reduces the bias generated by K14, yielding an RMA regression slope of 1.02. Our scheme's DAOD yields an $R^2$ of 0.73 and an RMSE of 0.072, marking modest improvements over K14 simulations. Overall, our scheme performs the best among three schemes in capturing the spatial AOD variability of AERONET sites.

835

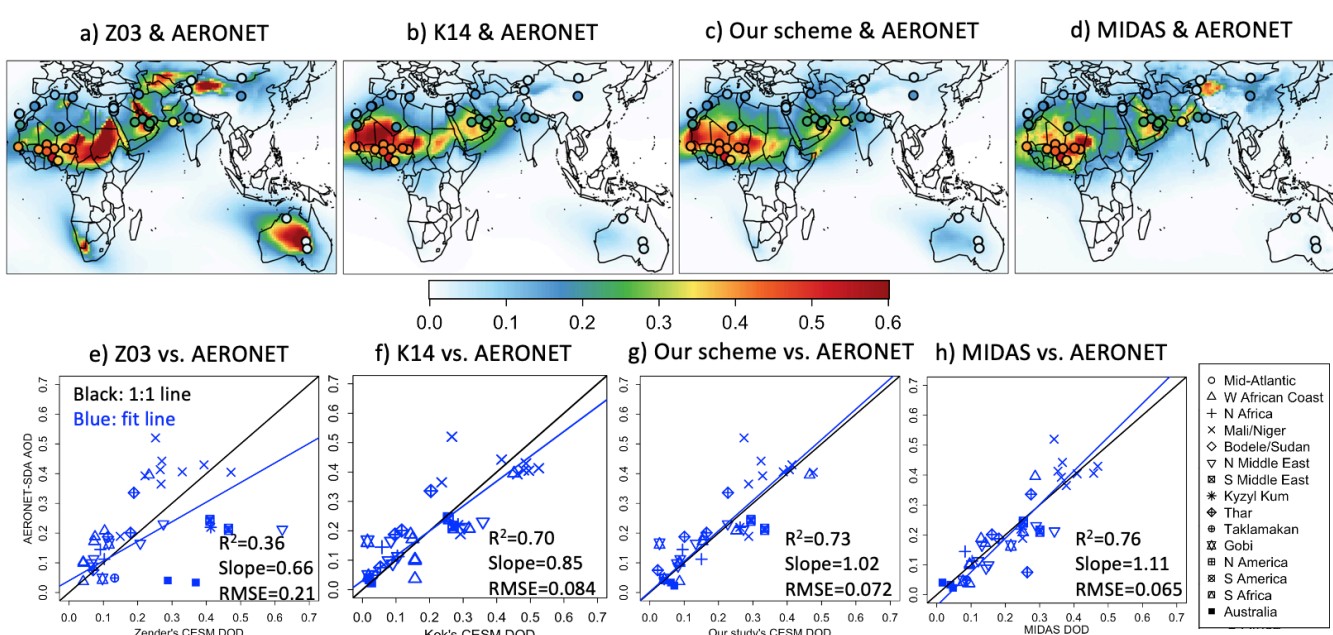

Figure 5. Gridded model/satellite DAOD vs AERONET–SDA coarse mode AOD for 2004–2008. Top panels show the global dust AOD for (a) MIDAS (b) Z03, (c) K14, and (d) our scheme, overlaid by AERONET sites of coarse mode AOD observations. (e–h) The respective scatterplots for AERONET
840 AOD versus (e) MIDAS DAOD, as well as CESM DAOD using (f) Z03, (g) K14, and (h) our scheme. The 15 source regions (labelled with symbols) follow the definition of Fig. S5 adopted from Ridley et al. (2016) and Kok et al. (2021).

845 **5.3 DAOD day-to-day variability**

  Apart from examining the spatial variability, we also examine the temporal variability of CESM2 dust using different dust emission schemes. Here we use globally gridded daily MIDAS DAOD across 2004–2008 and multiple stations of AERONET–SDA coarse mode daily AOD for evaluations. For
850 MIDAS DAOD, we calculate grid-by-grid daily Pearson correlations between MIDAS and CESM2 DAOD, yielding a global correlation map for each scheme (Fig. 6). We note that since MIDAS is a reanalysis dataset, it itself is also subject to errors due to the MERRA-2 assimilation errors and MODIS

instrumental and algorithmic errors. Gkikas et al. (2021) reported that the day-to-day variability of MIDAS DAOD is highly consistent over the Dust Belt (e.g., their Fig. 2d) when compared against the CALIOP satellite retrievals of dust (LIVAS; Amiridis et al., 2013; Marinou et al., 2017). While both MIDAS and CALIOP have uncertainties, we view the day-to-day variability of MIDAS dust as most accurate over the Dust Belt and thus focus our CESM–MIDAS comparison in Fig. 6 over the Dust Belt. In Fig. 6, we show the correlation results over gridboxes with MIDAS annual mean DAOD (Fig. 3d) larger than its annual mean DAOD uncertainty (Fig. 8b in Gkikas et al., 2021), which largely corresponds to the gridboxes over the Dust Belt (as shown in Fig. S9b). Gridboxes with MIDAS mean DAOD smaller than the mean DAOD uncertainty are masked out in Fig. 6, and Fig. S8 shows the correlation maps without any masking. Figure S9 shows the MIDAS global DAOD/AOD fraction for 2004–2008 (Fig. S9a) and the ratio of MIDAS mean DAOD to the mean DAOD uncertainty (Fig. S9b). We also further discuss the daily correlations of CESM modeled dust with its driving meteorological and land surface variables at the end of this subsection (see also Figs. S10 and S11).

We first examine the correlations between MIDAS DAOD and our CESM simulations of DAOD. In Fig. 6a, Z03 dust shows overall strong daily correlations with MIDAS dust over the Dust Belt and the tropical Atlantic. The correlations are generally lower over the eastern than the western Sahara, likely partially due to the strong extra dust sources represented by Z03 over Sudan/Egypt, which is absent in MIDAS DAOD. The predominant easterly trade winds bring dust signals from Sudan to the central and western Sahara, likely reducing correlations over dust sources such as the Bodélé Depression. Another possible reason is because the daily correlations between Z03 dust and the driving meteorological fields over the eastern Sahara are generally modest, with $R$ of only $\sim 0.1$–$0.2$ (see Figs. S10a-c). Another region of strong correlations occurs over the Arabian Sea, indicated in Fig. 6a as dominated by dust from the HoA, meaning both MIDAS and Z03 agree that dust advects from the HoA to central Asia and regulates dust air quality in downwind regions. Z03 also shows high correlations with MIDAS over the Thar Desert and moderately high correlations over China, especially over the Taklamakan Desert. This partially indicates that although the regional emission strengths of Z03 are likely overestimated as shown in the previous subsection, the Z03 source mask indeed helps emphasize the true source origins of dust, which subsequently benefit a more accurate temporal dust variability over the Taklamakan and its downwind regions.

K14 dust in Fig. 6b generally shows weaker correlations with MIDAS dust over the Dust Belt than the other two schemes. K14 has smaller correlations with MIDAS than Z03 (negative $\Delta R$ values in Fig. 6d), despite the fact that K14 emission has stronger daily correlations with the driving fields than Z03 dust (Figs. S10d-f). One possible reason is that K14 emissions over most of the Sahara are similarly strong (Fig. 2b), meaning K14 is less capable of distinguishing primary emission sources from secondary sources. As a result, simulated dust signals over downwind regions (western Africa and the Atlantic) could be contaminated by dust signals from secondary sources such as Sudan, Western Sahara, and western Mauritania. The same issue likely occurs over the eastern Sahara since the Arabian Peninsula (upwind of the eastern Sahara) emits similar orders of magnitude of dust across most of the Peninsula instead of coming primarily from the Rub' al Khali Desert. Correlations over the Taklamakan Desert also appear weaker than in Z03 (Fig. 6d), possibly because of the higher-than-observed dust emissions from the Karakum/Kyzylkum region in central Asia advected by the predominant westerlies that contaminate dust signals over the Taklamakan.

Our scheme in Fig. 6c captures similar correlations as Z03 overall, with higher correlations ($R \sim 0.7$–$0.8$) over western Africa, the Atlantic, the Arabian Sea, and India. Our scheme performs modestly better than Z03 over the northern Sahara (the Algerian Desert and the Libyan Desert) as well as the Sahel and the Gulf of Guinea (positive $\Delta R$ values in Fig. 6e), which is likely a result of dust coming from more correct source regions. Modestly better performance is also seen over the Rub' al Khali Desert likely due to the wind drag partition corrections. Additionally, our scheme's dust emission correlates better with

meteorological drivers than K14 and Z03 (Fig. S10g-i), especially with $u_{*s}$, which likely also helps improve the DAOD correlations with MIDAS DAOD. Meanwhile, a more significant reduction in correlations occurs over China when comparing K14 and our scheme with Z03 (Fig. 6e). Our scheme might produce weaker correlations than Z03's because our scheme with drag partitioning causes $u_{*s}$ to exceed the emission thresholds less often, resulting in both weakened annual mean DAOD and weakened seasonality of the DAOD time series. Apart from northwestern China, there are some additional moderate correlation differences over central China (negative $|\Delta R|$ values in Fig. 6e), which are metropolitan regions with vast anthropogenic aerosols (e.g., Leung et al., 2018). This again indicates that, as discussed in Fig. 3d, MIDAS DAOD might still contain some anthropogenic aerosol signals in urban regions.

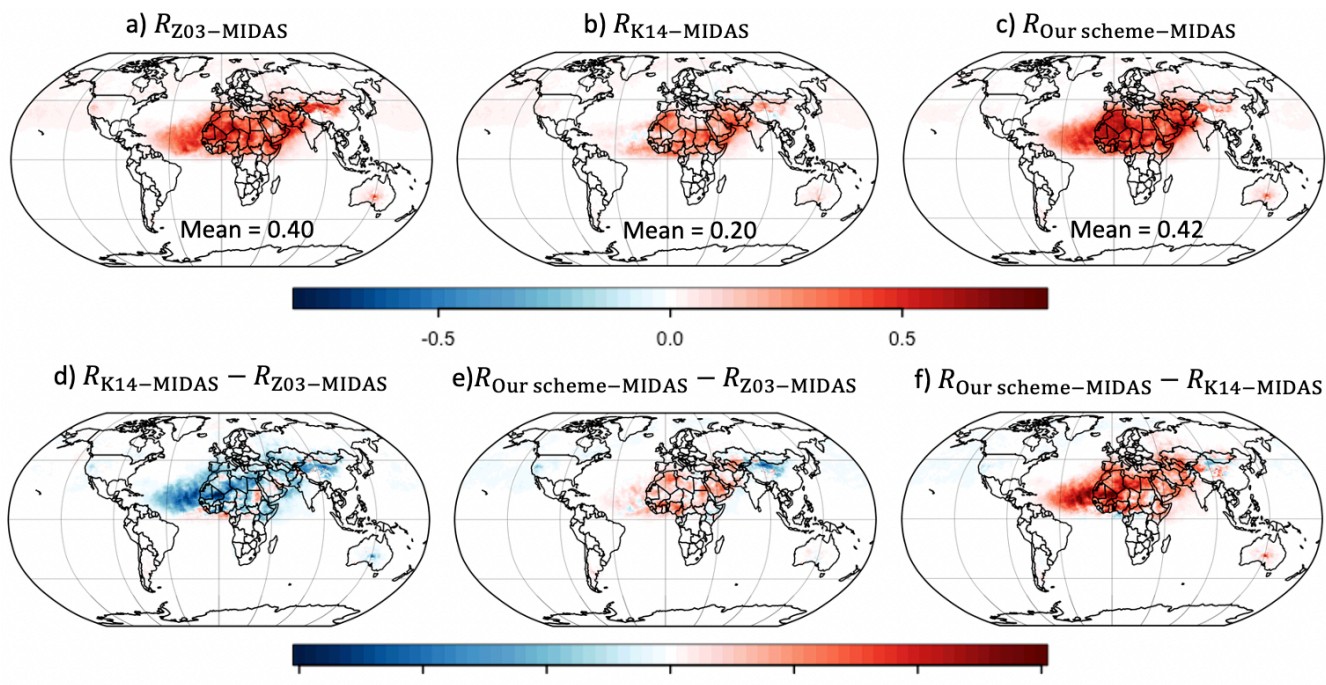

Figure 6. Grid-by-grid MIDAS DAOD daily Pearson correlation maps with CESM2 DAOD for 2004–2008. (a-c) Correlation maps $R$ of MIDAS daily DAOD time series vs. CESM2 daily DAOD time series using (a) Z03, (b) K14, and (c) our scheme. The correlation maps focus on gridboxes with MIDAS DAOD/AOD ratio > 0.25 only. Pixels with MIDAS annual DAOD uncertainty (defined by Gkikas et al., 2021) larger than annual mean DAOD (see Fig. 9) are filtered out (Fig. S8 shows the unfiltered correlation maps). The values at the bottom of the panels show the global mean correlation values (for all gridboxes with MIDAS DAOD/AOD > 0.25). (d-f) Changes ($\Delta R$) in correlation maps from (d) Z03 to K14, (e) Z03 to our scheme, and from (f) K14 to our scheme.

For AERONET data, we calculate daily Pearson correlations between the selected AERONET stations and the CESM2 grids that contain those stations. The conclusions are similar to the ones discussed in Fig. 6. For Z03 (Fig. 7a), strong correlations are generally seen over the Sahara and central Asia because of a relatively decent representation of the locations of dust sources. Z03 has a generally weaker representation of the temporal dust variability over Australia, as in K14 and our scheme. For K14 (Fig. 7b), the correlations over the Sahara tend to be weaker ($R$ around 0–0.4), likely due to the inadequate representation of dust source locations. The sites over northern and eastern Australia yield smaller correlations in K14 than Z03, because the modeled dust signals over there are contaminated by emissions from the west, which has higher emissions than the east. Our scheme yields overall the highest correlations

across the globe out of all schemes. Our scheme's dust highly correlates with MIDAS dust over the Sahara and the Middle East ($R \sim 0.5$–$0.8$). Correlations over the Australian sites in our scheme are also the highest among all schemes ($R \sim 0.3$–$0.5$), even though our scheme generates similar orders of magnitude of emissions across different parts of Australia (in Fig. 2c). One issue is the correlation over a Mongolian site to the north of the Gobi is about zero (the weakened correlation also occurs in the K14 simulation in Fig. 7b). As discussed in the previous paragraph, this is likely a result of our scheme's inability to generate high dust emissions from the Taklamakan than the Gobi, such that the DAOD signal over the Mongolian site is contaminated by the dust from other sources. Meanwhile, Z03 with high Taklamakan emissions and low Gobi emissions yields a high $R$ of $\sim 0.6$ over the Mongolian site.

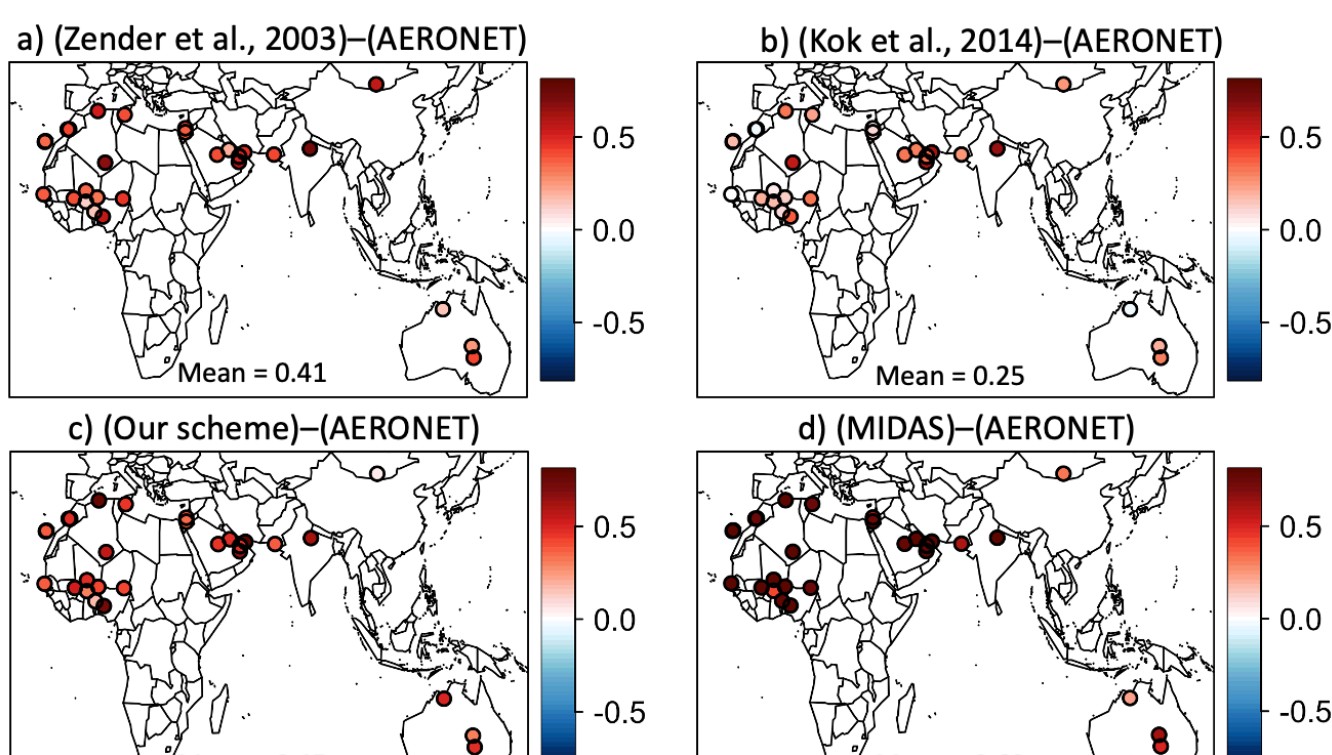

Figure 7. AERONET–SDA coarse mode AOD daily correlations for 2004–2008 over selected sites with CESM DAOD using (a) Z03, (b) K14, and (c) our study. (d) MIDAS DAOD versus AERONET–SDA AOD daily correlation. The values at the bottom of the panels represent the mean correlation across all AERONET stations.

As discussed above, our scheme's dust not only matches well with external dust datasets but also correlates better with meteorological drivers in day-to-day variability than Z03 and K14 (in Figs. S10 and S11), for a number of reasons. First, implementating more aeolian physics (Sect. 3) allows our scheme to better couple with the simulated boundary-layer dynamics, vegetation dynamics, and the water cycle in CESM2. For example, our scheme's emission strongly covariates with $u_{*s}$ (Fig. S10g) since the emission's dependence on $u_{*s}$ is not only in the K14 dust emission equation (Eq. 5) but also in the C19 intermittency scheme (Sect. 3.4), resulting in an enhanced sensitivity of emissions to the winds. Another example is that our emission's dependence on VAI is not only in the bare land fraction term (Eq. 4) but also in the vegetation drag partitioning (Eq. 9), enhancing the dust correlation with VAI (Fig. S10h and Fig. S11h). The second reason is because the use of $u_{*it}$ in the dust emission equation increases the likelihood of emission $F_d > 0$ in the $F_d$ time series. Z03 and K14 employing $u_{*ft}$ have a lot of times with emission $F_d = 0$ in the time series, weakening their emissions' temporal variability and thus the

covariation with $u_{*S}$. With more pronounced temporal fluctuations, our scheme's $F_d$ is further sensitive to the variability of $u_{*S}$ and correlates better with other driving variables than Z03 and K14. Thus, dust emission schemes using $u_{*it}$ will generate emissions that correlate better with the day-to-day variability of $u_{*S}$ than schemes using $u_{*ft}$. Third, the implemented aeolian processes allow more coupling between the driving fields such as boundary-layer meteorology and vegetation dynamics. For instance, as VAI regulates $u_{*S}$ through the vegetation drag partition effect, $u_{*S}$ carries both the temporal variability of $u_*$ and VAI. $u_{*S}$ thus almost dictates the temporal behavior of the our scheme's emission time series analogous to the concept of dimensionality reduction ($R \sim 1$ in the Sahara; Fig. S10g). Figures S10 and S11 show that our scheme's emission and DAOD are very sensitive to the day-to-day variability of meteorological and land surface variables, which means our scheme is likely also more sensitive to climate change and land use and land cover change (LULCC) in longer-term simulations.

### 5.4 Comparisons against other measurements of the dust cycle

We use more datasets of different dust cycle variables from other independent sources to evaluate our CESM2 dust cycle simulations regarding spatial variability. Figure 8 compares the simulated dust PM$_{10}$ concentrations (background colors) using various schemes versus observed dust PM from multiple stations (overlaid dots). Z03 has some strong overestimations compared with the measurements over the downwind regions of dust sources (dark red in the bottom panel of Fig. 8a), such as over Japan, southern Australia, and South Africa. Dust concentrations over the source regions are very high (e.g., the Taklamakan desert and the Australian desert in the top panel of Fig. 8a), due to the very localized and high Z03 emissions over the source regions (Fig. 2a). Our scheme in Fig. 8c reduces the exaggerations of dust strength in Z03 over Asia, Australia, and other secondary sources, mitigating the overestimations of dust PM as shown in the bottom panel of Fig. 8c compared to Fig. 8a. Our scheme mainly overestimates dust PM over the Sahara, which is commonly shared by Z03 and K14 and consistent with the previous discussion on regional DAOD (Zhao et al., 2022). Due to in the insufficient emissions over the Taklamakan, our scheme produces ~60 $\mu$g m$^{-3}$ of dust PM there, smaller than the ~100 $\mu$g m$^{-3}$ reported by other observational studies (e.g., van Donkelaar et al., 2016; Leung et al., 2018; van Donkelaar et al., 2021). Our scheme produces higher dust PM than K14 (Fig. 8b) over arid and semiarid regions, including the Gobi, the United States, and Patagonia. Compared with Z03's spatial correlation of $R = 0.80$ (in the log$_{10}$ space), our scheme yields a slight increase in the spatial correlation of $R = 0.90$. Overall, dust PM concentrations tend to be underestimated over the downwind regions (e.g., the Pacific and the Atlantic).

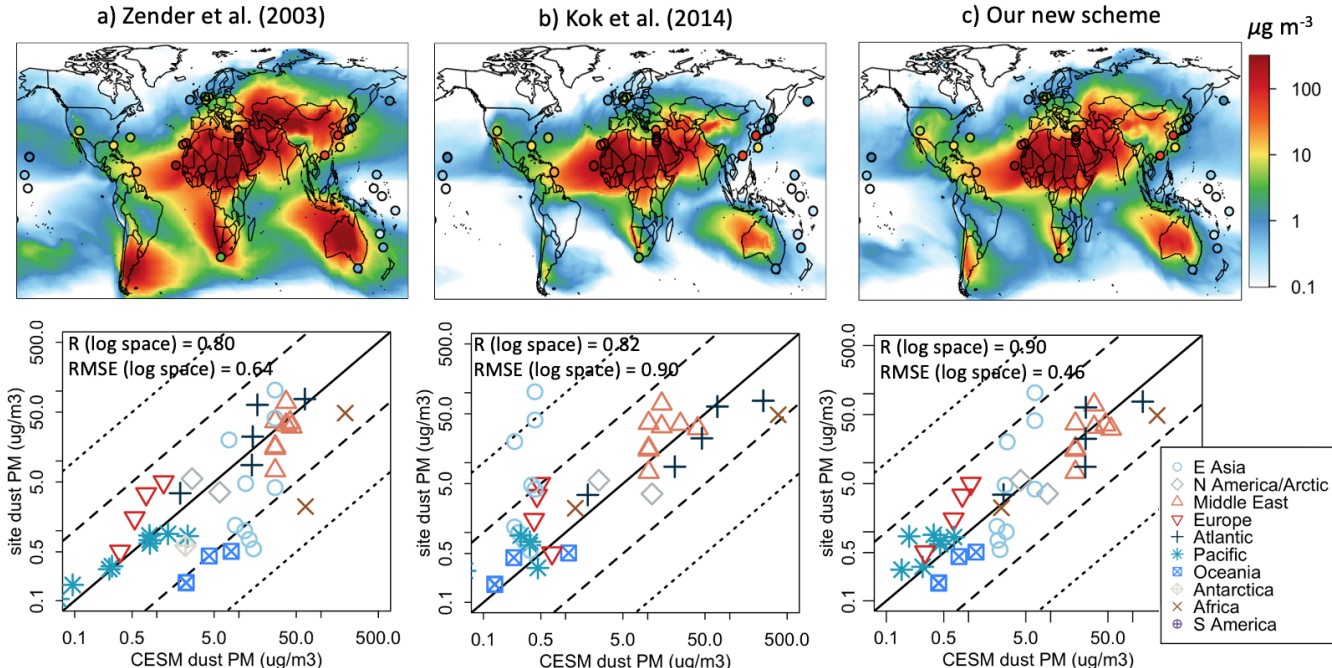

Figure 8. CESM2 dust PM$_{10}$ concentration (in $\mu g$ m$^{-3}$) vs. climatological in-situ PM$_{10}$ measurements (Sect. 4.4) for (a) Z03, (b) K14, and (c) our study. In the bottom panels, sites are labeled over different continents / oceans with different symbols and colors.

Figure 9 shows the dust deposition evaluation for all schemes. All schemes show that most deposition fluxes are concentrated over the source regions. Over remote areas (e.g., the central Pacific Ocean and the Southern Ocean), simulated deposition fluxes are small with an order of magnitude of ~10$^{-4}$ kg m$^{-2}$ yr$^{-1}$ or smaller (white color), whereas many measurements over those remote locations have an order of magnitude of 10$^{-3}$ kg m$^{-2}$ yr$^{-1}$ (light blue). It shows that the deposition schemes in CAM6 are problematic in that dust typically deposits too quickly; switching between dust emission schemes does not address nor mitigate the issue. Generally, the spatial patterns of dust depositions follow those of the DAOD (comparing Fig. 3 with Fig. 9). Our scheme has a higher correlation of $R = 0.65$ (in the log space) compared with $R = 0.49$ by Z03, but K14 has an even slightly higher $R = 0.69$. There is some underestimation of dust deposition over the downwind regions of Asia (e.g., the extratropical Pacific), likely due to the underestimated Asian dust in K14 and our scheme (but not in Z03 because of its abundant Asian dust). There is also some overestimation of dust deposition over the downwind regions of the Sahara (e.g., the equatorial Atlantic), which could be due to several possible reasons. There could be an overestimation of dry deposition due to an incomplete representation of deposition processes (e.g., Huang et al., 2021; Klose et al., 2021; Li et al., 2022; Meng et al., 2022). In particular, the dry deposition scheme in CAM6 (Zhang et al., 2001) was found to particularly overestimate dry deposition of fine dust (Li et al., 2022). In addition, previous studies indicated a possible overestimated tropical wet scavenging of dust, (e.g., Albani et al., 2014; van der Does et al., 2020). Fig. S12 shows the fraction of wet dust deposition flux to the total dust deposition flux from CESM2 using our scheme.

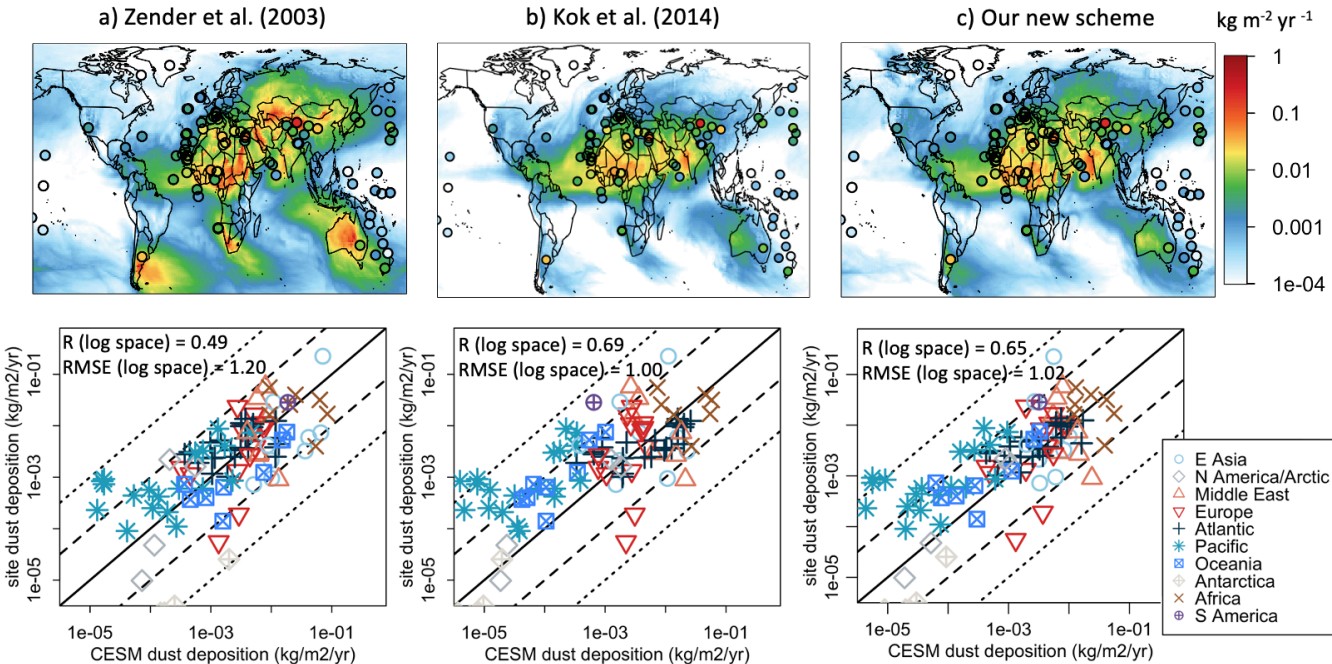

Figure 9. CESM2 dust total (dry + wet) deposition (in kg m⁻² yr⁻¹) vs. climatological in-situ deposition measurements for (a) Z03, (b) K14, and (c) our study. In the bottom panels, sites are labeled over different continents / oceans with different symbols and colors.

**5.5 Separating the contributions of drag partition and intermittency to our new scheme**

In this subsection, we briefly discuss a sensitivity experiment to separate the contributions of the hybrid drag partition scheme and the intermittency scheme to the improvements in dust cycle simulations produced by our new Leung et al. (2023) scheme. We performed the sensitivity experiment by turning off the Comola et al. (2019) intermittency scheme (experiment A using Sect. 3.1–3.3) to examine the effect of drag partitioning, and by turning off the hybrid drag partition scheme (experiment B using Sect. 3.1–3.2 and 3.4) to examine the effect of intermittency, respectively, on the resulting CESM2 dust cycle simulations. Here we focus on discussing the spatiotemporal variability of the simulated dust emission and DAOD.

Fig. 10 shows the main results of the sensitivity test. The left column shows experiment A with the effects of drag partitioning, and the right column shows experiment B with the intermittency effect. For expt. A, the maps of dust emission $F_d$ (Fig. 10a) and DAOD (Fig. 10c) show similar spatial patterns to those from our Leung et al. (2023) scheme (Figs. 2c and 3c). This means that the drag partition factor $F_{eff}$ dominates the spatial variability of our new scheme. It highlights the erodible regions across the globe and acts as a filter that shapes the spatial variability of $u_{*s}$ and $F_d$. $F_{eff}$ shift dust sources to more correct locations, such as the Bodélé Depression and El Djouf in the Sahara because of the use of the satellite-derived roughness map. For expt. B, which represents the intermittency effect, Fig. 10b shows substantially more emission fluxes from semiarid regions than in Fig. 10a due to the use of $u_{*it}$, which reduces the dust overestimations over hyperarid regions as previously discussed in Zhao et al. (2022). The $F_d$ pattern in Fig. 10b is different from our scheme's $F_d$ map in Fig. 2c, which means that Comola's intermittency scheme is sensitive to $u_{*s}$. The spatial variability of $F_{eff}$ will change that of $u_{*s}$, which subsequently changes the spatial variability of intermittency $\eta$ (Eq. 11c) and $F_d$ (Eq. 11b). Therefore, $\eta$ is controlled by $F_{eff}$ and the two variables share very similar spatial variability as shown in Fig. 1a–b. The DAOD pattern (Fig. 10d) also appear different from our scheme's DAOD in Fig. 3c. There is more dust

in various semiarid regions, and without using $F_{eff}$ the moderately high DAOD peaks are not constrained to the most erodible regions, such as El Djouf in Mauritania. Fig. 10e–f shows the daily DAOD correlation with MIDAS DAOD, which indicate that both drag partitioning and intermittency overall yield similar levels of correlations with MIDAS dust.

Overall, the sensitivity experiment shows that the drag partition scheme $F_{eff}$ dictates the spatial variability of our new scheme's dust. The drag partition scheme more correctly simulates the spatial pattern of dust emissions in major source regions, while the intermittency scheme more correctly simulate the balance between dust from major sources and marginal sources. For the intermittency scheme, the use of $u_{*it}$ enhances dust levels over semiarid regions, while $\eta$ is in general sensitive to $u_{*s}$ and the emission thresholds. Both the temporal variability of $F_{eff}$ and the intermittency contributes to the temporal variability of our scheme's dust to similar degrees.

Figure 10. Separating the contributions of the hybrid drag partition scheme (left) and the Comola et al. (2019) intermittency scheme (right) to our dust emission scheme (Leung et al., 2023). The rows represent simulated (a, b) dust emission, (c, d) dust aerosol optical depth (DAOD) with global means of 0.03, and (e, f) daily DAOD correlation with MIDAS DAOD from Gkikas et al. (2021).

## 5.6 Effects of employing a scale-aware adjustment to correct dust emission

In this subsection, we discuss the effects of using an empirical correction map ($\widetilde{K}_c$) to scale our scheme's dust emissions simulated in the native 0.9°×1.25° resolution to be consistent with 0.47°×0.62° emissions of our scheme (Sect. 3.5) on the simulated dust cycle in CAM6. We focus on the changes in the

DAOD spatial variability; changes in other dust cycle variables are shown in Fig. S13. Figure 11a shows the global DAOD of our scheme after correction, which is not visibly very distinct from the uncorrected DAOD in Fig. 3c. Figure 11b shows the ratio between the corrected DAOD and the uncorrected DAOD, both normalized to the same global mean, to better visualize their spatial variability discrepancies. It is worth comparing the map of DAOD discrepancies (Fig. 11b) to the map of emission discrepancies (Fig. 1c). The more prominent sources, e.g., the Sahara, have suppressed DAOD compared to other dusty regions ($\widetilde{K}_c \sim 0.8$–0.9; light blue in Fig. 11b). This is because, as discussed in Fig. 1c, high-resolution simulations produce more emissions from semiarid regions than low-resolution simulations but produce similar emission levels from primary sources as low-resolution simulations, leading to a relative suppression of dust over primary sources upon scaling to the same global mean DAOD. Many secondary dust sources have relatively enhanced dust levels, most noticeably the two American regions ($\widetilde{K}_c \sim 1.2$–1.8), but the absolute increases in DAOD are modest as the baseline DAOD levels over there are low. The Taklamakan/Gobi region also has a moderate rise in DAOD ($\widetilde{K}_c \sim 1.3$).

Since the high-resolution simulations generally pick up more emissions over semiarid regions, $\widetilde{K}_c$ tends to reduce the DAOD regional biases seen in Fig. 4 by enhancing the underestimated DAOD over semiarid regions and suppressing the overestimated DAOD over major sources. Comparing against the Ridley et al. (2016) regional DAOD (Fig. 11c-d), northern Africa has reduced DAOD and southern hemispheric regions have increased DAOD, hence slightly enhancing $R^2$ slightly from 0.82 to 0.84 and annual RMSE to drop from 0.053 to 0.048. Annual RMA regression slope modestly changes from 0.92 (in Fig. 4c) to 0.94. This shows that $\widetilde{K}_c$ helps reduce the biases of annual and regional mean DAOD predictions. However, since the errors mainly originate from seasonal biases, the improvements of using an annual $\widetilde{K}_c$ map are relatively modest. For instance, in Fig. 11d, the significantly underestimated MAM DAOD (in red) in Asia and the Middle East are still not resolved by using the annual $\widetilde{K}_c$. Using a seasonal or monthly $\widetilde{K}_c$ would more effectively reduce seasonal DAOD biases.

Although the correction map modestly improves the regional variability of DAOD, it does not necessarily produce improvements in comparisons against site-level dust observations. Figure 11e compares AERONET–SDA coarse mode AOD with the corrected DAOD of our scheme. Although the scatterplot has an increased RMA slope from 0.97 (in Fig. 5h) to 0.99, the $R^2$ value drops from 0.71 to 0.65 and the RMSE increases from 0.077 to 0.088. This is mainly due to the small DAOD underestimations over major sources like Mali/Niger and Bodélé/Sudan (see the "x" points). Our rescaled simulation has a reduced Mali/Niger DAOD that better matches Ridley's regional DAOD; however, it loses its local DAOD peaks and matches less well against AERONET–SDA AOD. There are also DAOD overestimations over the southern Middle East. This evaluation likely has a bias in geographical location because the errors are mainly from major sources; if more selected AERONET stations were in the Taklamakan/Gobi and the U.S., this evaluation against AERONET would possibly show better results because our rescaling reduces the DAOD underestimations over those regions. Overall, this evaluation shows that despite the better performance in capturing the regional DAOD variability using $\widetilde{K}_c$, it does not necessarily guarantee a better performance in the grid-scale or site-scale spatial DAOD variability.

Finally, Fig. 11f shows that the annual $\widetilde{K}_c$ has few effects on the temporal variability of DAOD simulations, which depicts the correlation map differences $\Delta R$ between our scheme's rescaled DAOD versus MIDAS DAOD ($R_{\mathrm{corrected}}$) and our scheme's uncorrected DAOD versus MIDAS DAOD ($R_{\mathrm{uncorrected}}$ from Fig. 6c). $\Delta R$ values are statistically insignificant across the globe (Sect. S1). It is reasonable that $\widetilde{K}_c$ changes little our scheme's DAOD temporal variability because $\widetilde{K}_c$ is a time-invariant map here that is meant to only change the spatial variability of the simulated dust.

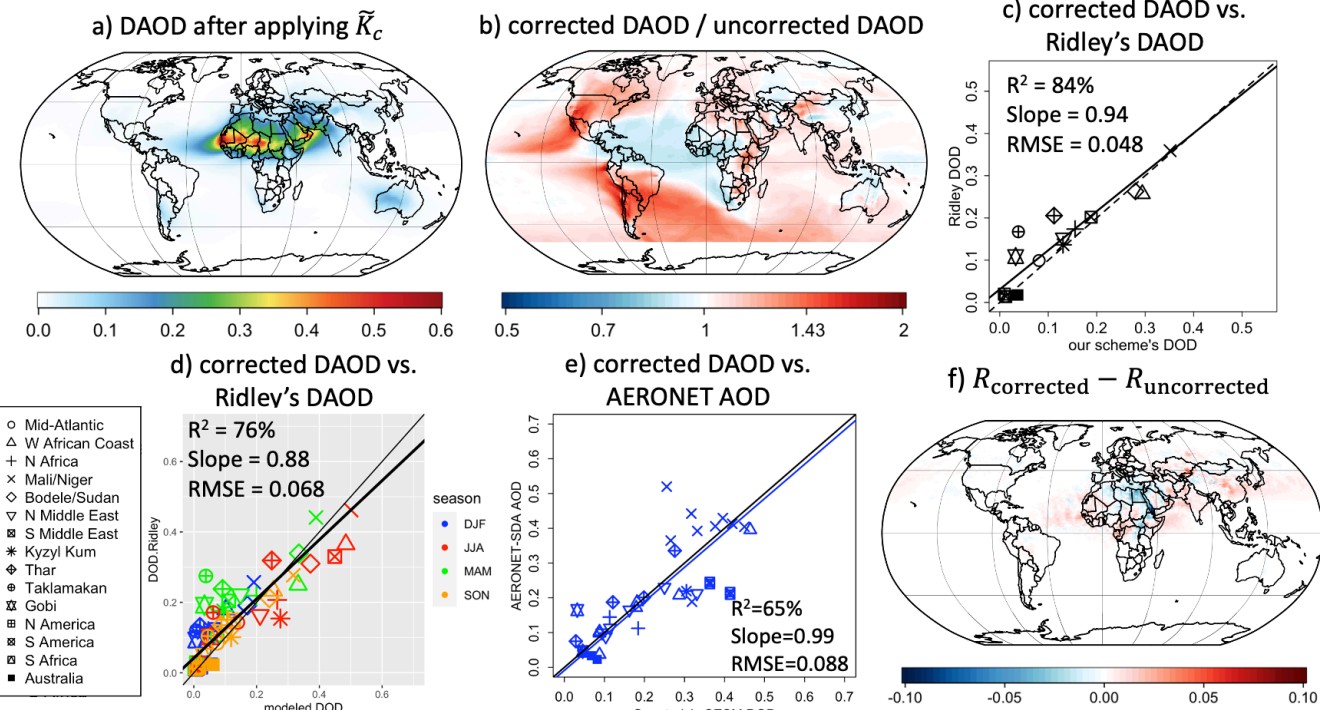

Figure 11. Effects of using the scaling map $\widetilde{K}_c$ to correct the 0.9°×1.25° CLM5 dust emissions on the CAM6 dust cycle. (a) DAOD spatial pattern simulated by our scheme after the global dust emission pattern is corrected by $\widetilde{K}_c$. (b) Corrected DAOD (Fig. 11a) divided by the uncorrected DAOD (Fig. 3c). Both DAOD maps are rescaled to have the same global mean to emphasize their difference in spatial variability. (c, d) Corrected DAOD versus Ridley regional DAOD (c) annually and (d) seasonally. (e) Corrected DAOD versus AERONET–SDA coarse mode AOD. (f) Changes in correlation maps ($\Delta R$) between corrected DAOD vs. MIDAS DAOD and uncorrected DAOD vs. MIDAS DAOD.

## 6. Discussion and Conclusions

This study has evaluated the new formulation of the dust emission scheme proposed in Leung et al. (2023) against measurements and compared its performance against existing emission schemes in CESM2. The major modifications implemented into CESM2 are the following: 1) updating the soil median particle diameter (as an input parameter to the dust emission threshold calculation) from 75 $\mu$m as proposed by Zender et al., (2003a) to 127 $\mu$m; 2) including a parameterization for the drag partition effect that accounts for the impact of not only rocks but also green and brown vegetation on reducing the wind stress available for soil erosion; 3) implementing the intermittent dust emission parameterization by Comola et al. (2019) that accounts for the effects of boundary-layer turbulence on dust emissions, and 4) rescaling the CESM2-native resolution dust emissions toward high-resolution emissions. Following Leung et al. (2023), these modifications are 5) implemented on a newer dust emission parameterization of Kok et al. (2014b; K14) instead of the default Zender et al. (2003a, b; Z03) scheme in CLM5, although the modifications 1–4 can also be implemented on top of Z03 or any other emission scheme. The major advances of Leung et al. (2023) are mainly that the drag partition effect successfully moves emissions toward important dust sources (e.g., the Bodélé Depression and El Djouf) and thus generates a more realistic spatial distribution of dust than Z03 or K14. Also, the intermittency scheme generates more emissions from semiarid regions and even high-latitude regions, in agreement with observations.

Our new scheme showed improvements over previous schemes (Z03 and K14) in terms of both the spatial and temporal variability of dust cycle variables. For instance, our scheme showed improved agreement against the annual and seasonal regional DAOD quantified by Ridley et al. (2016). Indeed, our scheme's annual regional DAOD had an $R^2$ of 0.82 and RMSE of 0.053, compared to Z03's $R^2$ of 0.44 and RMSE of 0.080. Evaluating against the AERONET–SDA coarse mode AOD, our scheme's DAOD yielded an $R^2$ of 0.71 and RMSE of 0.077 compared to Z03's $R^2$ of 0.36 and RMSE of 0.15. Our scheme also generated improved spatial distributions of dust $PM_{10}$ concentrations and depositions against site measurements of $PM_{10}$ and deposition fluxes than Z03 (Figs. 8 and 9). For day-to-day temporal variability, our scheme's DAOD also matched the MIDAS DAOD better over most of the Dust Belt than Z03 DAOD (Fig. 6e), with larger correlations of on average $\Delta R \sim 0.15$ (*p*-value < 0.05; Sect. S1). Our scheme's DAOD also showed high daily correlation values (with a mean of 0.45) against AERONET–SDA daily AOD time series (Fig. 7). However, our scheme's DAOD generally showed worse performance in representing the day-to-day dust variability over East Asia (Fig. 6e and Fig. 7c), likely because of the significant low bias of dust (DAOD $\sim 0.1$) over the Taklamakan Desert such that dust over East Asia was dominated by other transboundary dust signals instead of dust from the Taklamakan. Generally, our scheme better represented the spatial variability of Ridley's regional DAOD, the site-level AERONET DAOD, the site-level dust PM, as well as the day-to-day temporal variability of MIDAS DAOD than the default Z03 scheme. Our scheme's dust also shows better correlations with driving meteorological and land surface variables (e.g., $u_{*s}$, VAI, $w$; Figs. S10 and S11), and is thus likely more sensitive to climate change and LULCC than other emission schemes' dust. Since the more physically based Leung et al. (2023) scheme showed improvements in the model–observation comparison (Sect. 5), the developments in Leung et al. (2023) will be introduced into a future CLM (and CESM) version for the benefit and use of the dust community and the CESM community.

Regardless of which dust emission scheme is used, Fig. 4 shows that CESM2 tends to overestimate DAOD over major sources (e.g., the Sahara) and underestimate DAOD over marginal source regions (e.g., SH sources) and downwind regions (e.g., oceans). This result is consistent with previous findings across multiple ESMs (Zhao et al., 2022), likely due to the insufficient dust emissions coming from the semiarid regions. Theoretically, employing the intermittency scheme helps generate more emissions from semiarid regions, thereby reducing the DAOD biases and increasing the RMA slopes toward one. Our scheme did yield RMA slopes that most closely match the 1:1 line among all emission schemes (annual RMA slope = 0.92).

We then tested the proposed modification of rescaling dust emissions of lower resolutions toward higher resolutions by Leung et al. (2023). We used the 0.9°×1.25° and 0.47°×0.62° simulations from CESM2 to construct an annual correction map $\widetilde{K}_c$ (Eq. 12) used to rescale and correct the CESM2-native 0.9°×1.25° dust emissions to the spatial variability of the finer resolution (0.47°×0.62°) emissions. Employing the scaling map $\widetilde{K}_c$ further reduced the CESM DAOD over hyperarid regions and enhanced DAOD over secondary sources. Since $\widetilde{K}_c$ is a time-invariant map, employing $\widetilde{K}_c$ has little effects on improving the seasonal / day-to-day variability of the CESM DAOD (Figs. 10d and f). Employing an annual $\widetilde{K}_c$ to dust emissions modestly improved the spatial variability of atmospheric dust but altered little its temporal variability. This modification differs from other modifications proposed by Leung et al. (2023) in that it does not necessarily improve the process representation of the dust scheme, but the methodology makes the scheme more scale-aware and consistent toward high-resolution dust simulations. Our new process-based emission scheme can still be employed in ESMs and in regionally refined models (RRMs) with different horizontal resolutions without the use of a scale-aware adjustment.

Although CESM2 with our updated dust emission scheme thus shows an improved spatiotemporal pattern of DAOD, some important biases remain. Our scheme overestimates DAOD levels over the Horn of Africa (HoA) and Australia. There are also dust underpredictions over the Taklamakan. The DAOD hotspot over the HoA (in Fig. 3c) is due mainly to the very high $u_*$ in MERRA-2 nudged CESM2 (> 0.5 m s$^{-1}$ in Fig. S4), resulting in a high dust emission flux of ~ 1–2 kg m$^{-2}$ yr$^{-1}$ (Fig. 2c) that is almost as high as emissions over the Bodélé Depression. The unrealistically high emissions from the HoA produce a dust plume extending to the Middle East (e.g., Oman), central Asia, and as far as the Thar Desert due to the downwind transport. This problem also occurs in the default K14 and Z03 schemes (Figs. 2a-b), although Z03 uses a source mask that significantly reduces the HoA emissions. As for Australia, the relatively low soil moisture over the central and western parts of the country results in somewhat higher emissions in western than eastern Australia. Our study therefore shows a modest annual DAOD peak of ~0.2 (Fig. 3c) over western Australia (e.g., the Great Sandy Desert and the Gibson Desert), which is different from the smaller eastern peak of ~0.1 in MIDAS/Aqua DAOD (Fig. 3d). In addition, CESM2 shows an annual DAOD of only ~0.1 over the Taklamakan/Gobi region in China, which is a strong underestimation compared with the yearly DAOD of ~0.35 from MIDAS/Aqua. This DAOD low bias occurs because CESM2 simulates over there a low emission flux (Fig. 2c) as a result of the moderately high soil moisture $w$ and aeolian roughness length $z_{0a}$ (compared with the Sahara). Furthermore, CESM background dust levels over downwind regions (e.g., the tropical Atlantic and the extratropical Pacific) are generally underestimated compared with MIDAS DAOD, likely because of the strong dust depositions and short lifetimes of dust, leading to dust preferentially depositing over the land.

Although we have attempted to improve the dust emission model in both CLM5 and CAM6, there are more areas of dust cycle modeling that warrant further developments. We summarize several main issues in dust modeling that should be addressed in future versions of CESM and other ESMs to further enhance the dust modeling performance in the land and atmospheric models:

1. Dust emission physics: There are several mechanisms that affect the dust emission threshold that are not currently accounted for in most dust emission modules. First, soil crusts due to soil microbes can strongly aggregate soil particles and prevent winds from eroding the soils (Rodriguez-Caballero et al., 2018). Second, the impact of anthropogenic activities, such as agriculture/tillage, on dust emission is not explicitly included in dust emission modules, although new parameterizations for anthropogenic dust emissions are under development (e.g., Xia et al., 2022). Third, apart from saltation bombardment, soil particles can enter the atmosphere through direct aerodynamic entrainment (Klose and Shao, 2012). Models have been developed to represent direct particle entrainment into the atmosphere (Klose and Shao, 2013; Klose et al., 2014).

2. Dust size distribution: Apart from dust emission physics, there are problems in representing the dust aerosol size distributions in the atmosphere. Coarse and super-coarse dust particles are substantially underestimated (Adebiyi and Kok, 2020), and recent studies are working on implementing the coarse and super-coarse dust size bins (CAM4; Meng et al., 2022) or modes (Ke et al., 2021; CAM5) in different versions of CAM, such that CESM2 can represent the impacts of large dust particles on climate and ecosystem. Recent studies further found that the geometric standard deviations (GSDs) of the accumulation and coarse modes in CAM6 are too narrow (Wu et al., 2020; Li et al., 2022), which subsequently adversely impacted the dust deposition, lifetime, and size distribution of the CAM6 simulations.

3. Dust deposition: Dust deposition in ESMs is generally overestimated, and dust lifetime is underestimated (e.g., Albani et al., 2014; van der Does et al., 2020; Huang et al., 2021) due to a few reasons. First, recent studies found that dust particles are highly aspherical, which subsequently alters the aerodynamic resistance of dust and slows down the dry deposition velocity of dust (Huang et al., 2021). This finding increases the lifetime of coarser dust particles and also

reduces the mass extinction efficiency (Huang et al., 2023). This effect of dust asphericity on dry deposition and extinction is being implemented into climate models (e.g., Klose et al., 2021; Li et al., 2022; Meng et al., 2022). Second, the default dry deposition scheme in CAM6 (Zhang et al., 2001) is known to overestimate dry deposition of fine dust, and Li et al. (2022) has employed a newer dust deposition scheme (Petroff and Zhang, 2010) to resolve the issue. Third, the modal aerosol model (MAM) of CESM2 merges dust and other aerosols (e.g., sea salt) into the same modes (e.g., accumulation and coarse modes) with internal mixing, such that the wet deposition of dust is likely overestimated (e.g., the Atlantic Ocean) due to the higher hygroscopicities of other aqueous aerosols (Neale et al., 2012). Fourth, studies reported that modeled dust depositions are too high over the tropical oceans (Albani et al., 2014; van der Does et al., 2020).

4. Speciation of dust: CESM and other ESMs mostly parameterize dust as a single mineral (e.g., aluminum silicate; Emmons et al., 2020), which cannot adequately represent a suite of chemical reactions, radiative effects, and cloud processes that depend on mineralogy. Recent studies have initiated the modeling of multiple dust species (e.g., haematite, quartz, illite, feldspar, calcite) and better represented the dust optical properties and radiative effects (Li et al., 2021, 2022; Gonçalves Ageitos et al., 2023). Emergence of satellite measurements of global soil mineralogy such as from the Earth Surface Mineral Dust Source Investigation (EMIT; Green et al., 2020; Thompson et al., 2020) mission under NASA will help better represent dust species from specific source regions.

5. Chemistry and cloud processing: Having accurate simulations of the modeled spatiotemporal variability of dust requires dust chemistry and dust–cloud interactions in ESMs, because they are crucial for simulating dust aging and dust removal processes. A correct mineralogical representation of dust is essential for representing the role of dust in atmospheric chemistry and aerosol–cloud interactions. Previous studies have documented multiple chemical reaction pathways via which dust interacts with tracer gases and aerosols (Gaston, 2020; Adebiyi et al., 2023; Kok et al., 2023). Dust acts as a source or sink of multiple chemical species, such as oxidants (e.g., ozone), aerosol precursors (e.g., sulfur dioxide and nitric acid), aerosols (e.g., via coagulation), halogens (e.g., chlorine), and more (Tang et al., 2017; Mitroo et al., 2019; Gaston, 2020). Furthermore, although many ESMs include the impacts of dust on ice cloud formation (Storelvmo, 2017), dust seeding on warm cloud formation are quantified in only a few ESMs (e.g., McGraw et al., 2020) as dust is considered hydrophobic by many ESMs. Chemical dust aging is crucial for dust to gain hygroscopicity and become effective cloud condensation nuclei (CCN). A comprehensive mineralogical representation of dust and a more complex heterogeneous dust chemistry are required to adequately represent the roles of dust in the formation of warm, ice, and mixed phase clouds, as well as the effects of dust–cloud interactions on indirect radiative effects and forcings in ESMs.

6. Observations for dust modeling development: The uncertainties in dust modeling are due to not only the uncertainties in the parameterized dust processes but also the uncertainties in the input data of these parameterized processes. The availability of observations will influence the uncertainties of dust modeling both by entering the simulations as input datasets and by shaping the parameterization development. For instance, Leung et al. (2023) used a global soil particle diameter $D_p = 127\ \mu$m (Sect. 3.2) for computing the emission thresholds since there were too few site $D_p$ measurements, which hindered the accuracy of the simulation of the global distributions of emission thresholds. We also speculated in Sect. 5 that some of our simulated DAOD biases could be due to biases in the meteorological inputs rather than the missing physics in the dust scheme. More observations will also allow us to develop more accurate parameterizations for dust. For instance, recent coarse dust observations (e.g., Adebiyi and Kok, 2020) justified the importance of and quantified the necessary parameters for formulating the coarse dust modes in ESMs (e.g., Ke et al., 2022; Meng et al., 2022). Having more observations of dust and its dependent variables is

highly warranted to reduce the uncertainties of dust simulations by improving the dust schemes and reducing the uncertainties of input dependent variables.

Finally, while many dust modeling studies focused on improving and evaluating the spatial representation of modeled dust, the importance of evaluating the temporal variability of modeled dust is likely undervalued in global dust modeling studies. Relatively few dust studies (e.g., Zhang et al., 2013; Klose et al., 2021; LeGrand et al., 2023) provide evaluation of the temporal changes in dust emissions and DAOD. This study represents one of the early attempts to conduct a global-scale evaluation of the day-to-day variability (Figs. 6–7) of the simulated dust time series (along with studies like Klose et al., 2021). The next step in improving dust modeling should be on enhancing the long-term (interannual or interdecadal) variability of dust, especially since recent studies (e.g., Kok et al., 2023) found that ESM dust trends do not reproduce the historical increasing trends of dust. It is highly warranted to investigate how transient climate change and LULCC regulate the long-term variability of observed dust and reproduce them in ESMs. Improving long-term ESM dust predictions will also benefit the study of the epidemiological consequences of future dust changes on human health, risk management, and mitigation strategies (Philip et al., 2017; Achakulwisut et al., 2019; Bauer et al., 2019; van Donkelaar et al., 2021).

**Appendix. Mathematical symbols of major variables defined in this study.**

| | |
|---|---|
| $\eta$ | Intermittency factor |
| $\kappa$ | Fragmentation exponent |
| $\rho_a$ | Air density |
| $\rho_p$ | Soil particle density |
| $\varphi$ | Sandblasting efficiency in Zender et al. (2003) emission scheme |
| $\sigma_{\tilde{u}_s}$ | Standard deviation of instantaneous wind fluctuations |
| $A_r$ | Fractional rock area |
| $A_v$ | Fractional vegetation area |
| $a$ | Tuning constant for threshold gravimetric soil moisture |
| $C_d$ | Soil erodibility coefficient |
| $C_{MB}$ | Proportionality constant in Zender et al. (2003) emission scheme |
| $C_{tune}$ | Proportionality constant for Kok et al. (2014) emission scheme |
| $D_p$ | Soil particle diameter |
| $F_d$ | Dust emission flux |
| $F_{d,c}$ | Simulated dust emission in coarse resolution |
| $F_{d,f}$ | Simulated dust emission in fine resolution |
| $F_{eff}$ | Hybrid drag partition effect |
| $f_{clay}$ | Clay fraction (from 0 to 1) |
| $f_{eff,r}$ | Rock drag partition factor |
| $f_{eff,v}$ | Vegetation drag partition factor |
| $f_m$ | Soil moisture effect |
| $f_v$ | Vegetation cover fraction |
| $g$ | Gravitational acceleration |
| $\tilde{K}_c$ | Scaling map for correcting spatial variability of simulated dust emission in coarse resolution toward simulated emission in fine resolution |
| LAI | Leaf area index |
| $S$ | Preferential dust source filter in Zender et al. (2003) emission scheme |
| SAI | Stem area index |
| VAI | Vegetation area index |
| $VAI_{thr}$ | Threshold vegetation area index |
| $T$ | Proportionality constant in Zender et al. (2003) emission scheme |
| $u_{*ft0}$ | Dry fluid threshold |
| $u_{*ft}$ | Wet fluid threshold or static threshold (accounting for soil moisture effect $f_m$) |
| $u_{*it}$ | Impact threshold or dynamic threshold |
| $u_{*s}$ | Friction velocity at the soil surface |
| $u_{*t}$ | Dust emission thresholds (generic for indicating both fluid and impact threhsold) |
| $u_{*st}$ | Standardized fluid threshold |
| $\tilde{u}_s$ | Instantaneous wind velocity |
| $w$ | Gravimetric soil moisture |

| | |
|---|---|
| $w_t$ | Threshold gravimetric soil moisture |
| $z_{0a}$ | Aeolian roughness length (for rocks and plants) |
| $z_{0s}$ | Smooth roughness length (for soil grain) |

**Financial Support**

D. M. Leung and J. F. K. are funded by the National Science Foundation (NSF) grants 1552519 and 1856389. L. L. and N. M. M. acknowledge support from the Department of Energy (DOE) DE-SC0021302 and the Earth Surface Mineral Dust Source Investigation (EMIT), a NASA Earth Ventures-Instrument (EVI-4) Mission. M. K. has received funding through the Helmholtz Association's Initiative and Networking Fund (grant agreement no. VH-NG-1533). C. P. G.-P. acknowledges support from the

European Research Council under the European Union's Horizon 2020 research and innovation programme (grant n. 773051; FRAGMENT) and the AXA Research Fund (AXA Chair on Sand and Dust Storms based at the Barcelona Supercomputing Center).

**Acknowledgements**

Computing and data storage resources, including the Cheyenne supercomputer (https://doi.org/10.5065/D6RX99HX), were provided by the Computational and Information Systems Laboratory (CISL) at the National Center for Atmospheric Research (NCAR). We thank the AERONET team and site Principal Investigators and managers for the maintenance of the AERONET data record. D. M. Leung thank Dr. Catherine Prigent for providing the satellite-derived roughness length dataset for use

in the CESM2.

**Author Contributions**

JFK and DMLe conceptualized the study. DMLe performed the model development, conducted the simulations, analyzed the simulation results, and conducted the evaluations. DMLe wrote the original

manuscript and plotted all figures under JFK's supervision. LL, CPG-P, MK, ST, DMLa, NMM, and EK assisted the conceptualization and model development. All authors contributed to the manuscript preparation, discussion, and writing.

**Competing Interests**

The authors declare no competing interests.

**Data and Code Availability**

The MODIS Dust Aerosol (MIDAS) daily dust aerosol optical depth data from Gkikas et al. (2021) are available at https://zenodo.org/record/4244106#.ZEbR7OzMIlI. The AERONET site AOD data are

available on http://aeronet.gsfc.nasa.gov. The satellite-derived aeolian roughness data are available upon contacting Catherine Prigent. The in-situ dust PM and dust deposition measurements are available through https://zenodo.org/records/6989502/files/LLi2022GMD.Observations.tar.gz?download=1, which was originally from Table S2 of Albani et al. (2014) and Table S2 of Mahowald et al. (2009). ESA CCI land cover can be obtained from https://www.esa-landcover-cci.org/?q=node/164. The CESM2 code for the

new dust emission scheme is available at https://github.com/ESCOMP/CTSM/pull/1897/files.

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
