# Peer review of "A new process-based and scale-aware desert dust emission scheme for global climate models – Part II: Evaluation in the Community Earth System Model (CESM2)"

_EGUsphere, 2023_

## Referee Comment (RC2)

**Review on the manuscript**
**« A new process-based and scale-aware desert dust emission scheme for global climate models – Part II: evaluation in the Community Earth System Model (CESM2)”**

Submitted to Atmospheric Chemistry and Physics
(https://doi.org/10.5194/egusphere-2023-823)
By Danny M. Leung, Jasper F. Kok, Longlei Li, Natalie M. Mahowald, David M. Lawrence, Simone Tilmes, Erik Kluzek, Martina Klose, and Carlos Pérez García-Pando

**General comment :**

The objective of this paper is to evaluate of the improvement of the dust emission parameterizations of the Earth Systems Model (ESM) CESMS2 described in a companion paper, by comparison with available synthesis of measurements and with the previous parameterizations included in this ESM.

In its present state, thee paper is not totally self-sufficient since the modifications of the dust emission scheme are only briefly recalled and in a very factual way. In particular, the main arguments that justify the selection of a given parametrisation or the values of some key parameters are sometimes missing and only few elements concerning their possible limitations are given. Some sections need further clarification or verifications for consistency with the companion paper. The reader can understand that including this evaluation in the Part I paper would have make it much too dense and too long but the writing of this section should be improved.

The work presented in this paper results for several modification of the dust emission scheme that have been included in the CESM2 Earth Systems Model. These modification are mainly related to processes that include soil and surface parameters : the soil size distribution described here by the soil median diameter, the drag partition due to the presence of rocks and vegetation and the soil moisture that both modulate the erosion thresholds. Some of these parameterizations are not new so the main contribution of this work is their tuning, which largely depends on the used input data used for the simulations. The inclusion at the global scale of a parameterization of the intermittency of the saltation, and the differentiation between the fluid and the impact threshold, are two original aspects. The sensitivity of these two factors to the input wind speeds and soil moistures would have deserve more discussion since it has significant impact on the final results. To take into account the limitation due to the coarse spatial resolution of global model, a correction factor is used to scale the dust emissions computed at low spatial resolution to the dust emissions computed at a higher spatial resolution. This correction is also dependent on the initial input data set for both the surface properties (soils, land use, roughness) and the meteorological parameter but the associated uncertainties are not discussed. In addition, this up-scaling approach does not account for smaller scale dynamical features, such as meso-scale convection producing "haboobs" are not properly represented in global scale. This limitation and the consequences on the final scores of the simulations should, at least, be mentioned in the manuscript.

The two companion papers together represent a valuable piece of work regarding the examination of the different steps involved in the dust emission modelling and the possibility (or not) to improve the representation of the physical processes. The final configuration evaluated in this paper, provides a  significant increase of the agreement with observation compared to the two dust schemes used for comparison. However, the new scheme contains several tuning parameters that allows the compensation of uncertainties. In fact, the general approach adopted in this work assign almost all the uncertainty of the simulated dust emissions to the surface parameters. This implies that the final configuration of the dust emission scheme

is not independent of the meteorological input fields used for the simulation. Except at the very end of the manuscript, there is almost no mention of the uncertainties on the main drivers of the dust sources and fluxes : the wind velocity and the soil moisture. This can be a relevant approach to tune a model that will then be used in the same configuration than described in the manuscript. But the relevant of the new dust scheme for othet models is thus questionable. It is even questionable for the CESM2 model without nudging to MERRA meteorological reanalysis.

Finally, if the objective of this work is to improve the performed of the CESM2 model in the presented configuration (nudged to MERRA configuration, and thus as close as possible to present day available observations) the approach is relevant, but if the intention is to recommend these modifications as more generic parametrization, it is very questionable. In this context, the objective of "examining how including more Aeolian processes can benefit to dust modeling performance" is not fully reached. I thus encourage the authors to define more precisely their scientific objectives and to adapt their conclusions accordingly.

I would thus recommend the publication of this manuscript after some revisions based on these general comments and on the detailed comments described below.

**Detailed comments :**

**Abstract :**
Line 28 : the sentence "incorporating a realistic soil size distribution " should be replace by "incorporating a simplified soil size representation", since the soil size distribution is represented by a median diameter and not a distribution and that a unique diameter is used.

**Introduction :**

Line 79-90 : In this section, the limitations of GCM et ASM due to their coarse spatial resolution is discussed but without any link with the small-scale meteorological processes and phenomena that are poorly or not represented in global models such as mesoscale convective systems, sea breezes, low level jets ..

**Part 2. Description of CESMS and its default dust scheme**

Line 64-65 : I would suggest to replace "interparticle forces due to soil microbes" by "interparticle forces involved in soil crusting"

Line 145-149 : Please replace "uses a global optimal diameter of Dp=75µm for the threshold calculation" by " uses a unique diameter Dp=75µm that is optimum for wind erosion since it corresponds to the lowest threshold"

Line 129-135 : The authors argued on the necessity to distinguish the fluid and the impact thresholds whose values differ by 20%. Is the precision of the modelled wind speeds higher than 20% ?

Line 155-156 : Kok, et al. (2014b) tested three values of a (0.5, 1 and 2) but the presented results are for a =1, the values that gives the best agreement with observations.

**Part 3. Modification of the CESM2 dust emission scheme**

Line 250-254: the description of the "erodibility" and its link with moisture is not clear and $U^*_{st0}$ is not defined

Line 261 and 262 : What means "bridging the gap between" scales and how if the value of 0.5 determined ?

Line 283 : As a matter of fact, using $\gamma = 1.65 \ 10^{-4}$ kg s$^{-2}$ instead of the value of 3. $10^{-4}$ kg s$^{-2}$ recommended by Shao and Lu (2000) gives almost the same minimum threshold than IW82. So numerically, the change should not be very sensitive.

Line 265-270 : If the threshold of VAI is set to 1 instead of 0.3, it automatically increases the extend of the dust sources. How much does it increase the extent of semi-arid areas ? Since the total emissions and the global AOD are tuned, how does it impact the spatial distribution of the simulated emissions?

Line 290-291 : It is not clear how the value of a=2 has been selected. Has the value of a=1 being maintained for the K14 simulation as in Kok et al. (2014b) ?

Line 298-299 : The conclusions from Martin and Kok (2019) are based on results obtained for soils with median diameter ranging from 0.3 to 0.5 µm in coastal dunes areas and poorly represent the dynamics of fine particles due to sampling limitations. And the conclusion of equal susceptibility concerns particles in the $D_i/D_{50,bed}$ size classes. It is constant with the much lower median diameter considered in this study as a reasonable approximation (127µm) and more generally to desert soils ?

Line 301 : Please, indicate the number of data sets used investigate the link between PSD and soil texture.

Line 303-304 : The author argued that the range of minimum fluid threshold (0.204 to 0.298 m.s$^{-1}$) that correspond to the range of $D_{med}$ in the soil data set (40 to 250 µm) can be considered as negligible. But this difference is comparable to the difference between the fluid and the impact threshold (Line 129-131) and the authors argue on the necessity to account for it. This is not consistent. An acceptable argument is that the available data sets does not allow to define precisely the soil size distribution and thus the minimum wind threshold velocity in the arid and semi-arid areas.

Line 307 : Finally, what is the value of the minimum $U_{*t}$ corresponding to Dp=127 µm ? How different is it from the threshold for Dp=75 µm used for Z03 ? The minimum threshold used in Kok et al. (2014) is much lower (0.16 m.s$^{-1}$). How does it impact the difference in the dust simulations with K14 and the new scheme ? Is there any compensation with the coefficient of tuning of the moisture impact  (a=1 in K14 and a=2 in this study) ?

Line 339 : According to Marticorena and Bergametti (1995) $z_{0s} = D_p/30$ and not $2D_p/30$.
In Marticorena and Bergametti (1995), X was set to 10 cm,  while a value of 10 m is used here. A short justification of this choice would be welcome regarding the different values used in the literature ( X=10 cm in Marticorena and Bergametti (1995) for small roughness elements and low roughness density ; Y=12,225 cm in Mac Kinnon et al. (2004) for vegetated surfaces with relatively large and high shrubs and high roughness density; X=40 cm in Pierre et al. (2014) on

millet fields and savannah). And why not using different X for the different land cover categories ?

Line 357-375 : From the description given here, the drag partition due to vegetation and to rocks appear as represented independently : there is no possibility to represent a surface covered by rocks and vegetation. Since a data base allows to estimate the fractional area of rocky surface and the fractional area for vegetated surfaces, it is not clear why and average $F_{eff}$ is needed. The dust fluxes could have been computed for each type of surface and then summed up at the grid cell scale.

Line 380-392 : The inclusion of a parameterization that account for the intermittency of the saltation is relevant in terms of physical process. The parameterization from Comola et al. 2019 allows to account for the occurrence of saltation when the wind friction velocity range between the fluid and the impact threshold whose values differs by about 20%. This raises again the question of the accuracy of the simulated wind speed that is the input parameter for the computation of the wind friction velocity. Indeed in the paper by Comola, the intermittency factor is estimated either from in-situ measurements of the wind friction velocity either using an historical 10m wind speed data bases re-scaled to the in-situ measurements.
It is not explicitly mentioned in section 3.4 but the application of this parametrization also requires the estimation of the Monin-Obukov length from the meteorological model output which also questions the precision of the estimation.

Line 403-405 : Following the previous comment, it appears that decreasing the minimum threshold by using the impact threshold and the intermittency parametrization increase unrealistically high emission in marginal source region. The emissions are thus lowered down by decreasing K. The authors should examine the possibility that the modelled winds are not correct and may exceed the minimum threshold too often. Another source of uncertainty is the fact that the intermittency parametrization was established on a site with a defined wind speed distribution and thus may not be directly applicable in other places with a meteorological context.

Line 404-405. The author mention that following Leung et al (2022), they "cap K at 5". But in Leung et al (2022) the comments of equation 13a and 13b state that K is limited to 3.

Line 421-422 : It is not clear what Leung et al. (2022) demonstrated on the scale of the dust emission.

Line 437-450 : The capability of GCM and ESM that use spatial resolution of the order of 1° is a key question. This was questioned in particular for regions with complex topography such as the region of the Bodele Depression and addressed by regional modelling.
The upscaling method proposed here consists in weighted the dust emissions computed at coarse resolution by the one computed at a higher resolution, taking advantage of the fact that the input data are available at this scale. This allows to account for the smoothing of the input parameters but it does not account for small scale meteorological features such as meso-scale convective systems that can dominated the dust emissions in some regions like the Sahel. This limitation should be stated and discussed.
It is not clear why the dust emissions computed at the two spatial resolutions are normalized to the total emission to perform the scaling. The normalization implies that if dust emissions are increased in one place, it will be reduced in another. As a matter of fact in figure 1, large areas

in the Sahel, South Africa and Australia have correction factors lower than 1. Why not scaling directly the intensity of the computed emissions?

Final comment on section 3 : At almost all stages of the improvement of the dust emission scheme, adjustable parameters are involved that have been fixed to obtain realistic simulations. But the criteria used to evaluate what is realistic are generally not described. If it is the level of agreement with the observations presented in the following section, it should be clearly stated.

**Part 4. Observational and reanalysis datasets for evaluating the dust emission scheme**

Line 503-504 : Please precise that the mean AOD used by Kok et al. (2021) based on the data set Adebiyi et al (2020 ) (but not shown in the paper) are estimated from reanalysis that used satellite products.  It would be nice to have the values for the 15 dusty regions in the supplementary material.

Line 523-524 : The overlaying of the AERONET values on figure 5 is unreadable on such small figure. Please add the values and the location of the stations in the supplementary material. In addition, according to the AERONET guidelines for data use and publication, when publishing data from 'many' sites, a general acknowledgement is expected. Citing the appropriate key AERONET papers as well as citing relevant manuscripts pertaining to previously published site data is also expected.

Line 527-539 : Except for the temporal resolution, the MIDAS data set does not seems very different by construction than the combined data sets of Ridley et al. (2016) and Adebiyi et al (2020) used by Kok et al. (2021). Was the consistency of the MODAI data set and the global and seasonal average used by Kok et al. (2021) ?

Line 540-549 :  Please add, in the supplementary material, the mean values, the location of the stations and the time period for which PM and deposition measurements are available. The relevant references should be given in order to properly credit the measurements producers/owners.

**Part 5. Model evaluation**

Line 557-569 :  This section is a little bit confusing. Should the reader understand that there is a double tuning, one to reach a global annual DAOD of 0.03 and another to reach an annual mean DOAD of 0.1 in average for the 15 dusty regions ? Or only the second one ? In this case, the sentence "in this section we, we thus also scale our dust simulations with a global tuning factor .." should be removed. It is not clear why the Z03 simulation cannot be scaled in the same manner. This is a clear advantage for the simulations with the K14 and the new dust scheme proposed here and this may impact the scores of the comparison with observations.

Section 5.1 The section on the simulated dust emissions mainly comments the differences in the spatial distribution between the different dust emission schemes. The global annual dust emissions should be given. Even if there is no absolute reference, there is an expected range for the dust emission simulated with global models to refer to. The dust emissions simulated with Z03 looks very different from the one reported in the paper by Zender et al. 2003, which are not as discontinuous as the one presented here. Why is there such a difference ?
It could be comfortable for the reader to mention the countries in which the mentioned desert areas are located.

Line 584-585 : It is not clear why the capability of the K14 scheme to simulate major sources in the Sahara and the Arabian Peninsula and some emissions in the USA and Asian desert is attributed to the soil erodibility only. The wind fields obviously plays a role too.

Line 592-593 : It is the first time that a possible bias in the input meteorological field is mentioned ! Why should it impact the K14 scheme only  if the three emission schemes use the same meteorological inputs ?

Line 595-599 : At this stage of the manuscript, there is no objective argument to sustain the fact that the simulated emissions are over-supressed in the USA and the Thar desert or overestimated in the Tibetan Plateau.

Section 5.2 : In the following the simulated DAOD are regionally averaged AOD from Ridley and to the MIDAS data set for the spatial distribution and the daily scores. The two data sets should have been compared in section 4, and if they bring the same information, as the correlation, slope and RMSE between the two data sets suggest, then the most useful of the two should be used in the following.

Line 633-634 : How much is the DAO on the Strzelecki Desert based on MIDAS ? Is it significant ?

Line 644 : The end of the sentence is not clear. Do the authors mean that the input soil moisture in overestimated in the Taklamakan and the Thar Desert ?

Line 647-648 : Are the higher DAOD levels simulated over semi-arid regions in better agreement with MIDAS ?

Line 654-655 : Once again, if the data set from Ridley et al. (2016) and MIDAS are consistent with each other, why not using MIDAS only ?

Line 660 : Part of the difference obtained with between the Z03 dust scheme and the two other can be due to the tuning of the AOD over the dusty region for the two other dust schemes. Has this effect been evaluated ?

Line 680-685 : The authors argue that bias in the wet and dry deposition may explain the underestimation of the lower regional AOD. But since the deposition parametrization and the precipitation fields are the same for the three simulations, this would affect almost similarly the three simulations. This is not consistent with the fact that the new dust scheme seems to reproduce low DOAD quite correctly and not the others. This argument could be sustained by examining the level of agreement of the three schemes in transport region only.

Line 702-705 : Once again this comparison should have been shown in section 4.3 since the MIDAS data set seems to be considered as a "best guess" used as the main reference to evaluate the simulations.

Line 719-720 : Of course evaluating a model against local in-situ measurements point and regionally averaged reanalysis can yield different conclusion. At least because of the difference in the number of measurements in the different dusty regions used for the comparison.

Line 740-743. How have been selected the daily DAOD threshold value (0.25) ? What is meant by "smoothed out" ? What is the final sampling rate in number of days for the 4 years period ? The argument that regions containing less dust signal are less credible is quite weak. Is't it a requirement of dust emission scheme not to produce dust emissions in such areas ? It has been the main problem of many dust emissions scheme for a long time.

A table with the global mean and the regional mean pearson coefficients for the 15 dusty regions would be more convenient and readable than Figure 6. The differences between the schemes could be discussed from this table. The same suggestion applies to figure 7.

Line 843-844 : The very poor correlation between the dust emissions and U* for the Z03 scheme in figure S8 is extremely surprising regarding equation 3. But from the sentence on line 843-844, it seems that the correlation is computed even when the dust emission fluxes are null, which should be the major part of the time. This obviously impact strongly the correlation and makes the result confusing. A correlation restricted to the situations where F>0 and an indication of the frequency of dust emissions should be more relevant to fully understand the sensitivity of the schemes to the meteorological input data.

Figures 8 and 9 : On the lower panel, the color of the points represents the bias, which is already illustrated by the distance to the 1:1 line. Instead of the bias, using symbols to highlight the different regions of measurements (like on the previous figures) would be more useful and informative. This would also be more consistent with the comments of these figures.

Line 864 : It is not clear what "due to the emphasis by the Z03 source mask" means.

Line 885-869 : The first sentence says that the simulated depositions fluxes are of the order of $10^{-4}$ kg m$^{-2}$ yr$^{-1}$ or lower and the measurements order of magnitude of $10^{-4}$, which is quite consistent. But the second sentence suggests a strong bias in the simulations compared to measurements. Is there a mistake in the numbers?

Line 893-895 : To sustain the argument of an overestimated tropical wet scavenging of dust did you check that wet deposition dominates total deposition downwind of the Sahara ?

Section 5.5 : Since the methodology to up-scale the dust emissions was described in section 3, like all the other change in the dust scheme, it is very surprising to understand in the section 5.5 that this was not included in the simulations and in the deep comparison with observations of the previous sections ! In addition, this section shows that the improvement is quite modest. I would thus suggest to remove this section and to use the complete scheme (including the K correction) in the comparisons of section 5.1 to 5.4. The improvement brought by the scaling can be commented among the other factors.

Line 914-921 : The comments on the effect of this scaling raise again the question of the relevance of a "relative" scaling. As suspected, when the emissions are increased in some regions, they automatically decrease in others. Is it really what is expected from an upscaling method that is supposed to better represent high local dust events ?

Line 941-945 : Of course, the difference in the number of AERONET stations for the different dust sources regions has some implications. This is not specific to the evaluation of the scaling method. It could have been mentioned in section 4.2.

**Part 6. Discussion and Conclusions**

Line 978-980 : This is a very strong conclusion. It would have been interesting to quantify how much of the improvement in the level of agreement with observations is due to drag partition and to intermittency, like it has been done for K. Concerning the drag partition, it would have been interesting to evaluate the sensitivity to the roughness data set.

Line 1010 : The sentence is ambiguous : isn't it realistic to have extremely low soil moisture over hyper-arid areas ?

Line 1016-1034 : This section could be removed or largely reduced if the comparison with observations includes all the changes in the dust emission schemes including the K coefficients.

Line 2033-2024 : The fact that this "new process-based emission scheme will still work in ESM across different resolution and in regionally refined models" is very questionable regarding the number of parameters that have been adjusted and that compensate most of the uncertainties on the meteorological input data.

Line 1039-1041 : This is the first mention to a bias due to the meteorological forcing. It should have been mentioned and stated in the presentation of the results. The same remarks apply to the possible overestimation of soil moisture or the incorrect estimation of the roughness length in the Taklamakan (line 1052).

Line 1060 : Concerning the improvement of the dust emission physics, the uncertainties due to the process themselves and those due to the input data should be distinguished. As an example, Zender et al. (2003) argued that they used the drag partition scheme from Marticorena and Bergametti (1995) but since they used a unique global value of Z0 it has almost no impact on the simulated dust emissions. But at this period there was no regional or global data sets of aeolian roughness length available. The use of a unique global Dp to characterize the soil size distribution in sources in this work is another good example.

Line 1095-1115 : These factors are quite far from the focus of the paper and could be removed.

Line 1118-1120 : Evaluating the temporal variability of the dust simulations is much more common in regional modelling and could inspire global modellers.

---

## Author Comment (AC1)

Responses to Reviewers on "A new process-based and scale-aware desert dust emission scheme for global climate models – Part II: Evaluation in the Community Earth System Model (CESM2)" by Danny M. Leung et al. (MS No: egusphere-2023-823)

We thank the reviewers for their careful examinations and thoughtful comments. Our point-by-point responses are provided below. The reviewers' comments are *italicized*, and our new/modified text is highlighted in blue.

Reviewer 1:

*This paper summarizes an important work on improving the dust emission representation in CESM2. Overall the paper is well written and presents sufficient analysis of the dust model performance. I have some comments and questions, described as below.*

*Given the complexity of the model, consider adding a table to list all the mathematical symbols and abbreviations defined in the paper, and/or a flowchart of the model components.*

> Following the reviewer's suggestion, we added a table to list all the mathematical symbols or abbreviations for all major variables defined in the paper. The table is quite big and so we do not put it here; please see the revised manuscript for Appendix.

*Section 3.3:*

*Clarify whether Feff in Eq. 7 represents the fraction of wind drag available for dust lifting or consumed by non-erodible materials, to avoid confusion of the readers.*

> "The fraction of wind drag available for dust lifting" and "the fraction of wind drag remained after consumption by nonerodible materials" are both correct, and both are equal to one minus "the fraction of wind drag consumed by nonerodible materials". We clarify the description for Eq. 7 as follows:
> "… where $F_{eff} \in [0,1]$ is the drag partition factor, the fraction of wind drag available for wind erosion, which is reduced by wind momentum absorption by surface roughness elements (rocks and plants)."

*Section 3.4:*

*What is the physical rationale of representing the intermittent dust emission simply as a scaling factor on the saltation-driven dust emission? Turbulent/convective and saltation-driven dust emissions are two separate physical processes, and have their own separate forms of model parameterizations.*

> The rationale of using a scaling factor $\eta$ to represent the effect of intermittency on dust and sand fluxes is that the friction velocity $u_*$ contains information on turbulent fluctuations. We can thus use $u_*$ to calculate $\eta$, which is determined by the effect of wind fluctuations on initiating and terminating emission (through sweeping across emission thresholds), hence correcting the K14 emission. Please see more of the descriptions in Sect. 3.4 and Comola et al. (2019).
> The reviewer is right that convective turbulent dust emission (CTDE) is a separate process from saltation-driven dust emission that indeed can be parameterized

separately (e.g., Klose et al., 2014). C19 is not trying to represent CTDE but only saltation-driven intermittent dust emission due to instantaneous wind fluctuations as caused by shear- and buoyancy-driven turbulence. To address this comment, we've added a clarification to the text that our scheme does not intend to represent convective turbulent dust emission (CTDE; e.g., Klose et al., 2014). We clarified this in the text in Sect. 3.4:

"Since ESMs cannot explicitly resolve the high-frequency turbulent fluctuations, C19 employed turbulent statistics to estimate the effect of high-frequency turbulent winds on generating dust emissions within a time step. Note that the C19 scheme focuses on incorporating the effect of turbulent wind fluctuations on the saltation-driven dust emission. It does not address the convective turbulent dust emission (CTDE) with direct aerodynamic lifting of dust particles from the land surface, as addressed by other studies (e.g., Klose et al., 2014). Here we briefly describe the C19 scheme…"

*Also, I understand the authors' goal of incorporating as many physical processes as possible, which however may not always improve the model. I am curious whether the importance of dust emission intermittency is evaluated, e.g., via sensitivity analysis? Fig. S2 shows the intermittency factor is much like a global erodible fraction map with high values over subtropical regions, and low values elsewhere.*

Since both reviewers request a sensitivity experiment, we have now added a sensitivity test in Sect. 5.5 to separate the contributions of drag partitioning and intermittency to our final dust emission scheme. Please refer to the new Sect. 5.5 and Fig. 10 for results. Results show that the drag partition effect dominates the contribution of the spatial dust variability on our scheme, while accounting for intermittency expands emissions to semiarid regions and reduces the dust overestimations over hyperarid regions (which is a common problem in ESM as pointed out by Zhao et al., 2022). Both $F_{eff}$ and $\eta$ effects contain temporal variability due to day-to-day vegetation and turbulence variability, thus improving temporal correlations over the K14 scheme (without neither effects) overall to similar extents.

*Section 3.5:*

*The purpose is upscaling correction map (Kc) is explained, but confusing. If the authors intend to capture the finer-scale variability in wind and dust emission, why not just increase the model resolution? I assume computational resources is not a limiting factor since it's not mentioned in the paper.*

The reviewer inferred correctly that the upscaling correction map is intended for when computational resources are indeed a limiting factor, which is especially important for longer runs. We intend to introduce a general methodology that work for ESMs with different configurations, and for longer runs (e.g., CMIP6 runs) it will be very computationally expensive to run in grid resolutions finer than 1°. Using the $\widetilde{K}_c$ maps will enable native-resolution model runs with normal computation expense to achieve performance of high-resolution simulations.

Perhaps more importantly, this method can mitigate the long-standing grid scale dependence problem of modeled dust emissions (Ridley et al., 2013; Feng et al., 2022), which essentially means that using $\widetilde{K}_c$, simulations across different grid resolutions will simulate a similar spatial pattern of emissions. We add this explanation in Sect. 3.5 as follows:

"The final modification in Leung et al. (2023) intends to address the long-standing issue of grid-resolution dependence of ESM-modeled dust emissions (Ridley et al., 2013; Feng et al., 2022; Meng et al., 2022). The grid-scale dependence issue exists because ESMs normally use coarse gridboxes of ~100 km to simulate dust emission, which depends on local-scale processes with typical length scales smaller than 1 km (Marsham et al., 2012; Heinold et al. 2013; Ridley et al., 2013). … The grid-scale dependence problem here thus means that the simulated global dust emission maps are grid-resolution dependent and possess different magnitudes and spatiotemporal variability across resolutions. … There is a need to better upscale low-resolution dust emissions to match the variability of high-resolution emissions, such that dust emission simulations tend to be less resolution-dependent. In addition, upscaling the coarse-resolution dust emission simulations can have the advantage of reducing computational expense while achieving performance similar to that of high-resolution simulations."

*A standalone experiment is performed to calculate Kc. What's the time period of the standalone experiment? Does Kc vary significantly in time, e.g., between seasons and from year to year? If yes, explain why applying a constant Kc is acceptable for the multi-year simulation.*

We performed a simulation for year 2006, which we now added in the second paragraph of Sect. 3.5: "… We conduct a 0.47°×0.62° simulation and a 0.9°×1.25° simulation for year 2006 to yield a fine-resolution emission map $F_{d,f}$ and a coarse-resolution emission map $F_{d,c}$. …"

$\widetilde{K}_c$ processes some seasonality and interannual variability as shown in Fig. S11 of Leung et al. (2023), which is likely due to changes in land-surface characteristics such as vegetation and topography. Although sub-regionally there are some grid-to-grid differences in $\widetilde{K}_c$ from one season to another, the regions possess the same sign of mean $\widetilde{K}_c$ across seasons (e.g., North Africa has mean $\widetilde{K}_c < 1$ throughout the year). Deriving $\widetilde{K}_c$ for each season and for each year certainly makes the upscaling method more accurate and we encourage ESMs to do so. However, the annual $\widetilde{K}_c$ is representative of seasonal $\widetilde{K}_c$. We add the details of these arguments in the third paragraph of Sect. 3.5, which are largely borrowed from Leung et al. (2023): "Leung et al. (2023) proposed this method with the annual $\widetilde{K}_c$ scaling instead of seasonal scaling because, while dust emissions exhibit seasonality and interannual variability (e.g., see Fig. S10 in Leung et al., 2023), the mismatch between $F_{d,f}$ and $F_{d,c}$ are largely due to subgrid spatial heterogeneity such as local topography and soil properties, which are slowly varying variables and partially shared across different model configurations. $\widetilde{K}_c$ in Fig. 1c thus captures the main characteristics of this subgrid variability, even though the ability of $\widetilde{K}_c$ to represent higher-resolution emissions could be improved even further if $\widetilde{K}_c$ was derived specifically for each season, year, or model configuration. In Sect. 5.6 of this paper, we will adjust the spatial distribution of the CESM2 dust emissions for 2004–2008 by multiplying the 0.9°×1.25° dust emissions by the annual $\widetilde{K}_c$ map from 2006."

*Section 5.2*

*Since MIDAS relies on MERRA2 for deriving DAOD, it's subject to MERRA2 aerosol model*

*biases in addition to MODIS sensor/algorithm errors. What's the implication for the comparison with CESM2 model performance?*

We thank the reviewer for this helpful question and suggestion to add the implication in the manuscript. The reviewer is correct that MIDAS DAOD is not a direct satellite retrieval of dust but a reanalysis as a combination of MERRA-2 simulated DAOD-to-AOD ratio and the MODIS total AOD, so there could be some concerns about the accuracy of the spatiotemporal variability of the MIDAS DAOD product.

For spatial variability, since the spatial variability of MIDAS DAOD in Sect. 5.2 is highly consistent with the Ridley et al. (2016) regional DAOD values, we believe that MIDAS overall depicts a good spatial variability of DAOD as shown in Sect. 5.2, although it could accidentally show, for instance, some anthropogenic DAOD over certain megacities of China as discussed in Sect. 5.3.

For temporal variability, Gkikas et al. (2021) showed (in their Sect. 4.1) that the day-to-day variability of MIDAS DAOD is highly consistent with the CALIOP dust retrievals over the Dust Belt, with daily correlation coefficients as high as 0.9 over the oceans (their Fig. 2d). The correlations go down as dust gets further away from the Dust Belt. While both MIDAS and CALIOP have their own uncertainties, they agree well with each other only over the Dust Belt. Thus, the implication is that we will tend to only focus on employing the day-to-day variability of MIDAS DAOD over the Dust Belt for computing the daily correlation maps. We add this implication in the beginning of Sect. 5.3:

"… For MIDAS DAOD, we calculate grid-by-grid daily Pearson correlations between MIDAS and CESM2 DAOD, yielding a global correlation map for each scheme (Fig. 6). We note that since MIDAS is a reanalysis dataset, it itself is also subject to errors due to the MERRA-2 assimilation errors and MODIS instrumental and algorithmic errors. Gkikas et al. (2021) reported that the day-to-day variability of MIDAS DAOD is highly consistent over the Dust Belt (e.g., their Fig. 2d) when compared against the CALIOP satellite retrievals of dust (LIVAS; Amiridis et al., 2013; Marinou et al., 2017). While both MIDAS and CALIOP have uncertainties, we view the day-to-day variability of MIDAS dust as most accurate over the Dust Belt and thus focus our CESM–MIDAS comparison in Fig. 6 over the Dust Belt. In Fig. 6, we show …"

*Section 5.3 Line 785, any proof that the DAOD over Taklamakan has mixed signals from Karakum/Kyzylkum? Those are two desert regions separated by high mountains - there is a very small chance the dust from either one is transported to the other.*
*How does the dust emission amount affect the temporal (daily) correlation of DAOD? Could meteorology be important?*

The reviewer is correct that this claim is speculative. A simpler explanation could just be that the winds over the Taklamakan are often too weak to exceed the emission threshold, resulting in both weakened annual mean DAOD and the seasonality of the DAOD time series. We edit the sentence to frame it as a possible explanation:

"… Meanwhile, a more significant reduction in correlations occurs over China when comparing K14 and our scheme with Z03 (Fig. 6e). Our scheme might produce weaker correlations than Z03's because our scheme with drag partitioning causes $u_{*s}$ to exceed the emission thresholds less often, resulting in both weakened annual mean DAOD and weakened seasonality of the DAOD time series …"

*I find the authors' explanation of the model underperformance over China speculative at best.*

We agree that some discussions of the model performance over the Taklamakan and the Gobi are just hypotheses and are speculative. We deleted the hypothesis for the bad correlations over the Gobi Desert.

Reviewer 2:

*General comment:*
*The objective of this paper is to evaluate of the improvement of the dust emission parameterizations of the Earth Systems Model (ESM) CESMS2 described in a companion paper, by comparison with available synthesis of measurements and with the previous parameterizations included in this ESM.*
*In its present state, thee paper is not totally self-sufficient since the modifications of the dust emission scheme are only briefly recalled and in a very factual way. In particular, the main arguments that justify the selection of a given parametrisation or the values of some key parameters are sometimes missing and only few elements concerning their possible limitations are given. Some sections need further clarification or verifications for consistency with the companion paper. The reader can understand that including this evaluation in the Part I paper would have make it much too dense and too long but the writing of this section should be improved.*

We thank the reviewer for this suggestion. We wanted to minimize the details already described in the companion paper. However, we agree that the description and discussions were perhaps not always self-sufficient. Accordingly, we have expanded the descriptions of the new dust emission scheme especially in Sect. 3 (please see the revised manuscript). We have also addressed the corresponding detailed comments from the reviewer below.

*The work presented in this paper results for several modification of the dust emission scheme that have been included in the CESM2 Earth Systems Model. These modification are mainly related to processes that include soil and surface parameters : the soil size distribution described here by the soil median diameter, the drag partition due to the presence of rocks and vegetation and the soil moisture that both modulate the erosion thresholds. Some of these parameterizations are not new so the main contribution of this work is their tuning, which largely depends on the used input data used for the simulations. The inclusion at the global scale of a parameterization of the intermittency of the saltation, and the differentiation between the fluid and the impact threshold, are two original aspects. The sensitivity of these two factors to the input wind speeds and soil moistures would have deserve more discussion since it has significant impact on the final results. To take into account the limitation due to the coarse spatial resolution of global model, a correction factor is used to scale the dust emissions computed at low spatial resolution to the dust emissions computed at a higher spatial resolution. This correction is also dependent on the initial input data set for both the surface properties (soils, land use, roughness) and the meteorological parameter but the associated uncertainties are not discussed. In addition, this up-scaling approach does not account for smaller scale dynamical features, such as meso-*

*scale convection producing "haboobs" are not properly represented in global scale. This limitation and the consequences on the final scores of the simulations should, at least, be mentioned in the manuscript.*

> We thank the reviewer for these comments. We have addressed the corresponding comments especially related to the original aspects, illustrating the uses and the limits of these implementations. Please see our responses to your detailed comments.

*The two companion papers together represent a valuable piece of work regarding the examination of the different steps involved in the dust emission modelling and the possibility (or not) to improve the representation of the physical processes. The final configuration evaluated in this paper, provides a significant increase of the agreement with observation compared to the two dust schemes used for comparison. However, the new scheme contains several tuning parameters that allows the compensation of uncertainties. In fact, the general approach adopted in this work assign almost all the uncertainty of the simulated dust emissions to the surface parameters. This implies that the final configuration of the dust emission scheme is not independent of the meteorological input fields used for the simulation. Except at the very end of the manuscript, there is almost no mention of the uncertainties on the main drivers of the dust sources and fluxes : the wind velocity and the soil moisture. This can be a relevant approach to tune a model that will then be used in the same configuration than described in the manuscript. But the relevant of the new dust scheme for othet models is thus questionable. It is even questionable for the CESM2 model without nudging to MERRA meteorological reanalysis.*

> Thank you for your positive comments. As per the below detailed comments, we have now focused more on discussing the uncertainties and dependencies of the dust emission schemes to the input meteorological and land surface datasets. We hope to convince you that those several parameters you mentioned below are chosen scientifically rather than tuned to optimize simulation results. Moreover, we believe that the new dust emission scheme has relevance for other ESMs because of the newly added physically based mechanisms, although the choices of those parameter values might indeed need to be revisited in each ESM for scientific or empirical tuning reasons.

*Finally, if the objective of this work is to improve the performed of the CESM2 model in the presented configuration (nudged to MERRA configuration, and thus as close as possible to present day available observations) the approach is relevant, but if the intention is to recommend these modifications as more generic parametrization, it is very questionable. In this context, the objective of "examining how including more Aeolian processes can benefit to dust modeling performance" is not fully reached. I thus encourage the authors to define more precisely their scientific objectives and to adapt their conclusions accordingly. I would thus recommend the publication of this manuscript after some revisions based on these general comments and on the detailed comments described below.*

> We thank the reviewer for the comment. We think that CESM is a climate model that focuses on process representation, climate-timescale changes, and sensitivity experiments, and may not be always realistic in day-to-day timescales like weather models or assimilations. Nudging is probably the best and most realistic way to evaluate the new modifications given the most realistic land-surface and meteorological conditions. The model evaluation with nudging will also be more

indicative of how other weather models or chemical transport models (which prescribe input meteorology) will perform using our new dust emission scheme. Evaluating the ESM with nudging does not mean the additional parameterizations for dust are not useful or not relevant to free runs without nudging. Our scheme contains a better process representation, which should be what ESMs care about. We also showed that our scheme's dust is more sensitive to climate changes, and thus it will help us better understand how dust responds to climate change in long-term climate simulations. To justify the new modifications for a more generic dust emission parameterization, we argue that 1) this dust scheme is more scientifically comprehensive than other dust schemes for ESMs that concern process representation (such that the ESM sensitivity tests and climate experiments will be more scientifically inclusive and correct), and 2) the resulting dust simulation agrees better with observations than other schemes under a realistic (nudged) CESM configuration, both of which are provided in this paper. Please see more detailed responses to the comments below.

Detailed comments :

Abstract :

*Line 28 : the sentence "incorporating a realistic soil size distribution " should be replace by "incorporating a simplified soil size representation", since the soil size distribution is represented by a median diameter and not a distribution and that a unique diameter is used.*

Revised to "incorporating a simplified soil particle size representation" as suggested.

Introduction :

*Line 79-90 : In this section, the limitations of GCM et ASM due to their coarse spatial resolution is discussed but without any link with the small-scale meteorological processes and phenomena that are poorly or not represented in global models such as mesoscale convective systems, sea breezes, low level jets ..*

Thank you for pointing this out. We now added this link to mesoscale meteorology in the sentence as follows: "… Coarse GCMs with horizontal resolutions of 100 km cannot capture local-scale (~1km scale) wind maxima, as well as other small-scale meteorological processes such as mesoscale convective systems (MCS) and low-level jets, leading to an underestimation of emissions over specific regions (Ridley et al., 2013; Gliß et al., 2021; Meng et al., 2021). … "

*Part 2. Description of CESMS and its default dust scheme*

*Line 64-65 : I would suggest to replace "interparticle forces due to soil microbes" by "interparticle forces involved in soil crusting"*

Changed to "interparticle forces involved in soil crusts" as suggested.

*Line 145-149 : Please replace "uses a global optimal diameter of Dp=75μm for the threshold calculation" by " uses a unique diameter Dp=75μm that is optimum for wind erosion since it corresponds to the lowest threshold"*

Revised as suggested.

*Line 129-135 : The authors argued on the necessity to distinguish the fluid and the impact thresholds whose values differ by 20%. Is the precision of the modelled wind speeds higher than 20% ?*

We thank the reviewer for asking this clarifying question. The difference between the fluid and impact threshold could be much larger for semiarid soils since the impact threshold $u_{*it}$ does not increase with soil moisture. We hope to convey to the readers the idea that the effect of switching from fluid to impact threshold in the dust emission equation is significant over semiarid regions. Correspondingly, we edited the statement as follows:

"… Without considering the soil moisture effect $f_m$ on enhancing the fluid threshold (Eq. 1), $u_{*it}$ is about 80 % of the "dry" fluid threshold $u_{*ft0}$ (Sect. 3.4; Kok et al., 2012; Comola et al., 2019). However, if substantial soil moisture is present (e.g., over semiarid regions), the difference between $u_{*it}$ and $u_{*ft}$ could be very large (see Fig. S2a–b) since $u_{*it}$ is not a function of soil moisture (see Eq. 11). Nevertheless, most dust emission schemes in global and regional models employ $u_{*ft}$ as the single threshold …"

*Line 155-156 : Kok, et al. (2014b) tested three values of a (0.5, 1 and 2) but the presented results are for a =1, the values that gives the best agreement with observations.*

The reviewer is correct. We add this detail in the sentence such that it becomes: "where $a$ is a tunable constant typically around 0.5–2 ($a = 1$ was adopted in Kok et al., 2014b) and was set to be $1/f_{clay}$ for tuning purposes in CLM5 (Oleson et al., 2013)."

*Part 3. Modification of the CESM2 dust emission scheme*

*Line 250-254: the description of the "erodibility" and its link with moisture is not clear and U\*st0 is not defined*

$u_{*st0}$ is a constant defined in line 256, but we now further add their values here in text for clarity. Also, to make the definition of soil erodibility and its link to soil moisture clear, we now add in the text the physical meaning of $C_d$ and its relationship with soil moisture effect $f_m$:

"… where $C_{d0} = (4.4 \pm 0.5) \times 10^{-5}$, $C_e = 2.0 \pm 0.3$, and $u_{*st0} = 0.16$ m s$^{-1}$ are constants. The soil erodibility $C_d$ increases with the dryness of the soil and is a pure function of the standardized fluid threshold $u_{*st}$ (and thus $u_{*ft}$) and the soil moisture effect $f_m$. Following Kok et al. (2014b), … "

*Line 261 and 262 : What means "bridging the gap between" scales and how if the value of 0.5 determined ?*

To improve readability and answer the reviewer's question, we change the sentence to "$C_{tune} = 0.05$ is the proportionality constant (previously set in Kok et al., 2014b to scale their global K14 emission to the same global Z03 emission), …"

*Line 283 : As a matter of fact, using γ =1.65 10-4 kg s-2 instead of the value of 3. 10-4 kg s-2 recommended by Shao and Lu (2000) gives almost the same minimum threshold than IW82. So numerically, the change should not be very sensitive.*

We agree with the reviewer. The main reason for using S&L00 was to simplify the threshold calculation in CLM5. We did not want this change to cause a lot of changes in the numerical answers, so choosing $\gamma$ =1.65 $10^{-4}$ kg s$^{-2}$ so that S&L00 and I&W82 gave similar numerical values is ideal.

*Line 265-270 : If the threshold of VAI is set to 1 instead of 0.3, it automatically increases the extend of the dust sources. How much does it increase the extent of semi-arid areas ? Since the total emissions and the global AOD are tuned, how does it impact the spatial distribution of the simulated emissions?*

Using VAI = 1 in our scheme instead of 0.3 indeed enabled the emissions over the semiarid areas, as shown in Fig. 2 (e.g., by comparing K14 with our simulation). In terms of spatial distribution, this change mainly reduces the spatial contrast/gradient of dust emission/AOD between hyperarid and semiarid regions. We now mention this effect in the text as follows:
"… we set $VAI_{thr} = 1$ mainly because observations show that dust is emitted from semiarid regions with VAI > 0.3 (e.g., Okin, 2008). Using $VAI_{thr} = 1$ will thus enable emissions from more marginal dust source regions, which reduces the spatial contrast of dust emissions between hyperarid and semiarid regions."

*\* Line 290-291 : It is not clear how the value of a=2 has been selected. Has the value of a=1 being maintained for the K14 simulation as in Kok et al. (2014b) ?*

We eventually decided to use a value of $a = 2$ for our scheme mainly because CESM2 has higher global soil moisture than other soil moisture products such as MERRA-2/NOAH-MP (see the plot below) and CESM1/CLM4, which resulted in a relatively high soil moisture effect. We decided to use a larger $a = 2$ to suppress the soil moisture effect in order to achieve a similar soil moisture effect as in Leung et al. (2023) using MERRA-2/NOAH-MP. We maintained all the parameter values for K14 as in the original Kok et al. (2014) paper and for Z03 as in the default CESM2, so all the parameter values, such as $a = 1$ and $VAI_{thr} = 0.3$, are not changed in Z03 and K14.

[Figure]

a) MERRA-2/NOAH-MP     b) CESM1/CLM4

c) CESM2/CLM5

Surface gravimetric soil moisture
(kg water / kg soil)

0.01      0.1      1

We add this explanation in text as follows: "We set the tuning parameter for the moisture effect as $a = 2$ in Eq. 2b for our scheme to reduce the moisture effect on enhancing $u_{*ft}$. This is mainly because CLM5 in CESM2 has higher soil moisture across most of the globe than other soil moisture datasets, such as MERRA-2/NOAH-MP (Gelaro et al., 2017) and CESM1/CLM4 (see Fig. S1). For our simulations using the K14 and Z03 schemes in Sect. 5, we will maintain all the default parameter values in CLM5 (including $VAI_{thr} = 0.3$ for both schemes, $a = 1$ in K14 and $a = 1/f_{clay}$ in Z03)."

*Line 298-299 : The conclusions from Martin and Kok (2019) are based on results obtained for soils with median diameter ranging from 0.3 to 0.5 μm in coastal dunes areas and poorly represent the dynamics of fine particles due to sampling limitations. And the conclusion of equal susceptibility concerns particles in the Di/D50, bed size classes. It is constant with the much lower median diameter considered in this study as a reasonable approximation (127μm) and more generally to desert soils ?*

This is a good point. Martin and Kok (2019) did not show that equal susceptibility applies to dust-emitting soils because their field measurements were on sand beds. However, because dust emission is driven by the impacts of saltating sand particles in the size range measured by Martin and Kok (2019), we make the assumption that the movement of those particles (and thus dust emission) is similarly governed by a single threshold set by the median particle diameter. To address the reviewer comment, we now explicitly state this in the paper: "we thus assume here that $u_{*ft}$ for soils containing fine particles is also determined by the median particle diameter because emission of dust aerosols from these soils is driven by impacts of saltating sand particles (e.g., Shao et al., 1993)."

*Line 301 : Please, indicate the number of data sets used investigate the link between PSD and soil texture.*

We change the sentence as follows: "… Leung et al. (2023) then used soil PSD observations from a suite of 14 in-situ soil studies (47 data points) to show that…"

*Line 303-304 : The author argued that the range of minimum fluid threshold (0.204 to 0.298 m.s-1) that correspond to the range of Dmed in the soil data set (40 to 250 μm) can be*

*considered as negligible. But this difference is comparable to the difference between the fluid and the impact threshold (Line 129-131) and the authors argue on the necessity to account for it. This is not consistent. An acceptable argument is that the available data sets does not allow to define precisely the soil size distribution and thus the minimum wind threshold velocity in the arid and semi-arid areas.*

> We agree that the reviewer's suggestion is reasonable. We have added this argument to the text and removed the original argument: "… Regression analysis showed insignificant relationships between $D_p$ and other soil textures and properties, which indicated that the limited variability of the soil dataset did not allow us to precisely define the global $D_p$ distribution and thus its impact on the global distribution of the emission thresholds. Thus, Leung et al. (2023) simplified and approximated the global median $D_p$ by taking a mean across all the $D_p$ observations, which was 127 $\mu$m. Leung et al. (2023) also showed that the $D_p$ uncertainty range of 40–250 $\mu$m translates to a $u_{*ft0}$ range of 0.204–0.268 m s$^{-1}$ using S&L00, …"
> As discussed in lines 129–135, we want to clarify that the fluid threshold and impact thresholds could be very different in semiarid regions (see Fig. S3) because $u_{*ft}$ includes the soil moisture effect (Eq. 1). The ratio $u_{*ft}$ / $u_{*it}$ could be 2–5 in semiarid areas, much bigger than the 0.204–0.268 m s$^{-1}$ spread under the soil dataset uncertainty.

*Line 307 : Finally, what is the value of the minimum U\*t corresponding to Dp=127 μm ? How different is it from the threshold for Dp=75 μm used for Z03 ? The minimum threshold used in Kok et al. (2014) is much lower (0.16 m.s-1). How does it impact the difference in the dust simulations with K14 and the new scheme ? Is there any compensation with the coefficient of tuning of the moisture impact  (a=1 in K14 and a=2 in this study) ?*

> Using S&L00 and assuming air density $\rho_a$ = 1.225 kg m$^{-3}$ and $\gamma$ = 1.65×10$^{-4}$ kg s$^{-2}$, the dry fluid threshold $u_{*ft0}$ without soil moisture effect is 0.204 m s$^{-1}$ for $D_p$ = 75 $\mu$m and 0.215 m s$^{-1}$ for $D_p$ = 127 $\mu$m. We put this information in the companion paper but it is good to mention them here too. The 0.16 m s$^{-1}$ the reviewer refers to in Kok et al. (2014b) is the minimum standardized fluid threshold $u_{*st0}$, which is a separate concept and is different from the fluid threshold $u_{*ft}$.
> The value of 0.204 m s$^{-1}$ should be the minimum threshold value for K14 (although Kok et al. (2014) were using I&W82 instead of S&L00, which the reviewer already pointed out to be the same given this $\gamma$). So, switching from 75 to 127 $\mu$m has a minimal impact on the difference in $u_{*ft}$ between K14 and our scheme; we update the number for $D_p$ not because it changes the answer a lot, but just because it is more scientifically correct. We put this information in the description, as follows:
> "In this study, we introduce the use of $D_p$ = 127 $\mu$m as a global constant for the threshold schemes because it is conceptually more correct than using the optimal diameter of $D_p$ = 75 $\mu$m; however, the resulting value of $u_{*ft0}$ using the S&L00 scheme is  0.215 m s$^{-1}$, which is similar to $u_{*ft0}$ = 0.204 m s$^{-1}$ using $D_p$ = 75 $\mu$m with the I&W82 scheme in Z03."

*Line 339 : According to Marticorena and Bergametti (1995) z0s =Dp/30 and not 2Dp/30.  In Marticorena and Bergametti (1995), X was set to 10 cm,  while a value of 10 m is used here. A short justification of this choice would be welcome regarding the different values used in*

*the literature ( X=10 cm in Marticorena and Bergametti (1995) for small roughness elements and low roughness density ; Y=12,225 cm in MacKinnon et al. (2004) for vegetated surfaces with relatively large and high shrubs and high roughness density; X=40 cm in Pierre et al. (2014) on millet fields and savannah). And why not using different X for the different land cover categories?*

We did not use the value of $z_{0s} = D_p/30$ because this is technically for a homogeneous bed of monodisperse spheres and thus likely underestimates the roughness of real soil beds. We therefore used $z_{0s} = 2D_p/30$ following later studies such as Pierre et al. (2014) and Klose et al. (2021). Pierre et al. (2014) pointed out that the equation with a factor of two, formerly proposed in Sherman (1992) and Farrell and Sherman (2006), is more of a classical sedimentology-based roughness relation and more suited for natural surfaces. We add these citations behind the sentence for reader's references: "where $z_{0s} = 2D_p/30$ is the smooth roughness length (Sherman, 1992; Farrell and Sherman, 2006; Pierre et al., 2014; Klose et al., 2021), …"

$X$ is the distance downstream the location of an obstacle, a length parameter that scales with the internal boundary layer (IBL) height $\delta$ as assumed by Marticorena and Bergametti (1995), and $\delta \sim X$. We adopted $X = 10$ m in the companion paper because we wanted to model a general desertic environment where $\delta \sim X \sim 10$ m such that obstacles are a few meters away from each other. As mentioned by the reviewer, our parameter choice is well within the parameter choices from previous studies which concerned different environments, from ~100 m for large shrubs (Mackinnon et al., 2004) to 10 cm (Marticorena and Bergametti, 1995) for wind tunnel measurements with small and dense obstacle blocks (0.025 m of height, Marshall, 1971). Since our study does not focus on such small obstacles, nor on large shrubs over the Mojave Desert, our choice of $X$ is in between these values. No one choice of $X$ will work for the whole globe because there are many types of environments with many kinds of obstacles, and thus we agree with the reviewer that $X$ should be a function of land type and implicitly space and time (it should really be a gridded map). But, thus far most dust modeling studies have used a global constant of $X$ for simplicity (e.g., Darmenova et al., 2009 used a globally constant $X = 0.1$ m). It would be great to assign different $X$ values for different land types, but it requires further work for development and tuning to suit the CESM land use land cover (LULC) types, and thus is beyond the scope of this work.

We placed the above discussion in Sect. 3.3 when describing the rock drag partition. (To save space and not repeat the same information here, please see the revised manuscript in Sect. 3.3 for the revised description right below Eq. 8.)

*Line 357-375 : From the description given here, the drag partition due to vegetation and to rocks appear as represented independently : there is no possibility to represent a surface covered by rocks and vegetation. Since a data base allows to estimate the fractional area of rocky surface and the fractional area for vegetated surfaces, it is not clear why and average Feff is needed. The dust fluxes could have been computed for each type of surface and then summed up at the grid cell scale.*

The reviewer is correct about our scheme not being able to represent a surface covered by both rocks and vegetation. This formulation assumes a gridbox partially covered by rocks and partially covered by vegetation. This point is discussed in detail in our companion paper (in Sect. 5). We made this choice because 1) most LULC

datasets only indicate areas covered by either rock or vegetation, and we have no information of how rocks and plants overlap, and 2) We separated the treatments of rock and plant partition effects to avoid dealing with the need of adding roughness lengths of rocks and plants because roughness length is not an additive quantity.

The reviewer is also correct about simply summing up the emission fluxes. ESMs could simply sum up emissions from the vegetated area and from the bare area. This Eq. 10 here is an extension of the summation approach to obtain a $F_{eff}$ encompassing both rock and vegetation drag partition effects. Our companion paper provided this option of summing up emissions and we agree with the reviewer that that should be added here too. We thus modified the paragraph as follows:

"… as well as consolidated (gravels and rocks) and unconsolidated (soil) bare land. Leung et al. (2023) proposed to parameterize the total dust emission flus $F_d$ for each grid box as a function of its fractional rock area $A_r$ and fractional vegetation area $A_v$:
$$F_d = F_d\left(u_* F_{eff}\right) = A_r F_{d,r} + A_v F_{d,v} = A_r\, F_d\left(u_* f_{eff,r}\right) + A_v\, F_d(u_* f_{eff,v})$$
(10a)
where $F_{eff}$ is the hybrid drag partition factor. Leung et al. (2023) further formulated the hybrid drag partition factor $F_{eff}$ that encapsulates both rock and vegetation partition effects for these ESMs:
$$F_{eff}{}^3 = A_r\, f_{eff,r}{}^3 + A_v\, f_{eff,v}{}^3 \tag{10b}$$
where $F_{eff}$ is simply the weighted mean of drag partition effects, and the exponent of three is the dust emission exponent $(\kappa + 2)$ of ~3 over deserts. An advantage of this weighted mean approach is that…"

*Line 380-392 : The inclusion of a parameterization that account for the intermittency of the saltation is relevant in terms of physical process. The parameterization from Comola et al. 2019 allows to account for the occurrence of saltation when the wind friction velocity range between the fluid and the impact threshold whose values differs by about 20%. This raises again the question of the accuracy of the simulated wind speed that is the input parameter for the computation of the wind friction velocity. Indeed in the paper by Comola, the intermittency factor is estimated either from in-situ measurements of the wind friction velocity either using an historical 10m wind speed data bases re-scaled to the in-situ measurements. It is not explicitly mentioned in section 3.4 but the application of this parametrization also requires the estimation of the Monin-Obukov length from the meteorological model output which also questions the precision of the estimation.*

We agree with the reviewer that the use of the Comola et al. (2019) scheme requires good accuracy of input boundary-layer variables, such as $u_{*s}$, boundary-layer height $z_i$, and Monin-Obukhov length $L$. We agree that the intermittency scheme will possess some uncertainties given errors in the input wind speed and soil moisture (which means all simulated Earth system processes have errors). As we clarified before, the difference between fluid threshold $u_{*ft}$ and impact threshold $u_{*it}$ could be very large over semiarid regions ($u_{*ft}/u_{*it} > 3$; Fig. S3). For arid regions where $u_{*ft}/u_{*it}$ are close to 1, $\eta$ approaches a step function and thus the dust flux equation behaves like other existing schemes; for this situation, our scheme is subject to the errors in the simulated $u_*$ in a way similar to that of existing schemes. As $u_{*ft}/u_{*it}$ become much larger than 1 (thus a wide intermittency / hysteresis regime) in semiarid regions with moisture, $\eta$ in C19 will be less sensitive to the errors in the simulated $u_*$ (and will be no longer a step function).

*Line 403-405 : Following the previous comment, it appears that decreasing the minimum threshold by using the impact threshold and the intermittency parametrization increase unrealistically high emission in marginal source region. The emissions are thus lowered down by decreasing K. The authors should examine the possibility that the modelled winds are not correct and may exceed the minimum threshold too often. Another source of uncertainty is the fact that the intermittency parametrization was established on a site with a defined wind speed distribution and thus may not be directly applicable in other places with a meteorological context.*

These are good points. Note that, in terms of annual emissions, the use of the intermittency scheme in our scheme in CLM5 still generates lower high-latitude emissions (only ~0.2 %) than suggested by Bullard et al. (2016) (~3–5 %). Therefore, we did not attempt to cap $\kappa$ to lower semiarid emissions because of the use of $u_{*it}$. The reason for capping $\kappa$ is separate from this issue, as illustrated in our companion paper (Leung et al., 2023). It is because Kok et al. (2014a) only saw at most a $\kappa$ of ~6 (or $\kappa$+2 of ~8; their Fig. 4a), but the equation in Kok et al. (2014a) could allow a computed $\kappa$ of > 20 for regions with high soil moisture, which generates unrealistically high dust emissions. Thus, there needs to be a cap to a reasonable limit to prevent unrealistic emissions from exploding and causing the model to crash.

*Line 404-405. The author mention that following Leung et al (2022), they "cap K at 5". But in Leung et al (2022) the comments of equation 13a and 13b state that K is limited to 3.*

We thank the reviewer for pointing out this typo. We change the description here back to three as follows: "… Also, we follow Leung et al. (2023) to cap $\kappa$ at three in Eq. 5c since a large $\kappa$ (e.g., > 10) combined with a small $u_{*it}$ will occasionally produce unrealistically high emissions over semiarid regions …"

*Line 421-422 : It is not clear what Leung et al. (2022) demonstrated on the scale of the dust emission.*

We will cite other papers instead as follows, "… which is a local-scale process with a typical length scale smaller than 1 km (Marsham et al., 2012; Heinold et al. 2013; Ridley et al., 2013). …"

*Line 437-450 : The capability of GCM and ESM that use spatial resolution of the order of 1° is a key question. This was questioned in particular for regions with complex topography such as the region of the Bodele Depression and addressed by regional modelling. The upscaling method proposed here consists in weighted the dust emissions computed at coarse resolution by the one computed at a higher resolution, taking advantage of the fact that the input data are available at this scale. This allows to account for the smoothing of the input parameters but it does not account for small scale meteorological features such as meso-scale convective systems that can dominated the dust emissions in some regions like the Sahel. This limitation should be stated and discussed.*
*It is not clear why the dust emissions computed at the two spatial resolutions are normalized to the total emission to perform the scaling. The normalization implies that if dust emissions are increased in one place, it will be reduced in another. As a matter of fact in figure 1, large areas in the Sahel, South Africa and Australia have correction factors lower than 1. Why not scaling directly the intensity of the computed emissions?*

We agree with the reviewer about the lack of representation of small-scale meteorological features, because ESMs are incapable of representing them even in higher resolutions of 0.5°. Our scaling method only scales the spatial variability of the native-resolution emission $F_{d,c}$ to the spatial variability of the high-resolution emission $F_{d,f}$. If an ESM in 0.5° or finer can resolve MCS and other small-scale features (which is not the case for current ESMs), the $F_{d,f}$ induced by MCS will also be encapsulated in the scaling factor $\widetilde{K}_c$. So, logically, $\widetilde{K}_c$ could allow $F_{d,c}$ to account for small-scale meteorological features that native-resolution ESMs originally cannot. We try to make this clear in the text as follows:

"Finally, we note that the upscaling approach is different from other process-based formulations of saltation processes in Sect. 3.1–3.4, in that Sect. 3.5 is an empirical formulation. The need to employ this scale-aware adjustment will gradually mitigate with increasing ESM horizontal grid resolution, but the importance of the process-based modifications remains regardless of grid resolutions. Since ESMs nowadays at 0.47°×0.62° typically cannot fully resolve smaller-scale meteorological features that drive dust emission (e.g., mesoscale convections and low-level jets), the $\widetilde{K}_c$ derived from the 0.47°×0.62° $F_{d,f}$ will only remedy the scale dependence issue due to the smoothed meteorological inputs in coarser models, but will not represent emissions induced by those finer-scale meteorological features. As ESMs resolve the small-scale meteorology better in the future, $F_{d,f}$ and $\widetilde{K}_c$ will become more capable of capturing emissions generated by the small-scale meteorology."

Then, the main reason for normalizing the $F_{d,f}$ and $F_{d,c}$ to the same global total before constructing $\widetilde{K}_c$ is mainly because all simulations in the CESM2 and many other ESMs will eventually be scaled to match the global mean DAOD of 0.03. It is not very meaningful to multiply $F_{d,c}$ by $\widetilde{K}_c$ and then adjust the global tuning parameter (dust_emis_fact) in CESM2 again.

*Final comment on section 3 : At almost all stages of the improvement of the dust emission scheme, adjustable parameters are involved that have been fixed to obtain realistic simulations. But the criteria used to evaluate what is realistic are generally not described. If it is the level of agreement with the observations presented in the following section, it should be clearly stated.*

This is a good point and we agree that the paper should be transparent about the way in which these parameters were set. There are three adjustable parameters in our scheme: the VAI threshold $VAI_{thr}$, the soil moisture parameter $a$, and the global tuning factor (dust_emis_fact) described in Sect. 5. The only explicit tuning is done with the global tuning factor, which we explain in Sect. 5 is used to tune simulations toward a global mean DAOD of 0.03±0.01. We explained in Sect. 3.1 that we increased the value of $VAI_{thr}$ to 1 because the original value of 0.3 is so small that it even limits emissions coming out of the most hyperarid regions, which is physically inconsistent. Finally, $a = 2$ is used to compensate the high CESM2 soil moisture as explained in the previous comment for line 290–291, which we now explain more clearly in Sect. 3.2 per the earlier reviewer comment on this.

*Part 4. Observational and reanalysis datasets for evaluating the dust emission scheme*

*Line 503-504 : Please precise that the mean AOD used by Kok et al. (2021) based on the data set Adebiyi et al (2020 ) (but not shown in the paper) are estimated from reanalysis that used satellite products.  It would be nice to have the values for the 15 dusty regions in the supplementary material.*

> The reviewer is correct that we should make it clear that the Adebiyi et al. (2020) values are based on reanalysis products. We also agree that these values should be readily available to the reader but since both the Ridley DAOD values and the Adebiyi DAOD values are already listed as a table in Kok et al. (2021), we think it is more efficient to refer readers explicitly to the table in K21 as follows: "… we thus instead use the regional mean DAOD values estimated by Adebiyi et al. (2020), which is based on reanalysis products, with smaller uncertainties over the three SH sources. … This dataset represents seasonal DAOD values averaged over 2004–2008; the Ridley and Adebiyi regional DAOD values are listed in Table 2 of Kok et al. (2021). …"

*Line 523-524 : The overlaying of the AERONET values on figure 5 is unreadable on such small figure. Please add the values and the location of the stations in the supplementary material. In addition, according to the AERONET guidelines for data use and publication, when publishing data from 'many' sites, a general acknowledgement is expected. Citing the appropriate key AERONET papers as well as citing relevant manuscripts pertaining to previously published site data is also expected.*

> Thank you for pointing this out. Accordingly, we added the acknowledgements for the AERONET data in the data availability section: "… The AERONET site AOD data are available on http://aeronet.gsfc.nasa.gov." And in the acknowledgement section: "We thank the AERONET team and site Principal Investigators and managers for the maintenance of the AERONET data record."
> We also added more citations for the AERONET development papers in Sect. 4.2: "Additional observational dust properties are provided by the Aerosol Robotic Network (AERONET; e.g., Holben et al., 1998; Dubovik and King, 2000; Dubovik et al., 2000)."
> Finally, we replotted Fig. 5 for AERONET sites over the Dust Belt and Australia (see revised manuscript). We added a map in Fig. S6 showing the AERONET–SDA AOD values used in Fig. 5, and Table S1 which details the location and names of the selected stations. Please see supplement for Fig. S6 and Table S1, and the updated description in the end of Sect. 4.2.

*Line 527-539 : Except for the temporal resolution, the MIDAS data set does not seems very different by construction than the combined data sets of Ridley et al. (2016) and Adebiyi et al (2020) used by Kok et al. (2021). Was the consistency of the MODAI data set and the global and seasonal average used by Kok et al. (2021) ?*

> No, the MIDAS dataset was not used by Kok et al. (2021), although the regional DAOD values in K21 is quite consistent with MIDAS DAOD.

*Line 540-549 :  Please add, in the supplementary material, the mean values, the location of the stations and the time period for which PM and deposition measurements are available. The relevant references should be given in order to properly credit the measurements producers/owners.*

The information were documented in the Table S2 of Albani et al. (2014) for dust deposition:
https://agupubs.onlinelibrary.wiley.com/action/downloadSupplement?doi=10.1002%2F2013MS000279&file=jame20094-sup-0004-suppinfots01.xls
and Table S2 of Mahowald et al. (2009) for dust PM:
https://www.annualreviews.org/doi/suppl/10.1146/annurev.marine.010908.163727/suppl_file/ma.1.mahowald.pdf
Both datasets were also recently published in Li et al. (2022) on a repository link. We add relevant papers that collected and processed the data in the main text: "We also use site measurements of dust PM (e.g., Prospero and Nees, 1986; Prospero and Savoie, 1989) and dust deposition flux (e.g., Ginoux et al., 2001; Tegen et al., 2002; Lawrence and Neff, 2009; Mahowald et al., 2009; Albani et al., 2014) as climatological datasets for evaluating the spatial variability of dust PM and deposition flux simulations (see data availability section). … "
In the data availability section we put:
"The in-situ dust PM and dust deposition measurements are available through https://zenodo.org/records/6989502/files/LLi2022GMD.Observations.tar.gz?download=1, which was originally from Table S2 of Albani et al. (2014) and Table S2 of Mahowald et al. (2009)."

*Part 5. Model evaluation*

*Line 557-569 : This section is a little bit confusing. Should the reader understand that there is a double tuning, one to reach a global annual DAOD of 0.03 and another to reach an annual mean DOAD of 0.1 in average for the 15 dusty regions ? Or only the second one ? In this case, the sentence "in this section we, we thus also scale our dust simulations with a global tuning factor .." should be removed. It is not clear why the Z03 simulation cannot be scaled in the same manner. This is a clear advantage for the simulations with the K14 and the new dust scheme proposed here and this may impact the scores of the comparison with observations.*

We apologize for the unclear description. To avoid confusion, we now state clearly that all simulations are tuned to match the criterion of having the global mean DAOD of 0.03±0.01 (95 % CI) following Kok et al. (2021) and Klose et al. (2021). We do not need to rescale Z03 because the default CESM simulation using Z03 already produces a global mean DAOD of 0.03. We change the text as follows:
"… Past studies (e.g., Klose et al., 2021; Li et al., 2022) scaled the global dust emissions to produce a global mean modeled DAOD of 0.03±0.01 (95 % confidence interval), which is a global constraint given by Ridley et al. (2016). In this section, we thus also scale our dust simulations with a global tuning factor in the CAM6 namelist variable (dust_emis_fact) like past studies (e.g., Li et al., 2022) did. Here we scaled the simulations with K14 and our new scheme such that their simulated global mean DAOD in CESM2 is 0.03. We did not scale the Z03 simulation since the default CESM2 simulation using Z03 already yielded a global mean DAOD of 0.03 during the CESM2 benchmarking."

*Section 5.1 The section on the simulated dust emissions mainly comments the differences in the spatial distribution between the different dust emission schemes. The global annual dust emissions should be given. Even if there is no absolute reference, there is an expected range*

*for the dust emission simulated with global models to refer to. The dust emissions simulated with Z03 looks very different from the one reported in the paper by Zender et al. 2003, which are not as discontinuous as the one presented here. Why is there such a difference ? It could be comfortable for the reader to mention the countries in which the mentioned desert areas are located.*

As mentioned in Sect. 5.1, the main difference is because the source function or soil erodibility map *S* for Z03 in CESM2 is truncated for all values of S smaller than 0.1. This was done in previous CESM dust studies such as Longlei Li et al. (2022, GMD; their Fig. 2a).

And good point, we now mention the country names in Sect. 5.1 as follows: "The Z03 scheme captures some major and marginal dust sources, such as the Bodélé Depression in Chad, El Djouf in Mali and Mauritania, the Namib Desert in Namibia, the Nubian Desert in Sudan and Egypt, the Taklamakan Desert in China, Patagonia in Argentina, the Karakum/Kyzylkum Deserts in central Asia, and the Strzelecki Desert in Australia."

We also revised Fig. 2 and its description and included the global total emissions in each panel. Please see the revised manuscript.

*Line 584-585 : It is not clear why the capability of the K14 scheme to simulate major sources in the Sahara and the Arabian Peninsula and some emissions in the USA and Asian desert is attributed to the soil erodibility only. The wind fields obviously plays a role too.*

The reviewer is correct that not just soil erodibility but also wind fields play a role too. We have adjusted the sentence, replacing the soil erodibility with K14: "K14 successfully captures emissions not only over major sources such as the Sahara and the Arabian Peninsula, but also emissions over semiarid regions …"

*Line 592-593 : It is the first time that a possible bias in the input meteorological field is mentioned ! Why should it impact the K14 scheme only if the three emission schemes use the same meteorological inputs ?*

We actually discussed in the next paragraph that the biases impact both K14 and our scheme, not just K14. Indeed, biases in meteorological inputs could impact any part of the world, but the bias over the Horn of Africa (HoA) seems quite large and apparent. The bias over the HoA is not in Z03 mostly because the preferential source function *S* already filtered out the HoA emissions.

*Line 595-599 : At this stage of the manuscript, there is no objective argument to sustain the fact that the simulated emissions are over-supressed in the USA and the Thar desert or overestimated in the Tibetan Plateau.*

We agree with the reviewer. We deleted this sentence accordingly.

*Section 5.2 : In the following the simulated DAOD are regionally averaged AOD from Ridley and to the MIDAS data set for the spatial distribution and the daily scores. The two data sets should have been compared in section 4, and if they bring the same information, as the correlation, slope and RMSE between the two data sets suggest, then the most useful of the two should be used in the following.*

This is a good point by the reviewer. We used the Ridley DAOD data instead of MIDAS DAOD for regional evaluation in Fig. 4 because we considered the Ridley dataset as the most well-constrained and best quality-controlled dataset of regional DAOD. In principle, we could also use the MIDAS data set for this, but this would involve us deciding what regions to use and how to calculate the average, making it preferable to use the Ridley 2016 data set for which this has already been done. Ridley was only used in Fig. 4 and were not used anywhere else and so moving the Ridley–MIDAS comparison to Sect. 4 would not clarify any other plots to readers while adding an extra figure. We therefore prefer to leave the Ridley–MIDAS comparison in Fig. 4.

*Line 633-634 : How much is the DAO on the Strzelecki Desert based on MIDAS ? Is it significant ?*

The annual mean MIDAS DAOD ($\mu$) is 0.085 over the Strzelecki Desert; the associated annual mean DAOD uncertainty ($\sigma$) quantified by Gkikas et al. (2021) is 0.041. Thus, the MIDAS DAOD over the Strzelecki is significant to $2\sigma$ (95 % confidence). We plot the $\mu / \sigma$ ratio in Fig. S9b to show more clearly the significance of the MIDAS DAOD over different parts of the world. We also add the DAOD value of the Strzelecki in text: "which is not in agreement with observations which indicate the Strzelecki Desert (central Australia) has the highest DAOD across Australia (annual mean ~ 0.085)."

[Figure]

b) Mean DAOD / mean DAOD uncertainty

*Line 644 : The end of the sentence is not clear. Do the authors mean that the input soil moisture in overestimated in the Taklamakan and the Thar Desert ?*

We only see an underestimated DAOD over the Taklamakan Desert in multiple simulations in Klose et al. (2021) and Li et al. (2022) when using the Kok et al. (2014) dust emission scheme, but we are not entirely sure about the reason behind this. We know that CESM2 soil moisture is higher than other available soil moisture datasets such as MERRA-2/NOAH-MP, but we cannot conclude if CESM2 soil moisture is accurate or if it is overestimated. So, we have deleted the reasoning here: "Both K14 and our scheme underestimate DAOD levels over the Taklamakan/Thar Deserts, which is also seen in other studies employing K14, e.g., Li et al. (2022) and Klose et al. (2021). None of the scheme captures the DAOD levels over the Thar Desert…"

*Line 647-648 : Are the higher DAOD levels simulated over semi-arid regions in better agreement with MIDAS ?*

> We think this is the case, because MIDAS shows some DAOD values (light blue color) over most of the globe, whereas CESM shows DAOD of close to zero (white color). To make the text description clearer, we edit the sentence as follows: "Our scheme also has higher DAOD levels over semiarid regions, which is more consistent with MIDAS DAOD's predictions of more dust across the globe (light blue in Fig. 2d)."

*Line 654-655 : Once again, if the data set from Ridley et al. (2016) and MIDAS are consistent with each other, why not using MIDAS only ?*

> Please see the response above for Sect. 5.2.

*Line 660 : Part of the difference obtained with between the Z03 dust scheme and the two other can be due to the tuning of the AOD over the dusty region for the two other dust schemes. Has this effect been evaluated ?*

> Since Z03 was previously also tuned to have a global mean DAOD of 0.03 during CESM2 benchmarking, the evaluation and comparison between schemes should be fair. We apologize for not making this clear in the original version, but hopefully now we have clarified it in Sect. 5 above (see response to the comment for line 557–569).

*Line 680-685 : The authors argue that bias in the wet and dry deposition may explain the underestimation of the lower regional AOD. But since the deposition parametrization and the precipitation fields are the same for the three simulations, this would affect almost similarly the three simulations. This is not consistent with the fact that the new dust scheme seems to reproduce low DOAD quite correctly and not the others. This argument could be sustained by examining the level of agreement of the three schemes in transport region only.*

> Yes, we were indeed discussing that all three schemes have this tendency of underestimating lower regional DAOD values, which we proposed to be partially due to the dust deposition biases that equally impact all simulations regardless of the emission scheme employed. The main reason our scheme did a better job was that our intermittency scheme enabled more emissions from marginal source regions. To enhance readability, we decided to combine the two paragraphs together and to add the following text:
> "All simulations, regardless of the dust emission scheme employed, show systematic underestimations for lower regional DAOD values and overestimations for higher regional DAOD values … The reasons for the underestimations of lower regional DAOD values could be because the schemes (mainly Z03 and K14) are underestimating dust emissions from marginal source regions with lower regional DAOD values, which is partially corrected in our scheme by producing more emissions from semiarid regions. It could be further because ESMs overestimate wet depositions of dust over tropical oceans …"

*Line 702-705 : Once again this comparison should have been shown in section 4.3 since the MIDAS data set seems to be considered as a "best guess" used as the main reference to evaluate the simulations.*

Please see the responses above for Sect. 5.2 and lines 654-655.

*Line 719-720 : Of course evaluating a model against local in-situ measurements point and regionally averaged reanalysis can yield different conclusion. At least because of the difference in the number of measurements in the different dusty regions used for the comparison.*

We agree with the reviewer. This is also a conclusion we want to point out and remind the readers of. So, we have left this as is.

*Line 740-743. How have been selected the daily DAOD threshold value (0.25) ? What is meant by "smoothed out" ? What is the final sampling rate in number of days for the 4 years period ? The argument that regions containing less dust signal are less credible is quite weak. Is't it a requirement of dust emission scheme not to produce dust emissions in such areas ? It has been the main problem of many dust emissions scheme for a long time.*

There might be some confusions here. We did not use a daily DAOD threshold value of 0.25 but an annual DAOD / AOD ratio of 0.25 to filter out correlation values over non-dusty gridboxes in Fig. 6. The unfiltered correlation maps are shown in Fig. S8, but it is quite messy and thus we want to focus on the dusty regions only. We did not filter out any daily DAOD values < 0.25, and thus the sampling rate is 100 % for dusty regions.
However, we agree with the reviewer that the argument "regions containing less dust signal are less credible" could be improved. To address this, we now make use of the DAOD uncertainty $\sigma$, as quantified by Gkikas et al. (2021). We only show the DAOD correlations of regions where the DAOD exceeds the DAOD uncertainty $\sigma$ (which happened to be mostly the Dust Belt; see our Fig. S9b), which Gkikas et al. (2021) showed to have more trustworthy day-to-day variability (their Fig. 2d and Sect. 4.1) as consistent with the CALIOP dust retrievals. Thus, we use the day-to-day variability of MIDAS DAOD only over the Dust Belt to evaluate our CESM simulations. We add this discussion in the beginning of Sect. 5.2:
"… In Fig. 6, we show the correlation results over gridboxes with MIDAS annual mean DAOD (Fig. 3d) larger than its annual mean DAOD uncertainty (Fig. 8b in Gkikas et al., 2021), which largely corresponds to the gridboxes over the Dust Belt (as shown in Fig. S9b). Gridboxes with MIDAS mean DAOD smaller than the mean DAOD uncertainty are masked out in Fig. 6, and Fig. S8 shows the correlation maps without any masking. Figure S9 shows the MIDAS global DAOD/AOD fraction for 2004–2008 (Fig. S9a) and the ratio of MIDAS mean DAOD to the mean DAOD uncertainty (Fig. S9b). …"

*A table with the global mean and the regional mean pearson coefficients for the 15 dusty regions would be more convenient and readable than Figure 6. The differences between the schemes could be discussed from this table. The same suggestion applies to figure 7.*

We thank the reviewer's suggestion, but we think visualizing numbers into plots is often more effective in conveying messages than compiling a table with numbers. This could just be a matter of personal style preference. We have also included the global mean correlations in each panel.

*Line 843-844 : The very poor correlation between the dust emissions and U\* for the Z03 scheme in figure S8 is extremely surprising regarding equation 3. But from the sentence on line 843-844, it seems that the correlation is computed even when the dust emission fluxes are null, which should be the major part of the time. This obviously impact strongly the correlation and makes the result confusing. A correlation restricted to the situations where F>0 and an indication of the frequency of dust emissions should be more relevant to fully understand the sensitivity of the schemes to the meteorological input data.*

The reviewer is correct that the correlation between $u_{*S} = u_*$ and Z03 emission $F_d$ could be stronger if we use the timestep mean (30-minute mean) time series and truncate all timesteps with zero emissions. However, note that this figure used daily aggregated CESM outputs to calculate the daily correlations. The daily aggregated emission $F_d$ and the daily mean $u_*$ smooths out a lot of diurnal peaks of the 30-minute mean $F_d$ and $u_*$. Fig. S10 exactly shows that global models using Z03 (and K14) will result in dust emission time series that share relatively weak day-to-day covariation with the $u_{*S}$ time series, and one reason could be because of a lot more zero emissions with no variability for schemes using the fluid threshold $u_{*ft}$.

To address this comment, we now make this point more clearly by stating the following in the text: "… our scheme's $F_d$ is further sensitive to the variability of $u_{*S}$ and correlates better with other driving variables than Z03 and K14. Thus, dust emission schemes using $u_{*it}$ will generate emissions that correlate better with the day-to-day variability of $u_{*S}$ than schemes using $u_{*ft}$. Third, …"

If the reviewer is also interested in the daily correlation maps with all days of zero emissions truncated, below we have daily Z03 dust emissions with all $F_d = 0$ days truncated. (without applying the preferential source filter $S$ so that clearer correlation patterns could be seen). Note that we are "truncating all days with zero emissions", which is different from "truncating all timesteps with zero emissions and then aggregate to daily emissions".

[Figure]

As the reviewer pointed out, the magnitudes of the correlation increased moderately for the lower panels without the days with zero Z03 emissions, especially over semiarid regions (with many days of $F_d > 0$). The most hyperarid areas (e.g., the Sahara) do not get obvious changes because daily $F_d > 0$ for most days.

*Figures 8 and 9 : On the lower panel, the color of the points represents the bias, which is already illustrated by the distance to the 1:1 line. Instead of the bias, using symbols to highlight the different regions of measurements (like on the previous figures) would be more useful and informative. This would also be more consistent with the comments of these figures.*

We thank the reviewer for this excellent suggestion. Correspondingly, we have changed the lower panels of Figs. 8 and 9 and use different symbols and colors to represent sites belonged to different broader regions (see figures below). We add this information in the figure descriptions. For instance: "… Figure 8. CESM2 dust $PM_{10}$ concentration (in $\mu g\ m^{-3}$) vs. climatological in-situ $PM_{10}$ measurements (Sect. 4.4) for (a) Z03, (b) K14, and (c) our study. In the bottom panels, sites are labeled over different continents / oceans with different symbols and colors."

[Figure]

*Line 864 : It is not clear what "due to the emphasis by the Z03 source mask" means.*

Because of the need to rescale the global total dust emission such that the global mean DAOD is 0.03±0.01, the discrete Z03 emissions in CESM2 have been rescaled to very high values to compensate for zero emissions over most gridboxes. To address the confusion, we modified the sentence as follows: "… Dust concentrations over the source regions are very high (e.g., the Taklamakan desert and the Australian desert in the top panel of Fig. 8a), due to the very localized and high Z03 emissions over the source regions (Fig. 2a). … "

*Line 885-869 : The first sentence says that the simulated depositions fluxes are of the order of 10-4 kg m-2 yr-1 or lower and the measurements order of magnitude of 10-4, which is quite consistent. But the second sentence suggests a strong bias in the simulations compared to measurements. Is there a mistake in the numbers?*

We thank the reviewer for pointing this out, which was indeed a typo. We correct the sentence as follows: "… simulated deposition fluxes are small with an order of magnitude of ~$10^{-4}$ kg m$^{-2}$ yr$^{-1}$ or smaller (white color), whereas many measurements over those remote locations have an order of magnitude of $10^{-3}$ kg m$^{-2}$ yr$^{-1}$ (light blue). …"

*Line 893-895 : To sustain the argument of an overestimated tropical wet scavenging of dust did you check that wet deposition dominates total deposition downwind of the Sahara ?*

That's a good question. This is more of an explanation from other previous studies that points to the conclusion of overestimated deposition in ESMs, and we only have total but not wet dust deposition data to evaluate this explanation in CESM.

We plot the globally simulated $F_{\text{wetdep}} / (F_{\text{drydep}} + F_{\text{wetdep}})$ ratio below, where $F_{\text{drydep}}$ and $F_{\text{wetdep}}$ are dry deposition flux (kg m$^{-2}$ yr$^{-1}$) and wet deposition flux (kg m$^{-2}$ yr$^{-1}$), respectively. Blue color means that wet deposition dominates the total dust deposition. This plot shows that wet deposition (blue) dominates total deposition for the majority of the tropical Atlantic, while dry deposition could dominate over the coastal northwestern Africa.

$$F_{\text{wetdep}} / (F_{\text{drydep}} + F_{\text{wetdep}}) \text{ ratio}$$

[Figure]

We hope to stay conservative and point out some possible overestimations of dust, and mention a few possible reasons. We rewrite the paragraph as follows:

"There is some underestimation of dust deposition over the downwind regions of Asia (e.g., the extratropical Pacific), likely due to the underestimated Asian dust in K14 and our scheme (but not in Z03 because of its abundant Asian dust). There is also some overestimation of dust deposition over the downwind regions of the Sahara (e.g., the equatorial Atlantic), which could be due to several possible reasons. There could be an overestimation of dry deposition due to an incomplete representation of deposition processes (e.g., Huang et al., 2021; Klose et al., 2021; Li et al., 2022; Meng et al., 2022). In particular, the dry deposition scheme in CAM6 (Zhang et al., 2001) was found to particularly overestimate dry deposition of fine dust (Li et al., 2022). In addition, previous studies indicated a possible overestimated tropical wet scavenging of dust, (e.g., Albani et al., 2014; van der Does et al., 2020). Fig. S12 shows the fraction of wet dust deposition flux to the total dust deposition flux from CESM2 using our scheme."

*Section 5.5 : Since the methodology to up-scale the dust emissions was described in section 3, like all the other change in the dust scheme, it is very surprising to understand in the section 5.5 that this was not included in the simulations and in the deep comparison with observations of the previous sections ! In addition, this section shows that the improvement is*

*quite modest. I would thus suggest to remove this section and to use the complete scheme (including the K correction) in the comparisons of section 5.1 to 5.4. The improvement brought by the scaling can be commented among the other factors.*

This is a good point and we have puzzled over the best way to represent the results of the various modifications to the dust emission scheme in this paper. We ultimately decided to separate this upscaling methodology from other modifications because of several reasons. First, the upscaling method is empirical and thus different from the other modifications, which are additions or revisions of process-based dust emission mechanics. Including this empirical modification with the improvements to the parameterization of dust emission processes would thus obscure the effect that those improvements have on the dust cycle simulations. Moreover, the upscaling approach becomes less useful as model grid resolution improves, while other process-based modifications remain equally important regardless of grid resolution. High-resolution regional models might not need this modification. Third, as pointed out by the reviewer, the overall improvements of employing the scale-aware adjustment are relatively modest.

To communicate more clearly that we separately evaluate the process-based modification and scale-aware adjustment we edit the first paragraph of Sect. 5 to make the contents of each subsection clearer:
"In this section, we evaluate the performance of the different dust emission schemes in CESM2 – Z03, K14, and our scheme – by comparing the spatial and temporal variability of the modeled dust against observations and reanalysis datasets. We first evaluate in Sect. 5.1–5.4 the use of our process-based dust emission scheme (in Sect. 3.1–3.4) without the use of the empirical upscaling method. … Then, we also evaluate in Sect. 5.6 the effects of additionally using the empirical scaling map $\widetilde{K}_c$ (Sect. 3.5) to rescale our scheme's emissions on the resulting CESM2 atmospheric dust simulation, in order to clearly separate the effects of the process-based modification and the scale-aware adjustment."
We also add in this discussion at the end of Sect. 3.5 to make this separation further clearer.

*Line 914-921 : The comments on the effect of this scaling raise again the question of the relevance of a "relative" scaling. As suspected, when the emissions are increased in some regions, they automatically decrease in others. Is it really what is expected from an upscaling method that is supposed to better represent high local dust events ?*

This is a good point and similar to a previous comment from the reviewer. As responded above, in part because of the scale mismatch between the large scale of a grid box and the small scale at which dust emission parameterizations apply, it is not currently possible to determine the dust emission coefficient. Thus, all dust cycle simulations are scaled, explicitly or implicitly, to observations. Following previous work (e.g., Ridley et al., 2016; Klose et al., 2021; Li et al., 2022), we rescale the simulated dust emissions such that the global mean DAOD match an observationally constrained DAOD of 0.03±0.01. Therefore, using an absolute scaling method will not improve the scheme's ability to represent local dust events, and we therefore decided to yield a relative scaling map $\widetilde{K}_c$ which holds the global total dust emission unchanged. The reviewer is right that this is a bit unsatisfying but we see no way around this.

*Line 941-945 : Of course, the difference in the number of AERONET stations for the different dust sources regions has some implications. This is not specific to the evaluation of the scaling method. It could have been mentioned in section 4.2.*

Following the reviewer's suggestion, we add in Sect. 4.2 that: "… We selected 39 stations following Kok et al. (2014b) and Albani et al. (2014), based on the filtering criterion that only the dust-dominant AERONET sites are picked (see Fig. S11 in Kok et al., 2014b for all selected sites). We note that these "dusty" stations are mostly located over the Sahara (as seen in Fig. 5). …"

*Part 6. Discussion and Conclusions*

*Line 978-980 : This is a very strong conclusion. It would have been interesting to quantify how much of the improvement in the level of agreement with observations is due to drag partition and to intermittency, like it has been done for K. Concerning the drag partition, it would have been interesting to evaluate the sensitivity to the roughness data set.*

To address this comment, we have now added a sensitivity test in Sect. 5.5 to separate the contributions of drag partitioning and intermittency to our final dust emission scheme. Please refer to the new Sect. 5.5 and Fig. S13 for results. Results show that the drag partition effect dominates the contribution of the spatial dust variability on our scheme, while accounting for intermittency expands emissions to semiarid regions and reduces the dust overestimations over hyperarid regions (which is a common problem in ESM as pointed out by Zhao et al., 2022). Both $F_{eff}$ and $\eta$ effects contain temporal variability due to day-to-day vegetation and turbulence variability, thus improving temporal correlations over the K14 scheme (without neither effects) overall to similar extents.
We unfortunately do not have another global roughness dataset to evaluate the sensitivity of the drag partition scheme to the choice of roughness dataset. If newer datasets come out, these could be tested and employed in future CESM versions.

*Line 1010 : The sentence is ambiguous : isn't it realistic to have extremely low soil moisture over hyper-arid areas ?*

We agree with the reviewer. We will just point out that there are not enough emissions from the semiarid regions. We change the sentence as follows:
"Fig. 4 shows that CESM2 tends to overestimate DAOD over major sources (e.g., the Sahara) and underestimate DAOD over marginal source regions (e.g., SH sources) and downwind regions (e.g., oceans). This result is consistent with previous findings across multiple ESMs (Zhao et al., 2022), likely due to the insufficient dust emissions coming from the semiarid regions. … "

*Line 1016-1034 : This section could be removed or largely reduced if the comparison with observations includes all the changes in the dust emission schemes including the K coefficients.*

Reading the paragraph again, it seems we discussed too many results and statistics which are already reported in previous sections, and they could be largely cut off as the reviewer suggests. Please see the revised manuscript for the edited paragraph.

*Line 2033-2024 : The fact that this "new process-based emission scheme will still work in ESM across different resolution and in regionally refined models" is very questionable regarding the number of parameters that have been adjusted and that compensate most of the uncertainties on the meteorological input data.*

We would like to clarify that this sentence only suggests that regionally refined models (RRMs) do not need to employ $\widetilde{K}_c$ to use this scheme because the RRMs are already in high spatial resolutions. It will still take some tuning of parameters (e.g., the $a$ parameter for soil moisture effect and the global tuning factor dust_emis_fact) to make the scheme work, just like any other dust emission schemes. We revise the sentence as follows: "… Our new process-based emission scheme can still be employed in ESMs and in regionally refined models (RRMs) with different horizontal resolutions without the use of a scale-aware adjustment."

*Line 1039-1041 : This is the first mention to a bias due to the meteorological forcing. It should have been mentioned and stated in the presentation of the results. The same remarks apply to the possible overestimation of soil moisture or the incorrect estimation of the roughness length in the Taklamakan (line 1052).*

We agree that it is important to point out the biases due to the meteorological forcings. Note that we have mentioned and discussed the possible biases of meteorological forcings in several parts of Sect. 5, such as line 591 on winds and line 633 on soil moisture. We nonetheless agree that we should mention the potential biases in the meteorological forcing when discussing the Horn of Africa (HoA) results. To address this comment we have now added the following (in Sect. 5.2): "our scheme and K14 overestimate dust over Australia and the HoA, which is possibly due to biases in the meteorological variables (e.g., $u_*$ and $w$) of CESM2."

*Line 1060 : Concerning the improvement of the dust emission physics, the uncertainties due to the process themselves and those due to the input data should be distinguished. As an example, Zender et al. (2003) argued that they used the drag partition scheme from Marticorena and Bergametti (1995) but since they used a unique global value of Z0 it has almost no impact on the simulated dust emissions. But at this period there was no regional or global data sets of aeolian roughness length available. The use of a unique global Dp to characterize the soil size distribution in sources in this work is another good example.*

We agree with the reviewer. We mentioned several areas of dust processes that need improvements, but indeed the input data require improvements too. To address this comment, we have added the following to this section:
"6. Observations for dust modeling development: The uncertainties in dust modeling are due to not only the uncertainties in the parameterized dust processes but also the uncertainties in the input data of these parameterized processes. The availability of observations will influence the uncertainties of dust modeling both by entering the simulations as input datasets and by shaping the parameterization development. For instance, Leung et al. (2023) used a global soil particle diameter $D_p = 127\ \mu m$ (Sect. 3.2) for computing the emission thresholds since there were too few site $D_p$ measurements, which hindered the accuracy of the simulation of the global distributions of emission thresholds. We also speculated in Sect. 5 that some of our simulated DAOD biases could be due to biases in the meteorological inputs rather

than the missing physics in the dust scheme. More observations will allow us to develop more accurate parameterizations for dust. For instance, recent coarse dust observations (e.g., Adebiyi and Kok, 2020) justified the importance of and quantified the necessary parameters for formulating the coarse dust modes in ESMs (e.g., Ke et al., 2022; Meng et al., 2022). Having more observations of dust and its dependent variables is highly warranted to reduce the uncertainties of dust simulations by improving the dust schemes and reducing the uncertainties of dependent input variables."

*Line 1095-1115 : These factors are quite far from the focus of the paper and could be removed.*

We prefer to keep the fourth point (speciation of dust), thinking it is relevant to dust emission modeling since developments are still required to transfer bulk emission modeling to speciated emission modeling. The fifth point (chemistry and cloud processing) might be less relevant to dust emission modeling but certainly relevant to the spatiotemporal variability of simulated atmospheric dust, which is the main focus of the evaluation of model performance. Therefore, we add a sentence at the start of the fifth point:
"Chemistry and cloud processing: Having accurate simulations of the modeled spatiotemporal variability of dust requires dust chemistry and dust–cloud interactions in ESMs, because they are crucial for simulating dust aging and dust removal processes. A correct mineralogical representation of dust is essential for… "

*Line 1118-1120 : Evaluating the temporal variability of the dust simulations is much more common in regional modelling and could inspire global modellers.*

We agree with the reviewer and clarify in the text that evaluating the temporal variability is not common for global dust modeling studies: "Finally, while many dust modeling studies focused on improving and evaluating the spatial representation of modeled dust, the importance of evaluating the temporal variability of modeled dust is likely undervalued in global dust modeling studies. … "